# Antigen reactivity defines tissue-resident memory and exhausted T cells in tumors

CD8$^+$ T cells are an important weapon in the therapeutic armamentarium against cancer. While CD8$^+$CD103$^+$ T cells with a tissue-resident memory T ($T_{RM}$) cell phenotype are associated with favorable prognoses, the tumor microenvironment also contains dysfunctional exhausted T ($T_{EX}$) cells that exhibit a variety of $T_{RM}$-like features. Here we deconvolute $T_{RM}$ and $T_{EX}$ cells across human cancers, ascribing markers and gene signatures that distinguish these populations and enable their functional distinction. Although $T_{RM}$ cells have superior functionality and are associated with long-term survival post-tumor resection, they are not associated with responsiveness to immune checkpoint blockade. Tumor-associated $T_{EX}$ and $T_{RM}$ cells are clonally distinct, with the latter comprising tumor-independent bystanders and tumor-specific cells segregated from cognate antigen. Intratumoral $T_{RM}$ cells can be forced toward an exhausted fate when chronic antigen stimulation occurs, indicating that the presence or absence of continuous antigen exposure within the microenvironment is the key distinction between tumor-associated $T_{EX}$ and $T_{RM}$ populations. These results highlight unique functions for $T_{RM}$ and $T_{EX}$ cells in tumor control, underscoring the need for distinct strategies to harness these populations for cancer therapies.

T cell-mediated tumor control is a key facet of cancer immunotherapy, and pinpointing the most effective T cell subtypes for therapeutic targeting is a critical area of research. Two subsets of substantial interest are $T_{RM}$ cells and $T_{EX}$ cells. CD8$^+$ $T_{RM}$ cells are a nonrecirculating memory T cell population that reside in every organ examined[1–3]. Canonical $T_{RM}$ cells develop following the resolution of infection or inflammation, where they can provide rapid, localized immune protection[4,5]. In contrast, CD8$^+$ $T_{EX}$ cells form in the context of chronic infection and cancer, driven by persistent antigen recognition and inflammation[6–8]. T cell exhaustion entails loss of proliferation and function and, although immune checkpoint blockade (ICB) therapies temporarily reinvigorate $T_{EX}$ cell function, their long-term fate remains largely unaltered[9]. $T_{EX}$-phenotype cells also become resident in tumors, probably sharing some aspects of the $T_{RM}$ transcriptional program to limit recirculation[10]. Because many studies use genetic signatures to identify T cell subtypes, their transcriptional similarities have resulted in $T_{RM}$ and $T_{EX}$ cells being conflated in the literature[11–16]. Although tumor-associated $T_{RM}$ cells

may exist as a potential antipode to the dysfunctional $T_{EX}$ population, their identification within tumors and contribution to cancer control has not been adequately addressed.

## Results

### $T_{EX}$ cells share the $T_{RM}$ gene signature

Experimental systems such as parabiosis or transplantation have demonstrated the noncirculating behavior of $T_{RM}$ cells[4,5,17]. Such experiments in mice led to the identification of $T_{RM}$-associated markers including CD69 and CD103 that distinguish $T_{RM}$ cells from circulating memory T cells ($T_{CIRCM}$), with markers partially cross-validated in humans within transplanted organs[18–20]. $T_{RM}$ cells defined by CD69 or CD103 expression are transcriptionally distinct from $T_{CIRCM}$ in humans[1,21,22] and mice[2,23]. At the core of the $T_{RM}$ gene signature exists a transcriptional program designed to halt T cell migration, including downregulation of key regulators of recirculation such as *KLF2*, *S1PR1* and *S1PR5* (refs. 24,25). However, $T_{EX}$ cells also cease migration and become resident in

✉e-mail: lkmackay@unimelb.edu.au

tumors[10]. Thus, we hypothesized that $T_{EX}$ cells use a common transcriptional program to $T_{RM}$ cells to inhibit migration, resulting in considerable transcriptional overlap between these populations. Indeed, the $T_{RM}$ transcriptional profile derived from acute viral infection models (lymphocytic choriomeningitis virus (LCMV) and herpes simplex virus (HSV))[2] correlated significantly with the $T_{EX}$ transcriptional profile derived from chronic versus acute LCMV infection[26] (Fig. 1a). Given this overlap between $T_{RM}$ and $T_{EX}$ cells, we reasoned that $T_{RM}$ gene signatures would identify $T_{EX}$ cells in single-cell RNA sequencing (scRNA-seq) datasets. To test this, we used published data of CD8+ T cells from spleens of mice infected with acute or chronic LCMV[27]. We found the $T_{RM}$ gene signature was most enriched in terminal $T_{EX}$ cells during chronic infection (Fig. 1b–d and Extended Data Fig. 1a). In CD8+ T cells from murine breast cancer (BC) and adjacent tissue[10], the $T_{RM}$ gene signature was highly enriched in tumor-specific $T_{EX}$ cells isolated from tumors, similar to virus-specific cells in adjacent tissue (Fig. 1e,f). Thus, the utilization of $T_{RM}$ gene signatures to identify intratumoral $T_{RM}$ cells results in aberrant $T_{EX}$ cell identification.

Many studies have identified $T_{RM}$ cells in tumors via CD69 and CD103 coexpression[11–16]. CD69+CD103+ CD8+ T cells are present in both noncancerous tissue and tumors, the latter of which probably encompasses a mixed population of $T_{RM}$ and $T_{EX}$ cells (Fig. 1g,h and Extended Data Fig. 1b). Thus, CD69, CD103 and $T_{RM}$ gene signatures all appear insufficient to distinguish $T_{RM}$ and $T_{EX}$ cells should they coexist. Given this overlap in tissue-residency features, we set out to differentiate $T_{RM}$ and $T_{EX}$ populations in human tumors by formulating two testable assumptions. First, that cells enriched for core-residency genes and signatures in healthy, noninflamed tissues are predominantly bona fide $T_{RM}$ cells, while tumor-derived cells expressing these signatures comprise both $T_{RM}$ and $T_{EX}$ populations. Second, that tumor-derived cells expressing residency-associated signatures can be further segregated into $T_{EX}$ and $T_{RM}$ cells by the relative presence or absence of exhaustion-associated gene signatures. If bona fide $T_{RM}$ cells exist in tumors, they would be expected to exhibit transcriptional similarity to $T_{RM}$ cells in associated healthy tissue. To this end, we performed single-cell cellular indexing of transcriptomes and epitopes by sequencing (CITE-seq) with T cell receptor (TCR) profiling on CD3+ T cells from human BC tumors and normal breast tissue (Fig. 1i). CD8+ T cells from BC tumors and breast tissue were distributed over 15 clusters, classified into five main T cell subsets ($T_{EMRA}$, $T_{EM}$, mucosal-associated invariant T cells (MAIT), γδ T cells and CD103+ resident cells) based on protein and transcriptional profiles (Fig. 1j,k and Extended Data Fig. 1c). CD103+ 'resident T cells' shared expression of canonical residency genes including *ZNF683* (HOBIT) and *CXCR6*, and downregulation of *KLF2*, which controls principal tissue egress-promoting gene products[24] (Fig. 1k). As above, we reasoned that tumor-derived CD103+ resident T cells would include both $T_{RM}$ and $T_{EX}$ populations whereas healthy tissue-derived cells would contain primarily $T_{RM}$ cells. Accordingly, we defined two CD103+ clusters (c0, c5) as $T_{RM}$ cells, based on their over-representation (>90%) within healthy tissue, whereas clusters found primarily in tumors and largely absent from healthy tissue (c7, c12, c15), were classified as $T_{EX}$ cells (Fig. 1l,m). Pseudobulk principal component analysis (PCA) analysis confirmed the distinction between $T_{RM}$ and $T_{EX}$ clusters, while highlighting their similarity in the principal component (PC)1 axis, driven predominantly by the downregulation of genes associated with T cell egress including *KLF2* (Fig. 1n and Extended Data Fig. 1d).

Supporting our annotations, published $T_{RM}$ gene signatures[2,21,28] were enriched within both $T_{RM}$ and $T_{EX}$ populations whereas $T_{EX}$ gene signatures[28] were enriched selectively within $T_{EX}$ cells (Fig. 1o). Segregation of $T_{EX}$ and $T_{RM}$ cell populations revealed that, although they share expression of CD103, CD69 and *ZNF683* (HOBIT), and the downregulation of *KLF2*, $T_{EX}$ cells expressed higher levels of CD38, CD39, PD-1, CTLA4, *TIGIT* and *HAVCR2* (TIM3) (Extended Data Fig. 1e–i). These data show that $T_{EX}$ cells within human tumors co-opt a residency program,

and that commonly used $T_{RM}$ cell-associated proteins or gene signatures cannot distinguish between $T_{RM}$ and $T_{EX}$ populations.

## $T_{RM}$ and $T_{EX}$ cells exhibit disparate functional capacities

To disentangle tumor-associated $T_{RM}$ and $T_{EX}$ cells, we developed gene signatures from our BC dataset that could distinguish these populations accurately. Genes were included in the $T_{RM}$ gene signature based on differential expression (DE) in $T_{RM}$ cells compared to other T cell subsets, followed by successive DE analysis between $T_{RM}$ and $T_{EX}$ metaclusters. An analogous approach was used to derive the $T_{EX}$ gene signature (Fig. 2a–c and Extended Data Fig. 2a,b). Whereas both $T_{RM}$ and $T_{EX}$ cells were defined by CD103 expression and *KLF2* downregulation (Fig. 2d and Extended Data Fig. 2c), they were distinguished further by expression of markers including CD94, CD161, CD73, CD38, CD101, CD39, *GNLY* and PD-1 (Fig. 2e and Extended Data Fig. 2d). This enabled reliable discrimination of the two populations via cyclic immunofluorescence microscopy (CycIF[29,30]) and flow cytometry, allowing examination of their intratumoral location and functional properties.

Using a 47-marker CycIF panel, we identified CD103+ KLF2− CD8+ T cells across seven BC patients (Extended Data Fig. 2e,f). These cells were stratified based on the relative expression of GNLY and PD-1 into GNLYHi PD-1Lo and GNLYLo PD-1Hi populations, approximating $T_{RM}$ and $T_{EX}$ cells, respectively (Extended Data Fig. 2e–g). GNLYLo PD-1Hi cells expressed higher levels of CD39 and LAG3, consistent with an exhausted phenotype (Extended Data Fig. 2h). Unbiased clustering of CD103+ KLF2− CD8+ T cells reinforced this distinction: GNLYHi PD-1Lo cells were enriched in cluster C3, which expressed $T_{RM}$-associated markers including CD94, CD7 and NKG2A, whereas GNLYLo PD-1Hi cells dominated cluster C1 with increased TIM3, LAG3 and CD39 expression (Extended Data Fig. 2i–k), supporting our in situ gating strategy. Spatial analysis revealed that both populations localized near panCK+ tumor regions (Fig. 2f and Extended Data Fig. 2l). However, GNLYLo PD-1Hi (approximating $T_{EX}$) cells were, on average, significantly closer to panCK+ tumor cells, suggesting an increased potential for direct tumor interaction (Fig. 2g,h).

Beyond phenotypic and spatial differences, we also observed distinct functional capacities between $T_{RM}$ and $T_{EX}$ populations. Upon ex vivo restimulation, $T_{RM}$ cells exhibited higher production of interferon gamma (IFNγ), tumor necrosis factor (TNF) and interleukin (IL)-2 and showed elevated expression of granulysin (*GNLY*), while $T_{EX}$ cells expressed more granzyme A (*GZMA*) and granzyme K (*GZMK*) (Fig. 2i–k and Extended Data Fig. 3a–e). Moreover, deconvolution of these populations in transcriptomic datasets revealed that enrichment for the $T_{RM}$ gene signature was associated with improved overall survival in BC patients, whereas $T_{EX}$ gene signature enrichment correlated with poorer outcomes (Fig. 2l). This association appeared BC-subtype specific, given that triple-negative BC (TNBC) survival was best predicted by total CD103+ (both $T_{RM}$ and $T_{EX}$) cells, consistent with our previous work (Extended Data Fig. 3f–h)[11]. Further, we tested the association of the $T_{RM}$ and $T_{EX}$ signatures with responses to ICB (pembrolizumab/αPD-1) in the I-SPY2 trial[31], revealing that patients with higher $T_{EX}$ gene signature expression were more likely to achieve a pathologic complete response (pCR) following ICB, whereas elevated $T_{RM}$ signature expression was inconsequential to ICB responsiveness (Fig. 2m). These data indicate that, although $T_{RM}$ cells are associated with positive prognoses in BC, current ICB therapies exclusively target and enhance $T_{EX}$ cell-mediated anti-tumor responses.

## $T_{RM}$ and $T_{EX}$ gene signatures delineate $T_{RM}$ cells across tumors

To determine the applicability of these BC $T_{RM}$ and $T_{EX}$ gene signatures across tumor types, we next developed signatures from CD8+ T cells isolated from liver tumors for comparison. To this end we performed CITE-seq on CD8+ T cells isolated from liver metastases from colorectal cancer (CRC) patients, paired noncancerous liver tissue, and liver tissue from cancer-free donors (Fig. 3a). Two CD103+ resident

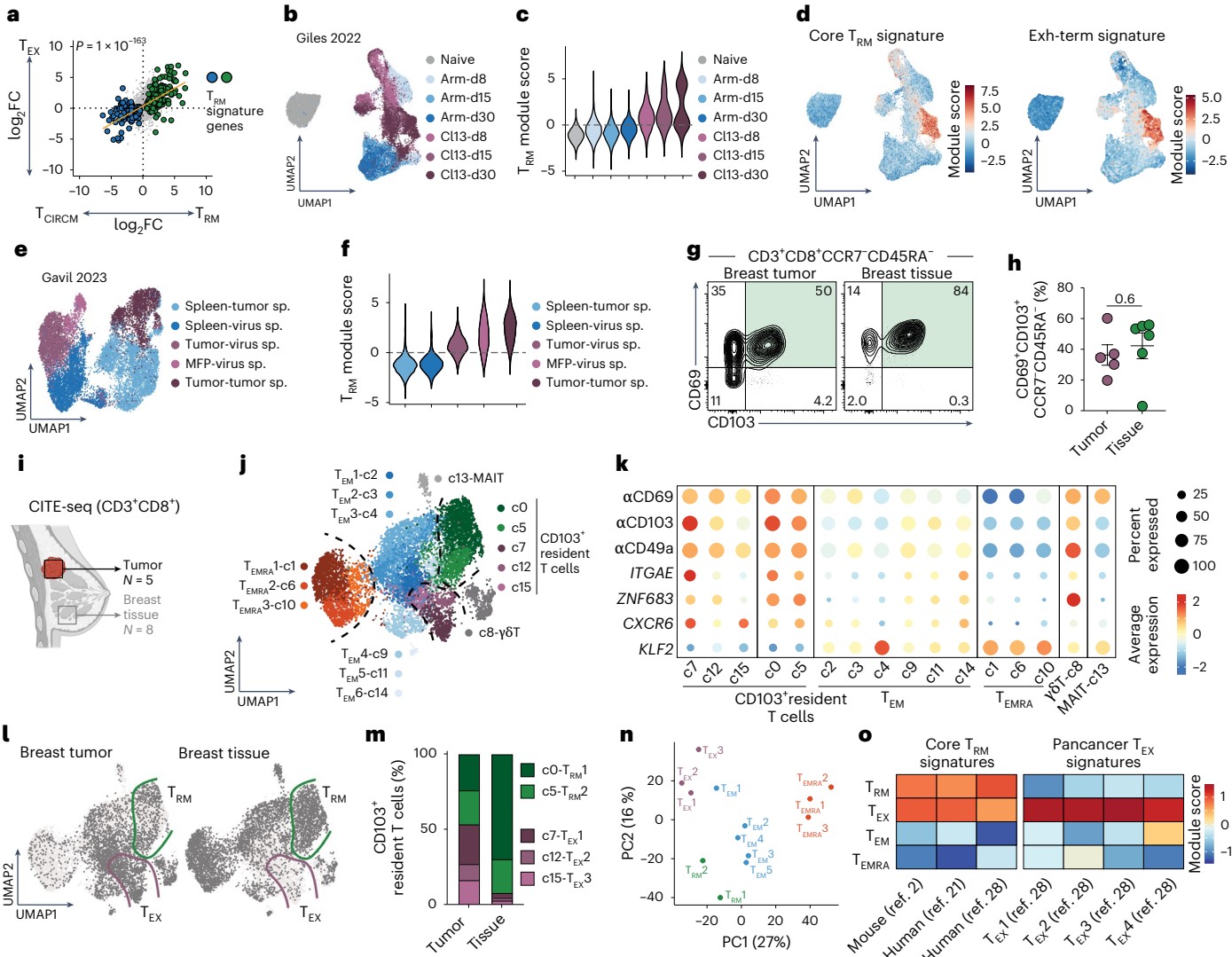

**Fig. 1 | Canonical T_RM proteins and gene signatures do not deconvolute T_RM and T_EX cells. a**, The log₂FC of differentially expressed T_RM (ref. 2) and T_EX (ref. 26) cell genes compared to T_CIRCM. T_RM signature genes[2] are highlighted (green, up; blue, down); *P* value indicates two-sided Fisher's exact test for association. **b–d**, scRNA-seq data from LCMV-specific T cells after acute (Arm) or chronic (Cl13) infection isolated from spleens at day 8 (d8), day 15 (d15) or day 30 (d30) postinfection[27]: UMAP projection of scRNA-seq data annotated by infection and timepoint (**b**), quantification of T_RM module score[2] (**c**) and overlay of core T_RM (ref. 2) and terminally exhausted (Exh-Term)[27] transcriptional signatures on respective populations (**d**). **e,f**, UMAP projection (**e**) and quantification of T_RM module score[2] (**f**) on LCMV-specific P14 (virus-sp.) or tumor-specific (tumor-sp.) OT-I T cells from CITE-seq analysis from EO771-OVA BC tumors, surrounding mammary fat pad (MFP) and spleen[10]. **g,h**, Flow cytometry of CD8⁺ T cells isolated from BC tumors or breast tissue from *N* = 5 BC patients. Representative plots (**g**)

and summary data for percentage of CD69⁺CD103⁺CCR7⁻CD45RA⁻ of CD8⁺ T cells (**h**). Error bars, mean ± s.e.m. **i–o**, CITE-seq of CD3⁺CD8⁺ T cells from primary BC tumors (*N* = 5) and noncancerous breast tissue (*N* = 8): **i**, Schematic. **j**, Data were Harmony integrated, and unified protein and RNA-seq data were represented on weighted nearest neighbors UMAP and colored by cluster. **k**, Expression of respective cell-surface proteins (αCD103, αCD69, αCD49a) and transcripts (*ITGAE*, *ZNF683*, *CXCR6*, *KLF2*) across annotated clusters. **l,m**, CD8⁺ T cells segregated by tissue of origin (**l**) and relative cluster composition of CD103⁺ resident T cells isolated from BC tumors or tissue (**m**). **n**, PCA of pseudobulked clusters annotated in **j**. **o**, Average module scores of published T_RM (refs. 2,21,28) and T_EX (ref. 28) gene signatures by annotated subsets. Data in **c** and **f** were analyzed by two-sided Wilcoxon rank-sum test with continuity correction, all pairwise comparisons *P* < 1 × 10⁻¹⁵. Data in **h** were analyzed by two-sided *t*-test. Illustration in **i** created using BioRender.com.

---

(CD69⁺, CD103⁺, *KLF2⁻*) T cell clusters were identified and denoted as T_EX and T_RM populations as described above, with T_EX cells mostly present in tumor-derived tissue and T_RM clusters present in both tumor and noncancerous tissue (Extended Data Fig. 4a–d). In line with our findings in BC, we found that both T_RM and T_EX populations were enriched for published T_RM but not T_EX gene signatures (Extended Data Fig. 4e). Liver T_RM and T_EX signatures could accurately identify BC T_RM and T_EX cells (Fig. 3b) and, similarly, BC T_RM and T_EX signatures identified liver T_RM and T_EX cells, respectively (Extended Data Fig. 4f). Overall, BC T_RM signature genes were highly enriched in liver T_RM cells (and vice versa), which was also true for the respective T_EX gene signatures highlighting

shared gene expression across tumor-associated T_RM and T_EX cells from disparate tumor types (Fig. 3c and Extended Data Fig. 4g).

Whereas the collection of genes within T_RM and T_EX signatures correlated strongly across BC and liver tumors, the expression of individual genes and surface proteins was not universally consistent. For example, CD94, CD101, CD161 and CD73 were enriched specifically in T_RM cells in BC (Fig. 2e) but not in liver-derived tumors (Extended Data Fig. 4h). Therefore, caution must be applied when extrapolating individual genes and proteins across T cell populations in distinct settings. Nonetheless, by focusing on the leading-edge genes (Fig. 3c and Extended Data Fig. 4g), we defined broadly applicable pancancer T_RM

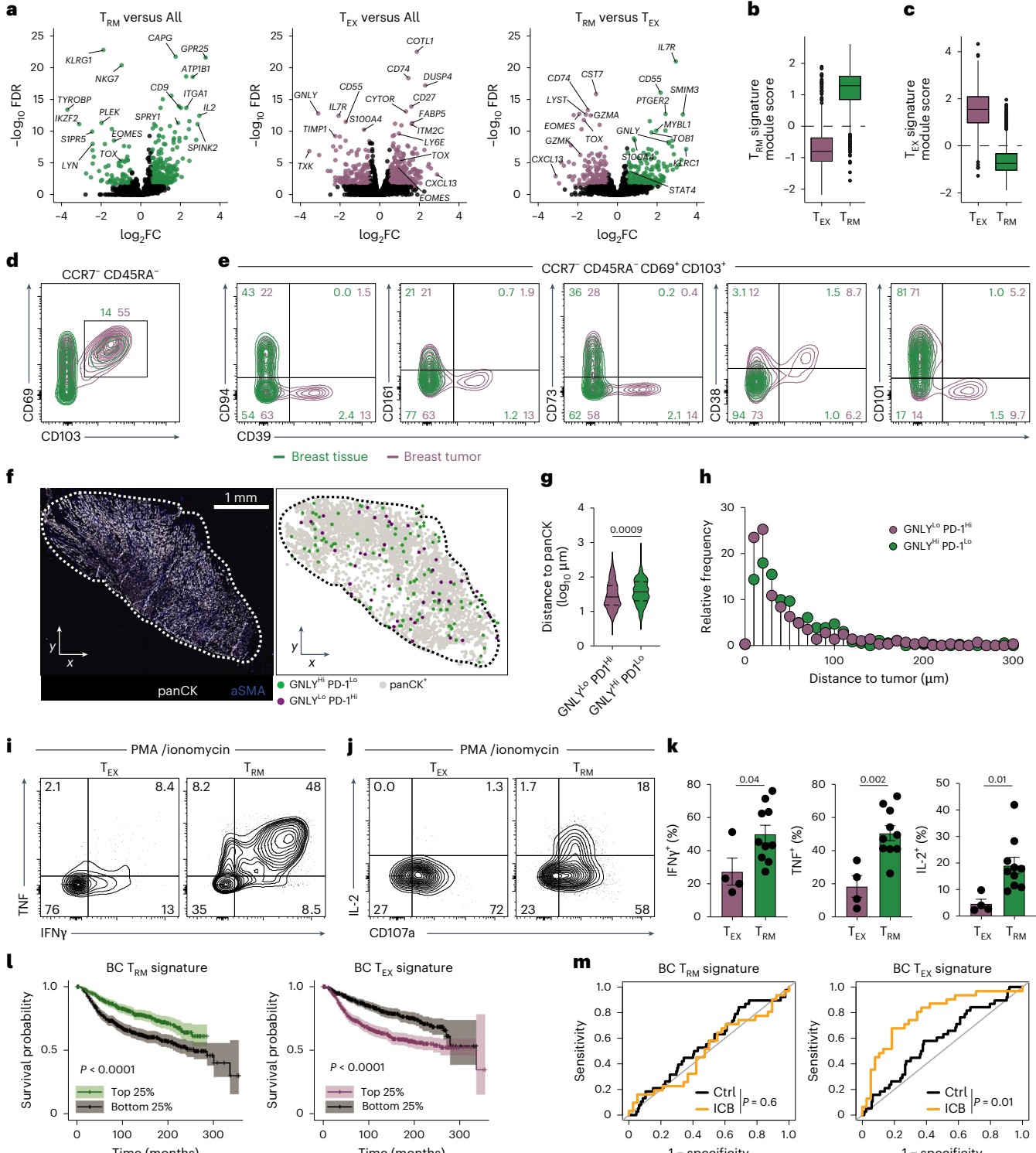

**Fig. 2 | Spatial and functional characterization of CD103⁺ T$_{RM}$ and T$_{EX}$ cells in tumors. a**, Volcano plots showing DE between T$_{RM}$, T$_{EX}$ and all other subsets in BC dataset. **b,c**, Enrichment of T$_{RM}$ (**b**) and T$_{EX}$ (**c**) gene signatures on labeled subsets. Box and whisker plots indicate minimum, maximum, median, and first and third quartiles. **d,e**, Flow cytometry, concatenated from $N = 13$ donors ($N = 11$ with tumor sample and $N = 10$ with healthy tissue). **d**, Expression of CD69 and CD103 on CCR7⁻CD45RA⁻ CD8⁺ T cells. **e**, Expression of T$_{RM}$-defining and T$_{EX}$-defining proteins on CD69⁺CD103⁺ T cells. **f–h**, CyclF imaging of BC tumors. **f**, Representative image of tumor section showing panCK (tumor) and aSMA (stroma) expression, and relative location of respective annotated cell types. Distance to nearest panCK⁺ cell (**g**), and frequency in bins (10 μm) segregated by distance of respective cell types to panCK⁺ cells (**h**), pooled from $N = 7$ donors

(data in **g** analyzed by two-sided $t$-test). Expression of TNF, IFNγ (**i**), IL-2 and CD107a (**j**) on T$_{EX}$ and T$_{RM}$ cells from PMA-ionomycin stimulated CD8⁺ T cells from BC tumors or tissue as per clusters in Extended Data Fig. 3b. **k**, Summary of **i–j**. Donors contributing a minimum of ten cells within a cluster (T$_{RM}$ or T$_{EX}$ isolated from either tumor or healthy tissue) were enumerated and plotted ($N = 4$ T$_{EX}$, $N = 10$ T$_{RM}$), analyzed by two-sided $t$-test. Error bars, mean ± s.e.m. **l**, Survival of BC patients from the METABRIC dataset[58] with the highest (top 25%) T$_{RM}$ or T$_{EX}$ gene signature score enrichments compared to patients with lowest (bottom 25%) gene enrichment scores, plotted on Kaplan–Meier curves with log-rank test. **m**, ROC curves from BC patients, either control (Ctrl, $N = 210$) or treated with ICB (pembrolizumab/anti-PD-1, $N = 69$) from the iSPY trial[31]. Clinical response to ICB associated with T$_{RM}$ or T$_{EX}$ gene signatures, respectively.

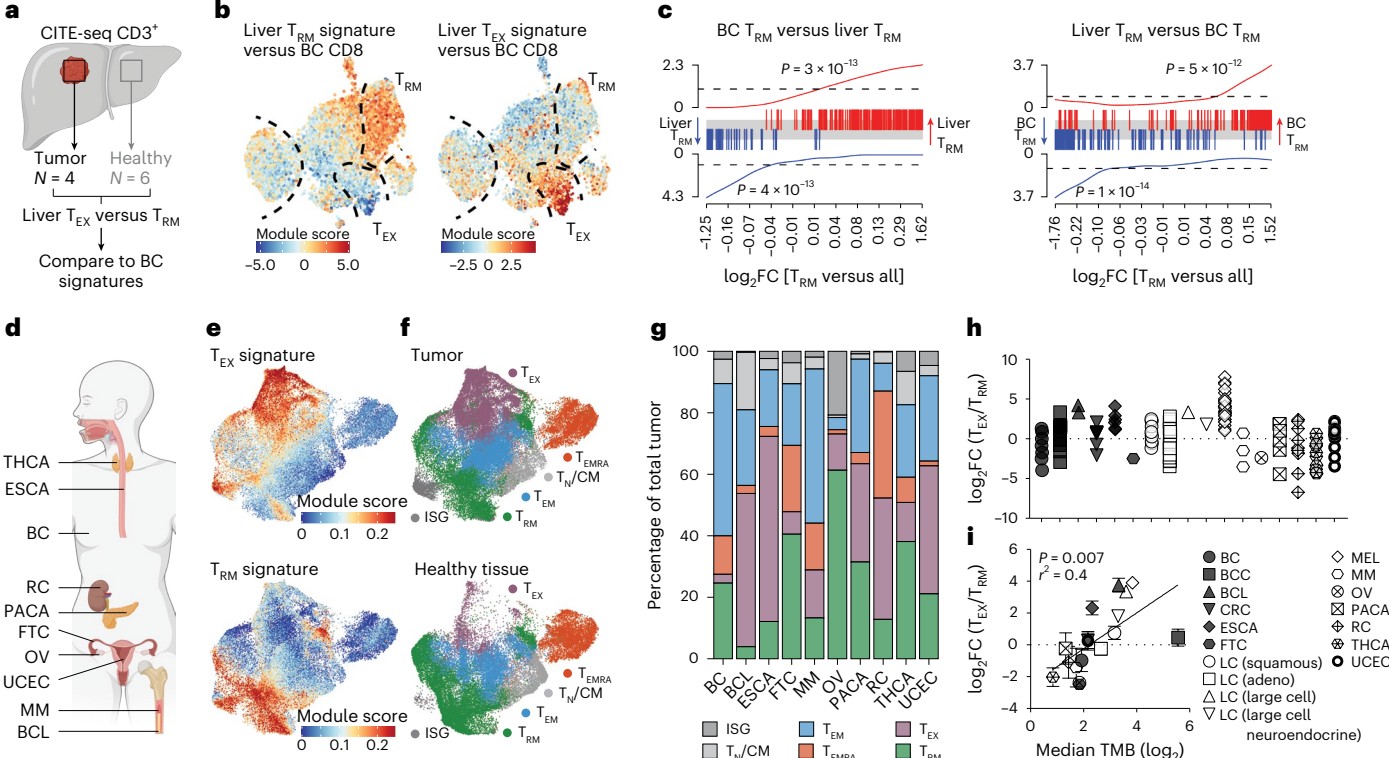

**Fig. 3 | Specific $T_{RM}$ and $T_{EX}$ gene signatures enable $T_{RM}$ and $T_{EX}$ cell identification across tumors. a**, CITE-seq of CD3+CD8+ T cells isolated from secondary liver tumors (CRC patients, $N = 4$) and noncancerous liver tissue ($N = 6$). Liver $T_{RM}$ versus $T_{EX}$ gene signatures were acquired as described for BC signatures. **b**, Overlay of liver $T_{RM}$ and $T_{EX}$ signature module scores on BC dataset. **c**, Gene set enrichment analysis (GSEA) of BC $T_{RM}$ versus liver $T_{RM}$ signatures and vice versa. Barcode plots show GSEA using a running-sum statistic; $P$ value determined by permutation testing. **d–g**, scRNA-seq of tumor-associated CD8+ T cells from various cancers. **d**, Cancers from pancancer atlas[28]. BCC, basal cell carcinoma; FTC, fallopian tube cancer; LC, lung cancer; MM, multiple myeloma;

OV, ovarian cancer; PACA, pancreatic cancer; RC, renal cancer; THCA, thyroid cancer; UCEC, uterine corpus endometrial cancer. **e**, Overlay of refined $T_{EX}$ (top) and $T_{RM}$ (bottom) gene signatures on CD8+ T cells on UMAP of pancancer scRNA-seq dataset[28]. **f**, Tumor-derived (top) and healthy tissue-derived (bottom) CD8+ T cells and annotated clusters. **g**, Relative frequencies of different CD8+ T cell subsets by cancer[28]. **h**, The $\log_2$ ratio of $T_{EX}$ to $T_{RM}$ across different cancers[28,32–37]. **i**, $T_{EX}/T_{RM}$ ratio association with median TMB[38]. Error bars, mean ± s.e.m. $P$ value indicates that the slope of the regression line departs from zero. Illustrations in **a** and **d** created using BioRender.com.

and $T_{EX}$ signatures (Extended Data Fig. 5a,b and Supplementary Table 2). Using these gene signatures, we could distinguish $T_{RM}$ and $T_{EX}$ cells across a range of tumors from several datasets, including a pancancer atlas[28] and additional studies[32–37] (Fig. 3d–g and Extended Data Fig. 5c). A composite of the samples from the pancancer atlas showed that $T_{EX}$ cells were detected predominantly in tumor-derived tissue and largely absent from noncancerous, healthy tissue (Fig. 3e,f). When analyzed by tumor type, the relative abundance of $T_{EX}$ and $T_{RM}$ cells varied, with melanoma (MEL), esophageal cancer (ESCA) and B cell lymphoma (BCL) displaying the highest $T_{EX}$ to $T_{RM}$ ratio among the cancers examined (Fig. 3g,h). This ratio correlated positively with tumor mutational burden (TMB)[38], indicating that neoantigen load may preferentially promote $T_{EX}$ cell formation (Fig. 3i). Overall, these data demonstrate our ability to deconvolute $T_{RM}$ and $T_{EX}$ cells across various tumor settings, facilitating a detailed investigation of their functional and developmental differences.

## Tumor-associated $T_{RM}$ are clonally distinct and enriched for virus-specific cells

Given the correlation between $T_{EX}$ cell abundance and TMB, we hypothesized that $T_{EX}$ and $T_{RM}$ populations may be clonally distinct, reflecting differences in antigen specificity. We therefore examined clonal overlap among the top 100 expanded TCR clones across T cell subsets in both our BC dataset and the pancancer atlas. In both datasets, $T_{RM}$ and $T_{EX}$ cells displayed limited clonal overlap with each other and instead showed greater clonal overlap with $T_{EM}$ cells (Fig. 4a). High Jaccard

dissimilarity scores supported this, indicating that TCR repertoires of tumor-derived $T_{RM}$, $T_{EX}$ and $T_{EM}$ cells were largely distinct (Fig. 4b). Within our BC dataset, expanded clones were shared occasionally across subsets, with most sharing occurring between $T_{RM}$ and $T_{EM}$ or $T_{EX}$ and $T_{EM}$, rather than directly between $T_{RM}$ and $T_{EX}$ (Fig. 4c). Since no public TCRs were present across donors in our BC dataset, liver dataset or the pancancer atlas, we pooled the TCR data for further examination. We assessed clonotype sharing among $T_{RM}$, $T_{EX}$ and $T_{EM}$ cells at various thresholds based on overlapping cell numbers for each clonotype, and found that most clonal overlap among $T_{RM}$, $T_{EX}$ and $T_{EM}$ cells was due to a single shared cell despite substantial numbers of expanded clones (Fig. 4d and Extended Data Fig. 6a). These data indicate that, while a single clone can adopt $T_{RM}$, $T_{EX}$ or $T_{EM}$ phenotypes, there is preferential development of one subset for a given TCR.

Structurally similar TCRs are predicted to recognize similar epitopes. Using TCRdist[39,40], we identified clear structural segregation between $T_{RM}$ and $T_{EX}$ cell TCRs, with substantial structural similarity predicted only among cells of the same phenotype. This suggests that $T_{RM}$ and $T_{EX}$ populations possess distinct epitope specificities (Fig. 4e). To investigate further, we integrated TCR sequences and human leukocyte antigen (HLA) allele expression with VDJdb[41–43] to predict viral reactivity of CD8+ T cells in both our BC dataset and the pancancer atlas. Predicted virus-specific clonotypes were associated predominantly with a $T_{RM}$ phenotype, whereas virus-specific $T_{EX}$ cells were exceedingly rare (Fig. 4f–h). The propensity to adopt a $T_{RM}$ cell phenotype varied depending on viral specificity. Predicted influenza A-specific cells most

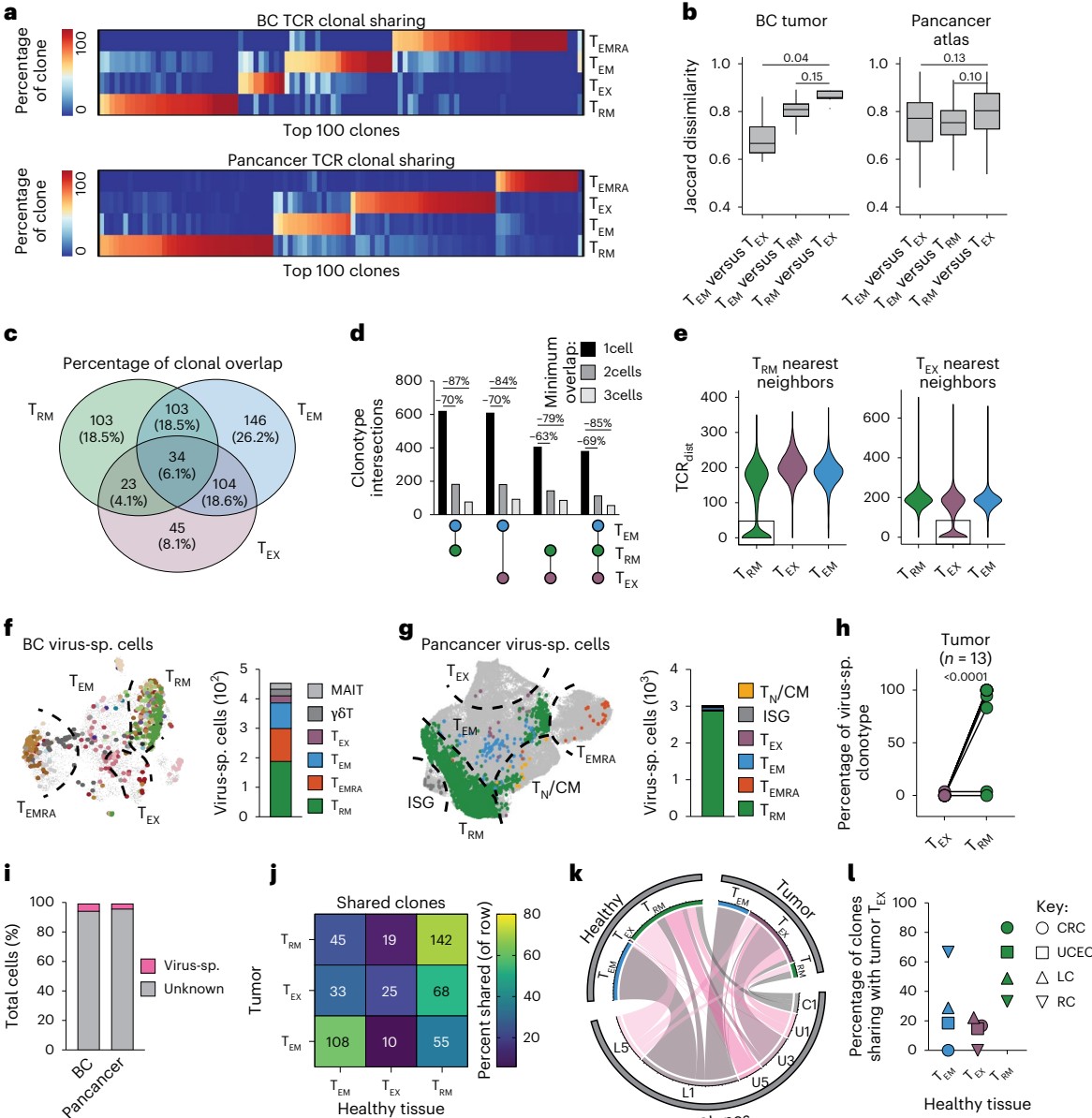

**Fig. 4 | Tumor-associated T_RM and T_EX cells are clonally distinct with discrete specificities. a**, Relative frequency of the top 100 expanded TCR clones within each metacluster from the BC dataset (top) or pancancer atlas (bottom)[28]. **b**, Jaccard dissimilarity index scores (1 − Jaccard index) for expanded (minimum two cells) tumor-derived T cell clones showing dissimilarity between respective populations from BC dataset or pancancer atlas, analyzed by two-sided *t*-test. Box and whisker plots indicate minimum, maximum, median, and first and third quartiles. **c**, Venn diagram indicating clonal overlap between expanded T_RM, T_EM or T_EX clonotypes from BC dataset. **d**, UpSet plot showing the overlap among T_EM, T_RM and T_EX cell types from pooled BC, liver and pancancer datasets, with minimum overlap cutoffs of one, two or three cells. **e**, Violin plots representing TCR_dist analysis of pooled BC, liver and pancancer datasets, showing the distance between T_RM or T_EX with respective subsets, where lower values indicate greater similarity. **f,g**, Distribution and enumeration of cells expressing virus-specific

TCRs as determined by VDJdb[41–43] in the BC (**f**) and pancancer (**g**) datasets. Individual dots indicate virus-specific cells, uniquely colored by clone (**f**) or phenotype (**g**). **h**, Frequency of virus-specific cells within each clonotype that adopt T_RM or T_EX cell phenotypes, analyzed using a two-sided paired *t*-test. **i**, Frequency of virus-specific T cells of total CD8+ T cells in respective datasets. **j–l**, Clonal sharing between tumor and healthy tissue-derived CD8+ T cells[36]. **j**, Clonal sharing between subsets detected across tumor versus healthy tissue-derived CD8+ T cells, filtered on clones identified in both tissues. Heatmap scaled by percentage sharing in each row, numbers indicate shared clones pooled from CRC, UCEC, LC and RC[36]. **k**, Circos plot indicating selected clones from distinct cancers, and number of cells from each clonotype occupying each tissue and phenotype. **l**, Summary of clonal sharing between tumor T_EX and respective phenotype in healthy tissue, split by cancer.

frequently exhibited a T_RM phenotype, whereas EBV-specific T_RM-like cells were rarely identified (Extended Data Fig. 6b,c).

Critically, only a small fraction of total CD8+ T cells in both datasets was predicted to be virus-specific (Fig. 4i), yet these tumor-antigen-independent cells were enriched for the T_RM gene signature. Previous studies have shown that tumor-antigen-specific cells express markers such as CD39 (refs. 44,45), which are characteristic of T_EX cells

in our classification. We therefore hypothesized that tumor-derived clones with a T_EX gene signature, when detected in healthy tissue where tumor antigen is presumed absent, would instead be enriched for the T_RM gene signature.

Given clonal sharing between tumor and healthy tissue-derived cells in our BC and pancancer datasets was limited, probably due to inadequate sampling, we leveraged a dataset in which T cell clones were

identified in both tumor and adjacent healthy tissue[36]. We extracted CD8+ T cell clones shared across sites and annotated them based on our $T_{RM}$ and $T_{EX}$ gene signatures. Consistent with our hypothesis, there was significant clonal sharing between tumor-derived $T_{EX}$ and healthy tissue-derived $T_{RM}$ cells (Fig. 4j,k and Extended Data Fig. 6d,e). Indeed, when clonal overlap was observed, tumor-derived $T_{EX}$ clonotypes were shared more frequently with healthy $T_{RM}$ than with other T cell types (Fig. 4l). Together, these data indicate that the $T_{RM}$ gene signature is enriched in predicted tumor-antigen-independent memory T cells, and tumor-specific cells in healthy noncancerous tissues.

## Antigen drives the distinction between $T_{RM}$ and $T_{EX}$ cells in tumors

Given that putative tumor-antigen-specific T cells preferentially adopt a $T_{RM}$ phenotype in surrounding tissues where tumor antigen may be absent, we hypothesized that tumor-antigen reactivity drives the divergence between tumor-associated $T_{RM}$ and $T_{EX}$ cells. To test whether tumor-specific $T_{RM}$ can form in tumors, we used an orthotopic murine BC (AT3) line expressing the model antigen ovalbumin (OVA). High-dimensional flow cytometry of tumor-infiltrating T cells (TILs) revealed four clusters enriched for tumor-specific OT-I T cells expressing CD69 and PD-1, and differing in markers such as CD39, CD103, TCF1, TOX and TIM3 (Fig. 5a–f and Extended Data Fig. 7a). We annotated these subsets based on phenotypic and functional characteristics. First, we defined resident cells as nonmigratory, using indirect indicators of tissue residency, including intravenous (IV) labeling and FTY720 treatment. Over 90% of TILs were IV-negative, and CD69+ TILs showed reduced IV-staining relative to CD69− TILs (Extended Data Fig. 7b). TIL frequencies were also unaffected by FTY720 treatment (Extended Data Fig. 7c), supporting that most TIL are resident.

Second, we considered cells exhausted if they exhibited functional impairment, and memory cells as those capable of persisting without ongoing antigen stimulation. Accordingly, CD39+ CD103+ cells were annotated as $T_{RM}$ cells based on superior functionality, whereas CD39+ CD103− TOX^Hi cells represented the most dysfunctional, exhausted population (Fig. 5g and Extended Data Fig. 7d,e). A separate CD69+ CD39− CD103− population expressed TCF1, PD-1 and TOX, consistent with progenitor exhausted T cells ($T_{PROG}$), which were depleted selectively when *Tcf7* expression was ablated (Fig. 5e,f and Extended Data Fig. 8a–d). The number of $T_{RM}$ phenotype cells correlated inversely with tumor size at early timepoints (d14) (Fig. 5i), and acquisition of the $T_{RM}$ phenotype correlated with the loss of OVA–GFP expression from tumor cells over time (d23) (Fig. 5j). In contrast, TOX^Hi exhausted cells were enriched in tumors that maintained high tumor-antigen load, consistent with the role of antigen in driving T cell exhaustion (Fig. 5j). Inducible TCR depletion in tumor-specific T cells increased both the frequency and number of CD103+ T cells (Extended Data Fig. 8e–h), further supporting that CD103 expression marks memory T cells in this model, and that tumor-specific $T_{RM}$ cells form or persist preferentially when cognate antigen is absent, consistent with classical T cell memory paradigms.

Given that intratumoral TCR signaling appeared to influence $T_{EX}$ versus $T_{RM}$ cell fate, we next assessed whether reducing TCR signal strength would also influence this distinction. Indeed, OT-3 T cells, which have reduced TCR signaling compared to OT-I T cells, displayed increased CD103 and decreased PD-1 expression (Fig. 5k,l), indicating that low-avidity TCR signaling promotes $T_{RM}$ cell differentiation within tumors. Consistent with the $T_{RM}$ phenotype of virus-specific T cells in human tumors, virus-specific memory T cells generated >100 days after LCMV infection infiltrated AT3-OVA tumors and adopted a $T_{RM}$ phenotype (Fig. 5m,n). Transfer of tumor-specific (OT-I) and nonspecific bystander (gBT-I) effector T cells ($T_{EFF}$) into tumor-bearing mice showed that bystander T cells acquired a $T_{RM}$ phenotype more readily (Extended Data Fig. 8i,j). These bystander T cells expressed significantly lower levels of $T_{EX}$-related proteins including PD-1

(Extended Data Fig. 8j). The development of intratumoral bystander $T_{RM}$ cells required intrinsic TGFβ-signaling (Extended Data Fig. 8k,l), and their efficient tumor entry was dependent on CXCR3 and CXCR6 expression (Extended Data Fig. 8m,n).

We next speculated that the distinct transcriptional $T_{EX}$ and $T_{RM}$ cell signatures derived from patient samples above may correlate similarly with tumor-antigen reactivity. To assess this, we performed scRNA-seq on tumor-specific (gBT-I) and bystander (OT-I) T cells derived from B16-gB tumors and observed discrete clustering of these populations (Fig. 5o,p). Notably, tumor-specific T cells displayed enrichment of the human BC $T_{EX}$ gene signature (Fig. 5q), whereas bystander T cells exhibited high expression of the BC $T_{RM}$ gene signature (Fig. 5r). Consistently, bystander T cells from E0771 BC tumors[10] expressed the $T_{RM}$ gene signature and tumor-specific T cells expressed the $T_{EX}$ gene signature (Extended Data Fig. 8o–q). Thus, the relative presence or absence of intratumoral TCR signaling drives the distinct transcriptional profiles of tumor-associated $T_{RM}$ and $T_{EX}$ cells. Altogether, these data indicate that strong and persistent TCR signaling is antithetical to $T_{RM}$ cell development, suggesting that bona fide $T_{RM}$ cells within human tumors are probably tumor-agnostic bystanders, low-affinity tumor-specific T cells or T cells not in contact with their cognate antigen.

## Tumor antigen drives $T_{RM}$ cells towards a $T_{EX}$ cell fate

The observation that $T_{RM}$ and $T_{EX}$ cells are clonally distinct in human tumors probably reflects intrinsic differences in antigen reactivity, rather than indicating that these populations derive from separate precursor cells. Notably, this clonal distinction does not preclude the presence of tumor-specific $T_{RM}$ cells, nor the possibility of clonal overlap with $T_{EX}$ cells. Our analyses suggested that such clonal overlap could occur when tumor-specific $T_{RM}$ cells are present in tumors or in tissues where cognate antigen is presumably absent. These results did not resolve whether tumor-specific $T_{RM}$ and $T_{EX}$ cells arise from a shared progenitor or from distinct developmental lineages. To address this, we employed the SPLINTR barcoding system[46] to introduce unique barcodes into mono-specific naive or $T_{EFF}$ OT-I T cells before adoptive transfer into AT3-OVA tumor-bearing mice (Fig. 6a). Naive barcoded T cells were generated by transducing OT-I hematopoietic stem cells with SPLINTR lentivirus, followed by intrathymic injection into sublethally irradiated recipients. After 8 weeks, naive T cells pooled from 20 chimeric donors were transferred ($2 \times 10^3$ or $1 \times 10^4$) into AT3-OVA-bearing mice. At 24 days postinoculation, SPLINTR-barcoded OT-I T cells were sorted from spleens and tumors (CD69−, $T_{PROG}$, $T_{RM}$ and CD39+CD103− '$T_{EX}$-like' subsets) and barcode distribution assessed by DNA sequencing.

Minimal barcode sharing was observed between mice receiving $2 \times 10^3$ OT-I T cells, and sharing was detectable only in mice that received $1 \times 10^4$ cells (Extended Data Fig. 9a), indicating that most transferred barcodes were unique, minimizing the possibility of PCR artifacts. We excluded any barcodes identified across several mice before assessing barcode sharing across splenic and tumor populations in individual mice. These analyses showed that antigen-specific, tumor-derived $T_{PROG}$, $T_{EX}$-like and $T_{RM}$ cells were more similar to each other, and distinct from CD69− and spleen-derived populations (Fig. 6b,c). We conducted a parallel experiment using $T_{EFF}$ OT-I cells transduced with SPLINTR-encoding retrovirus, transferred into AT3-OVA-bearing mice. Barcode diversity was again determined across spleen and tumor populations, confirming that barcoded OT-I library pools were unique (Extended Data Fig. 9b). As per the naive T cell experiment, barcode distribution reinforced that tumor-derived $T_{PROG}$, $T_{EX}$-like and $T_{RM}$ cells displayed a high degree of barcode overlap and were distinct from CD69− and spleen-derived populations (Fig. 6d and Extended Data Fig. 9c). Altogether, these data indicate that tumor-specific $T_{RM}$ cells do not arise from a distinct T cell lineage but instead share common progenitors with other intratumoral populations.

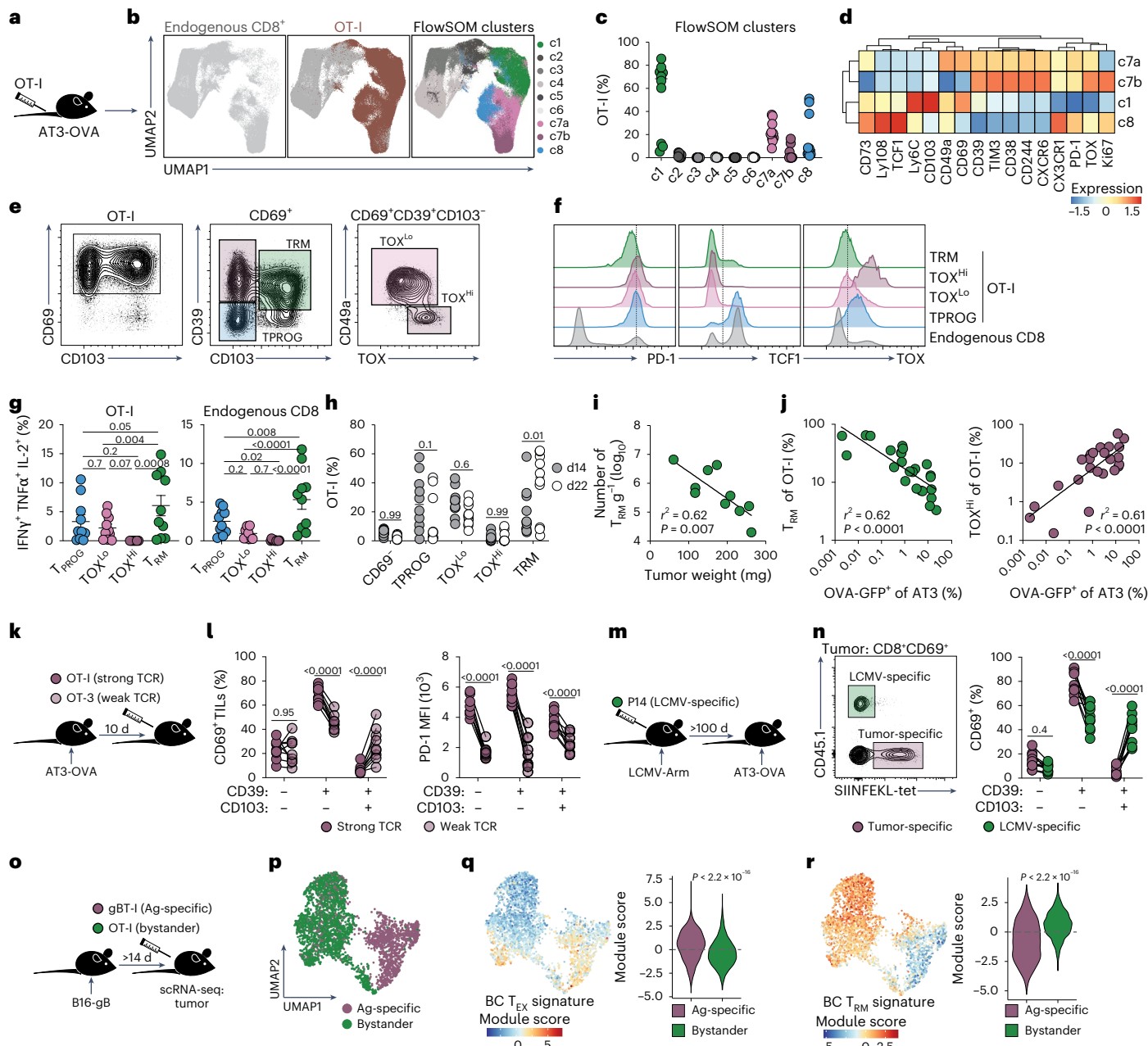

**Fig. 5 | Low-avidity and bystander CD8⁺ T cells preferentially adopt a $T_{RM}$ phenotype in tumors. a–f**, Mice received $1 × 10^4$ naive OT-I T cells and were challenged with AT3-OVA; TILs were analyzed on day 22. **a**, Experimental schematic. **b**, UMAP of CD8⁺ TILs (defined by markers in Extended Data Fig. 7a) showing OT-I T cells and FlowSOM clusters; Cluster 7 was stratified by TOX expression (into c7a ($TOX^{Lo}$) and c7b ($TOX^{Hi}$)). **c**, Distribution of OT-I T cells across FlowSOM clusters. **d**, Relative marker expression by cluster. Representative gating (**e**) and marker expression (**f**) from tumor-derived OT-I T cells. **g**, Cytokine expression post-PMA/ionomycin restimulation. Error bars, mean ± s.e.m. **h**, OT-I frequencies at days 14 and 22. **i**, $T_{RM}$ cell number per gram tumor versus tumor mass at day 14. **j**, Proportion of $T_{RM}$ and $TOX^{Hi}$ OT-I versus %GFP-OVA⁺ AT3 at day 22. **k,l**, Effector OT-I (strong TCR signaling) and OT-3 (weak TCR signaling) T cells were cotransferred into tumor-bearing mice (day 10); intratumoral T cells were analyzed on day 28. **k**, Schematic. **l**, Frequency of CD69⁺ cells and PD-1 expression. **m,n**, LCMV-immune mice (receiving $5 × 10^4$ P14 before infection)

were challenged with AT3-OVA >100 days post-LCMV infection; intratumoral T cells analyzed on day 23 post-tumor inoculation. **m**, Schematic. **n**, Frequencies of CD45.1⁺ LCMV-specific (P14) versus SIINFEKL-tetramer⁺ tumor-specific cells among CD8⁺CD69⁺ TILs. **o–r**, scRNA-seq of bystander (OT-I) and Ag-specific (gBT-I) CD8⁺ T cells cotransferred into mice with B16-gB melanoma tumors. **o**, Schematic. **p**, UMAP of sorted cells colored by transgenic T cell. **q,r**, Overlay of BC $T_{EX}$ (**q**) and $T_{RM}$ (**r**) gene signatures. *P* values from two-sided Wilcoxon signed-rank test. Statistics: **b–j,l**, pooled from two independent experiments; **n**, representative of two experiments, *N* = 10 mice; **b–f**, representative of more than eight total experiments, minimum *N* = 5 mice per group per experiment; **g**, one-way repeated-measures ANOVA with Tukey post-test; **h**, two-way ANOVA with Sidak test; **i,j**, $r^2$ indicates fit of linear regression line and *P* value indicates slope departs from zero. **l,n**, Two-way repeated-measures ANOVA with Sidak test. **l,n**, Connected points from individual mice. Panels **a, k, m** and **o** adapted from ref. 47, Springer Nature America.

---

The developmental association between each cell type left unresolved whether tumor-antigen-specific $T_{RM}$ cells that develop after antigen loss or following spatial segregation from cognate antigen can transition into $T_{EX}$ cells upon antigen re-encounter. To investigate

this, we isolated intratumoral OT-I T cells exhibiting $T_{PROG}$, $T_{EX}$-like or $T_{RM}$ phenotypes from AT3-OVA tumors and retransferred them intravenously (i.v.) into secondary recipient mice bearing AT3-OVA tumors (Extended Data Fig. 9d). Among these populations, $T_{EX}$-like OT-I T cells

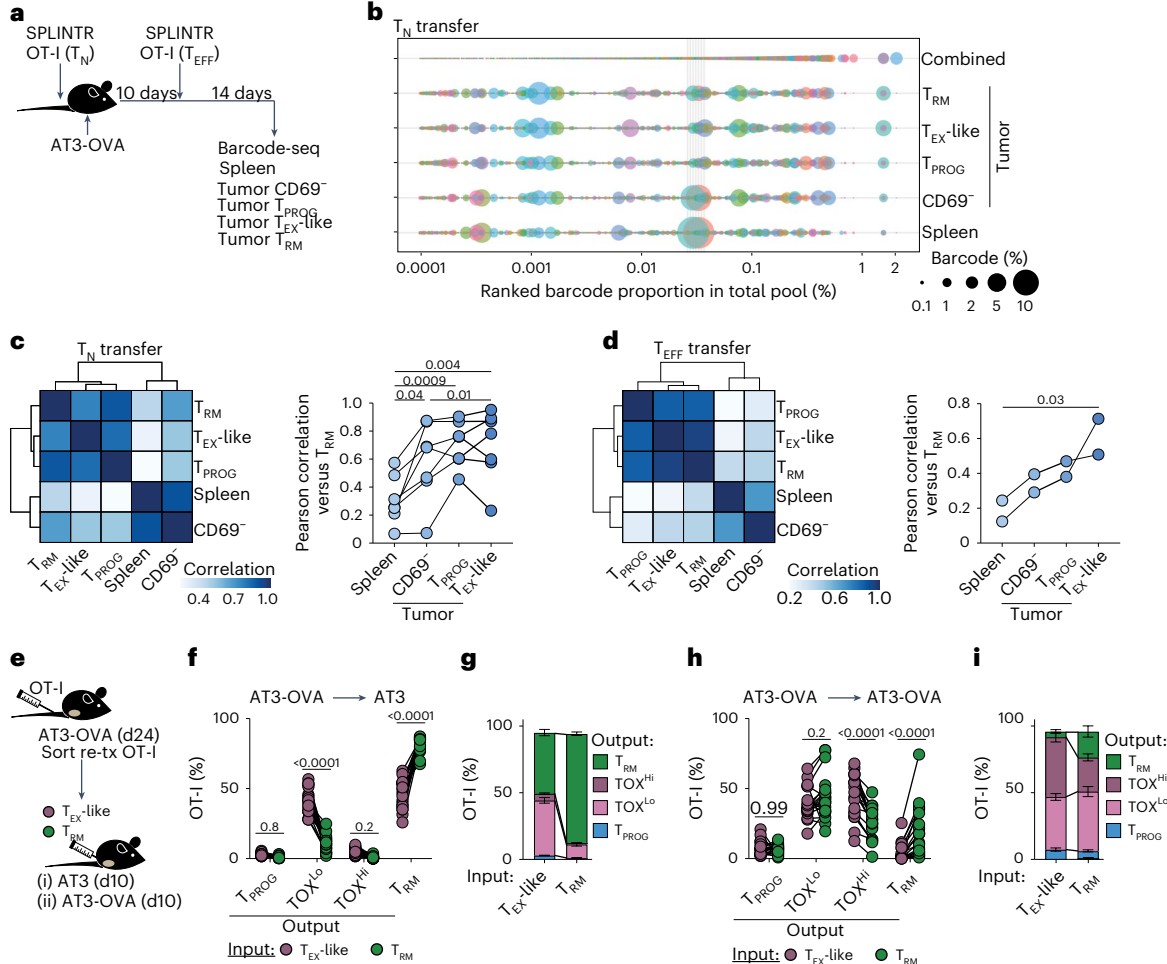

**Fig. 6 | Tumor-associated $T_{RM}$ cells can be driven towards exhaustion.**
**a–d**, SPLINTR barcode-seq of naive ($T_N$) or $T_{EFF}$ OT-I T cells transferred into mice AT3-OVA tumor-bearing mice. **a**, Schematic. TIL populations as defined in Fig. 5e. $T_{EX}$-like population includes $TOX^{Lo}$ and $TOX^{Hi}$ populations. **b**, Barcode representation in a $T_N$ transferred mouse sorted by total pooled barcode where bubble size reflects clone proportion in sample. **c**,**d**, Pearson correlation of barcodes identified in respective populations from representative mouse (heatmaps) and all repeats compared to $T_{RM}$ population (line-plot) from $T_N$ (**c**) or $T_{EFF}$ (**d**) transfer. **e–i**, Sort-retransfer of respective OT-I populations from AT3-OVA mice. **e**, Schematic. Sorted cells were cotransferred intratumorally into recipient mice. **f**,**g**, Percentage of populations isolated from AT3-bearing recipient mice,

split by input cell phenotype and output phenotype in recipient mouse, showing independent mice (**f**) and summary (**g**). **h**,**i**, Percentage of populations isolated from AT3-OVA bearing recipient mice, split by input cell phenotype, and output phenotype in recipient mouse, showing independent mice (**h**) and summary (**i**). Statistics: **b**,**c**, representative of eight independent biological replicates from three experiments; one-way repeated-measures ANOVA with Holm–Sidak post-test; **d**, representative of two independent biological repeats each with technical replicates; one-way repeated-measures ANOVA with test for linear trend. **g**,**i**, Error bars, mean ± s.e.m.; **f**–**i**, pooled from two independent experiments, $N = 19$ total mice; two-way repeated-measures ANOVA with Sidak post-test. Panels **a** and **e** adapted from ref. 47, Springer Nature America.

were significantly less efficient at repopulating tumors compared to $T_{PROG}$ or $T_{RM}$ cells (Extended Data Fig. 9e,f). Due to the relatively low recovery of transferred cells, we performed composite protein expression profiling from each transferred group by experiment, and compared these profiles to $T_{PROG}$, $T_{EX}$-like and $T_{RM}$ OT-I T cells isolated from concurrently analyzed primary tumors (Extended Data Fig. 9g,h). This analysis revealed that all retransferred populations in secondary tumors most closely resembled the $T_{EX}$-like phenotype observed in primary tumors.

To determine whether differences in trafficking influenced the ability of cells to repopulate tumors and adopt distinct phenotypes, we sorted congenically distinct $T_{EX}$-like and $T_{RM}$ OT-I T cells from primary AT3-OVA tumors and cotransferred them directly into secondary tumors that either expressed or lacked OVA (Fig. 6e). Notably, direct intratumoral transfer eliminated the repopulation advantage observed previously for $T_{RM}$ cells, potentially reflecting their enhanced ability to repopulate distal sites following i.v. transfer (Extended Data Fig. 9i). In the absence of cognate antigen, transferred $T_{RM}$ cells largely retained their

phenotype, whereas approximately 50% of the heterogenous $T_{EX}$-like population (input, comprising $TOX^{Lo}$ and $TOX^{Hi}$) could adopt the $T_{RM}$ phenotype (Fig. 6f,g). The dysfunctional $TOX^{Hi}$ population was absent within AT3 tumors lacking antigen, consistent with data showing that these cells are sustained in the presence of antigen (Fig. 6f,g). Conversely, when the same populations were transferred into AT3-OVA tumors, a fraction of $T_{RM}$ cells maintained their phenotype; however, most transitioned to a $T_{EX}$-like state, including a substantial proportion acquiring the $TOX^{Hi}$ phenotype (Fig. 6h,i). Thus, $T_{RM}$ cells can be driven towards terminal exhaustion within the antigen-rich environment of secondary tumors. Overall, these data indicate that, unlike nontumor reactive bystander $T_{RM}$ cells, tumor-specific $T_{RM}$ and $T_{EX}$ populations can arise from a common origin, with tumor-specific $T_{RM}$ cells having the capacity to be driven towards exhaustion upon chronic antigen re-encounter.

## Discussion

Our study reconciles two critical T cell subsets associated with tumor control, namely CD8$^+$ $T_{RM}$ and $T_{EX}$ cells. We show that these subsets

are distinct T cell populations that have been conflated in the literature. This has arisen due to $T_{EX}$ cells engaging a residency program that relies on the same transcriptional machinery used by $T_{RM}$ cells to inhibit tissue egress. Thus, $T_{RM}$ gene signatures developed without $T_{EX}$ cell consideration cannot disentangle these two populations. This study resolves this issue by establishing broad gene signatures that reliably distinguish these subsets across human tumors. Our approach to disentangle $T_{RM}$ and $T_{EX}$ cells in tumors provides a framework for identifying other phenotypically and transcriptionally similar populations. We also emphasize that, just as $T_{RM}$ cells exhibit tissue-specific phenotypes[47–49], the features of tumor-associated $T_{RM}$ and $T_{EX}$ cells also vary between tumor types. Accordingly, the use of specific markers and gene signatures should be validated in the relevant biological context.

The deconvolution of tumor-associated $T_{RM}$ and $T_{EX}$ cells highlights their potential to play distinct roles in anti-tumor immunity and their differential impact on ICB responses. High $T_{EX}$ gene signature expression in BC correlates with positive ICB outcomes consistent with the elevated expression of PD-1 and CTLA4 by $T_{EX}$ cells. The association of high TMB with increased $T_{EX}$ cell frequency suggests that increased new tumor epitopes enable $T_{EX}$ cell development. $T_{EX}$ cells, as defined in this study, consistently co-express CD39 and CD103—markers known to enrich for tumor-specific CD8+ T cells[44,45,50]. High TMB is generally associated with better ICB responses, including BC subtypes such as TNBC, where TMB predicts favorable outcomes to therapies such as pembrolizumab[51–54]. The observed link between high $T_{EX}$ gene signature scores, TMB and ICB efficacy underscores the critical role of $T_{EX}$ cells in tumor control.

We found that most tumor-associated $T_{RM}$ cells identified in humans are clonally and developmentally distinct from $T_{EX}$ cells. Tumor-antigen agnostic T cells, including T cells specific for previous infections, can adopt a residency phenotype within tumors. Consistently, we show that TCR signaling is antithetical to bona fide $T_{RM}$ cell formation in tumors, aligning with the conceptual notion that immunological memory can develop only following clearance of cognate antigen. However, we also reveal that tumor-antigen-specific $T_{RM}$ cells can develop in settings of reduced TCR signaling and diminished antigen sensing. Tumor-specific $T_{RM}$ cells probably exist in lower numbers compared to tumor-specific $T_{EX}$ cells as it is expected that they have encountered less antigen, and therefore have had reduced proliferative bursts, accounting for the limited clonal sharing that we observe between $T_{RM}$ and $T_{EX}$ cells in human tumors.

Although $T_{RM}$ gene signatures are associated with improved overall survival in BC patients, the mechanisms underlying $T_{RM}$-mediated tumor control remain unclear. The accumulation of $T_{RM}$ cells may coincide with other features of tumor control, such as tumor antigen clearance. However, our analyses suggest that tumor-specific $T_{RM}$ cells persist in healthy tissues adjacent to tumors, raising the possibility that they contribute to long-term immune surveillance and protection against tumor recurrence. The clinical benefit of tumor-specific $T_{RM}$ cells may lie in their ability to maintain equilibrium with residual cancerous cells, preventing tumor recurrence, or in prophylactic settings following vaccination[55]. Given that these cells form following antigen clearance and can re-enter tissues and repopulate distal sites, the presence of $T_{RM}$ cells in human tumors may reflect an effective immune response capable of providing durable protection at both primary and metastatic sites.

Finally, substantial populations of bona fide, nonexhausted, tumor-agnostic $T_{RM}$ cells can be identified in diverse tumors. These cells maintain superior functionality in humans and murine models but are not targeted by current immunotherapies. Thus, approaches to activate bystander $T_{RM}$ cells through TCR-independent pathways or administration of viral peptides[45,56,57], or to mobilize functional tumor-specific $T_{RM}$ cells, combined with current $T_{EX}$ cell-targeted ICB therapies could raise the ceiling of effective anti-tumor responses.

## Online content

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

Thomas N. Burn ⓘ [1], Jan Schröder[2], Luke C. Gandolfo ⓘ [1,3,4], Maleika Osman ⓘ [1], Elanor N. Wainwright[5,6], Enid Y. N. Lam ⓘ [5,6], Keely M. McDonald ⓘ [1], Rachel B. Evans[1], Shihan Li[2], Daniel Rawlinson ⓘ [2], Lachlan Dryburgh[2], Ali Zaid ⓘ [1], Zoltan Maliga[7], Dominick Schienstock[1], Philippa Meiser[1], Hyun Jae Lee[1], Hongjin Lai[1,8], Marcela L. Moreira ⓘ [1], Pirooz Zareie[1], Louis H-Y. Lee ⓘ [1], Lutfi Huq[1], Susan N. Christo[1], Justine J. W. Seow[1], Keith A. Ching[9], Stéphane M. Guillaume ⓘ [1], Kathy Knezevic[5,6], Simone L. Park[1,18], Maximilien Evrard ⓘ [1], Jason Waithman ⓘ [10,11], Thomas Gebhardt ⓘ [1], Scott N. Mueller[1], Georgina E. Riddiough[12,13], Marcos V. Perini ⓘ [12,13], Simon C. H. Tsao[12,14], Terence P. Speed ⓘ [3,4], Peter K. Sorger[7], Sherene Loi ⓘ [5,6], Francis R. Carbone[1], Stephanie Gras ⓘ [15,16,17], Timothy S. Fisher ⓘ [9], Bas J. Baaten[9], Mark A. Dawson[5,6] & Laura K. Mackay ⓘ [1] ✉

[1]Department of Microbiology and Immunology, The University of Melbourne at the Peter Doherty Institute for Infection and Immunity, Melbourne, Victoria, Australia. [2]Department of Microbiology and Immunology, Computational Science Initiative, The University of Melbourne at the Peter Doherty Institute for Infection and Immunity, Melbourne, Victoria, Australia. [3]Bioinformatics Division, Walter and Eliza Hall Institute of Medical Research, Melbourne, Victoria, Australia. [4]School of Mathematics and Statistics, University of Melbourne, Melbourne, Victoria, Australia. [5]Peter MacCallum Cancer Centre, Melbourne, Victoria, Australia. [6]The Sir Peter MacCallum Department of Medical Oncology, University of Melbourne, Melbourne, Victoria, Australia. [7]Laboratory of Systems Pharmacology, Harvard Medical School, Boston, MA, USA. [8]Department of Thoracic Surgery and Institute of Thoracic Oncology, West China Hospital, Sichuan University, Chengdu, China. [9]Oncology Research Unit, Pfizer Inc, San Diego, CA, USA. [10]School of Biomedical Sciences, The University of Western Australia, Perth, Western Australia, Australia. [11]Telethon Kids Institute, The University of Western Australia, Perth, Western Australia, Australia. [12]Department of Surgery, The University of Melbourne, Heidelberg, Victoria, Australia. [13]HPB and Liver Transplant Unit, Austin Health, Heidelberg, Victoria, Australia. [14]School of Natural Sciences, Macquarie University, Sydney, New South Wales, Australia. [15]Infection and Immunity Program, La Trobe Institute for Molecular Science (LIMS), Bundoora, Victoria, Australia. [16]Department of Biochemistry and Chemistry, La Trobe University, Bundoora, Victoria, Australia. [17]Department of Biochemistry and Molecular Biology, Monash University, Clayton, Victoria, Australia. [18]Present address: Institute for Immunology and Immune Health, Perelman School of Medicine, University of Pennsylvania, Philadelphia, PA, USA. ✉e-mail: lkmackay@unimelb.edu.au

## Methods

### Mice

C57BL/6, gBT-I:CD45.1.2, gBT-I:CD45.1.1, gBT-I:CD90.1.2, P14:CD45.1.2, OT-I:CD45.1.2, OT-I:CD90.1.2; OT-3:CD45.1.2:TCRα$^{-/-}$ (ref. 59), OT-I:TGFβRII$^{fl/fl}$:Cre$^{dLck}$:CD45.1(TGFβRII$^{-/-}$) and TCRα$^{fl/fl}$Cre$^{ERT2}$ female mice were bred and maintained in the Department of Microbiology and Immunology, University of Melbourne under a 12 h/12 h light/dark cycle, at 19–22 °C and 40–70% humidity. All mice were maintained on standard chow. All experiments were approved by the University of Melbourne Animal Ethics Committee (ID nos. 21651 and 21938). All mice were between 6 and 14 weeks of age at the beginning of the experiments.

### Isolation of lymphocytes from human tumors and tissues

Following tumor excision, a representative tumor fragment was processed to generate single-cell suspensions as described previously[1]. Briefly, tumor or healthy tissues were diced finely in RPMI1640 containing 10% FCS and 0.5 mg ml$^{-1}$ collagenase D (Worthington Biochemical) and were incubated for 30 min at 37 °C. Digested pieces were mashed through 70-μm strainers and washed with RPMI1640 with 10% FCS. Lymphocytes from liver tumors and healthy liver tissues were enriched through density gradient centrifugation (500g, 20 min at 25 °C) on a 44%/70% isotonic Percoll gradient (GE Healthcare), diluted in HBSS, and cells at the solution interphase were isolated and washed with HBSS. Blood was diluted 1:1 in HBSS, overlaid on Ficoll-Paque PLUS (Sigma-Aldrich), centrifuged (400g, 15 min, 25 °C), and peripheral blood mononuclear cells were isolated from interphase. Cells were either used immediately for flow cytometry, or frozen in 10% dimethylsulfoxide:90% FCS freezing medium (breast, breast tumor, and blood samples) or Cryostor CS10 Freeze Medium (Stem Cell Technologies, cat. no. 07930; liver and liver tumor samples) for CITE-seq and restimulation assays.

### Mouse tumor and infection models

The TNBC cell line AT3-OVA was provided by P. Darcy (Peter MacCallum Cancer Center)[60,61]. AT3-OVA cells were cultured with complete DMEM (DMEM, 10% FCS, 2 mM L-glutamine, 100 U ml$^{-1}$ penicillin, 100 mg ml$^{-1}$ streptomycin). A total of 5 × 10$^5$ AT3-OVA cells in exponential growth phase (~70–80% confluency) were injected orthotopically into the fourth mammary fat pad in a total volume of 50 μl HBSS.

The melanoma B16F1-gB.GFP (B16-gB) cell line was provided by J. Waithman (University of Western Australia)[55]. B16-gB cells were cultured and passaged in complete RPMI (RPMI1640, 10% FCS, 2 mM L-glutamine, 100 U ml$^{-1}$ penicillin, 100 mg ml$^{-1}$ streptomycin, 50 mM 2-mercaptoethanol) at 37 °C and 5% CO$_2$. B16-gB was inoculated epicutaneously as described previously[55]. Tumors were measured every 2–3 days following the development of palpable tumors using vernier callipers, and tumor volume was calculated (length × width$^2$)/2. Mice were euthanized when tumors reached an ethical limit of 1.0 cm$^3$. gBT-I or OT-I T$_{EFF}$ cells were transferred into mice following the observation of tumor growth, between 14 and 20 days following tumor inoculation, and then collected at tumor endpoint (when tumors reached a maximum of 1,000 mm$^3$), which occurred 14–20 days following T cell transfer. AT3-OVA tumors were measured similarly and mice were euthanized before tumors reached the ethical endpoint volume of 1,000 mm$^3$.

LCMV infection was performed by intraperitoneal (i.p.) injection of 2 × 10$^5$ pfu of the Armstrong strain of LCMV.

### T cell transfer

For the adoptive transfer of naive T cells, 1–5 × 10$^4$ transgenic (P14 or OT-I) T cells were transferred i.v. to recipient mice 1–2 days before infection with LCMV or inoculation with AT3-OVA. For effector T cell transfer, transgenic (P14, OT-I, OT-3 or gBT-I) T cells were activated in culture for 5 days with gp$_{33-41}$ (KAVYNFATM–P14; Auspep), OVA$_{257-264}$ (SIINFEKL–OT-I/OT-3; Auspep), or gB$_{498-505}$ (SSIEFARL–gBT-I; Auspep)

peptide-pulsed splenocytes, in the presence of recombinant human IL-2 (25 U ml$^{-1}$; Peprotech) in complete RPMI (as above) at 37 °C and 5% CO$_2$. T cells were split 1:1 with fresh medium and IL-2 on days 2–4. T cells were resuspended in 200 μl HBSS for IV transfers. Unless stated otherwise, 1 × 10$^4$ effector OT-I, 5 × 10$^6$ gBT-I or 1 × 10$^6$ OT-3 were transferred 10 days post-AT3-OVA inoculation.

### In vivo treatments

For in vivo intravascular staining, 3 μg of αCD90.2-biotin was injected i.v. in 200 μl PBS, 3 min before euthanasia. To inhibit S1P-signaling pathways, mice were administered FTY720 (Cayman Chemical) diluted in 2% (2-hydroxypropyl)-β-cyclodextrin (Sigma-Aldrich) or vehicle daily via i.p. injection (1 μg g$^{-1}$) for the times indicated in the figure legend (Extended Data Fig. 7c). For tamoxifen treatment, mice were administered 2 mg of tamoxifen (or ethanol as vehicle control) diluted in sunflower seed oil (both from Sigma) i.p. daily for a total of five injections.

### Isolation of lymphocytes from mouse tissues

Lymphocytes from spleens and lymph nodes were isolated by grinding through 70-μm strainers. AT3-OVA tumors and MFPs (avoiding the inguinal lymph node) were collected into collagenase III solution (3 mg ml$^{-1}$; Worthington) containing DNase I (2.5 mg ml$^{-1}$; Sigma), chopped into fine pieces and incubated for 60 min at 37 °C. B16-gB tumors were collected into liberase TL research grade solution (0.25 mg ml$^{-1}$; Roche), chopped into fine pieces, and incubated for 60 min at 37 °C. Spleens and tumors were passed through a 70-mm strainer and erythrocytes were lysed in red cell lysis buffer (eBioscience) before staining for flow cytometry. Cells isolated from B16-gB tumors were cryopreserved in 10% dimethylsulfoxide:90% FCS for scRNA-seq.

### Flow cytometry and cell sorting

Single-cell suspensions were stained with fluorescently conjugated antibodies at 4 °C for 30–45 min in FACS buffer (1% BSA and 0.05 M EDTA in PBS). Dead cells were excluded by staining with Zombie Aqua or Near Infrared dyes (Biolegend). For transcription factor and cytokine staining, samples were fixed and permeabilized using the Foxp3/transcription-factor-staining buffer kit (eBioscience) according to the manufacturer's instructions and stained with fluorescent antibodies against intracellular proteins in permeabilization buffer (containing 2% rat and mouse serum (eBioscience)). All relevant information regarding the antibodies used in this study (clone, manufacturer, catalog number and dilution) is provided in Supplementary Table 1. Cells were enumerated using SPHERO calibration particles (BD Biosciences). Fluorescently labeled cells were acquired on a Cytek Aurora, unmixed with SpectroFlo software and analysis was performed using FlowJo (v.10.10.0; Treestar), or OMIQ for high-dimensional flow cytometry analysis. For cell sorting experiments, cells were sorted using a BD FACS Aria, using a 100-μm nozzle. For human cell sorting experiments for CITE-seq, CD3$^+$ Zombie NIR$^-$, cells were sorted into 50% FCS in RPMI before downstream processing. For B16-gB sorting experiments, DAPI$^-$ OT-I or gBT-I cells were sorted into 50% FCS in RPMI before downstream processing. For mouse AT3-OVA sort-transfer experiments, mice were treated with anti-ARTC2 nanobody i.v. (50 mg per mouse; S+16a, Biolegend) 10 min before organ collection, stained as above, and respective populations (see figure legends) sorted into 50% FCS in RPMI before washing with HBSS and transfer into recipient mice (8 × 10$^4$ cells in 200 μl i.v.).

### In vitro stimulation assays

To assess cytokine production capacity by T cells, cells were stained with surface stain antibodies (as above) and then incubated with phorbol myristate acetate (PMA; 50 ng ml$^{-1}$; Sigma-Aldrich), Ionomycin (1 mg ml$^{-1}$), Brefeldin A (10 mg ml$^{-1}$, Sigma-Aldrich), GolgiStop (1:1,500, BD) in complete RPMI (as above) for 4–5 h before intracellular staining

and flow cytometry. Unstimulated controls were included to confirm that stimulation did not alter cell-surface staining and to serve as negative controls for staining.

## Bulk RNA-seq analysis

Scatter plot of $\log_2$ fold changes (FCs) (for Fig. 1a) was produced using raw RNA-seq count data from GEO for Man et al.[26] (accession GSE84820) and Mackay et al.[2] (accession GSE70813). From the Man data, wild-type day 30 chronic and acute LCMV samples were selected for analysis, whereas, from the Mackay data, the following samples were selected for three separate analyses: (1) Skin $T_{RM}$, $T_{CM}$ and $T_{EM}$ HSV samples; (2) Gut $T_{RM}$, $T_{CM}$ and $T_{EM}$ LCMV samples; and (3) Liver $T_{RM}$, $T_{CM}$ and $T_{EM}$ LCMV samples. $T_{CM}$ and $T_{EM}$ samples were subsequently treated as a single '$T_{CIRC}$' group. For each analysis, genes were annotated with information from the National Center for Biotechnology Information and those with obsolete symbols or annotated as rRNA were removed, as were genes that failed to achieve a count above ten in all samples in at least one group. Each dataset was further processed by applying the imputation strategy published previously[47,62]. Counts-per-million values were calculated using the edgeR package[63], together with scaling factors derived from the TMM method[64], $\log_2$ transformed with a previous count of 1, followed by application of the normalization method RUV-III[65] with biological replicates nominated as replicates, mouse housekeeping genes[56] nominated as 'negative control' genes, and $k = 1$ factors of unwanted variation. Normalization success was assessed with relative log expression plots[66], PCA plots[67], and histograms of $P$ values. The limma package[68] was used to fit gene-wise linear models for the given group structure with the output from RUV-III as an additional model covariate. The $\log_2$FC estimates from the Mackay Skin, Gut, and Liver analyses were averaged to create a 'pooled $T_{RM}$' expression profile and then plotted against the estimates from the Man analysis. A $P$ value was calculated by constructing a two-way contingency table, counting the number of genes with concordant/discordant $\log_2$FCs, and applying Fisher's exact test for association. Core $T_{RM}$ gene signature was used as described[69].

## CITE-seq and scRNA-seq library preparation

Following sorting of cell populations, sorted cells were stained with TotalSeq-C Universal Cocktail v.1.0 (Biolegend; Human CITE-seq experiments) or TotalSeq-C Hashtag antibodies (Biolegend; mouse B16-gB scRNA-seq experiment) according to the manufacturer's instructions. Cells were filtered through a 40-mm Flowmi cell strainer, loaded onto a 10x Genomics chromium controller, and prepared for sequencing using a Chromium Next GEM Single Cell 5′ kit with feature barcoding and immune receptor mapping (v.2, Dual Index) and VDJ enrichment kits for mouse or human TCR from 10x Genomics. Libraries were generated according to the manufacturer's instructions. Libraries were profiled on an Agilent Tapestation and quantified using a Qubit before sequencing on an Illumina NextSeq2000 P2 or P3 kit.

## CITE-seq and scRNA-seq analysis

Sequencing reads of three separate lanes were aligned to the hg38 reference genome and TCR reference (VDJ) and counted with cellranger v.6.1.2. CITE-seq antibodies were aligned to their custom reference sequences from BioLegend. Patient samples were demultiplexed into genotypic donors using vireo[70] on aligned BAM files from three lanes. Donors were matched to genotypes using known cell frequencies in each sample. Three batches of single-cell data were merged and processed using Seurat[71] (v.4.3.0). Specifically, cells were filtered if they contained fewer than 500 genes, more than 5% mitochondrial RNA and were annotated to have more than one beta chain in the VDJ assay or two genotypes in the vireo analysis (cell doublets). Then, for the RNA assay, we used NormalizeData to normalize the counts data and determine the top 2,000 variable genes using FindVariableFeatures. We excluded TCR components (^TRA/^TRB), mitochondrial genes (^MT),

and ^HLA from the variable genes as unwanted factors of variance and performed PCA (RunPCA). The CITE-seq assay was processed similarly with centered log ratio normalization and margin = 2. We removed the individual effects between the donors using Harmony[72] on the RNA-seq and CITE-seq assays individually and then combining the correction reductions using the FindMultiModalNeighbors functions (using 25 dimensions from the RNA reduction and 18 from CITE-seq). Uniform manifold approximation and projection (UMAP) reductions and cell neighbors were calculated using 25 dimensions from the weighted nearest-neighbor reduction. Clusters were detected with a resolution of 3. We then determined T cell subsets within the data by assessing cluster markers (FindAllMarkers) and their overall expression and comparing them to published markers in the literature (merging clusters as need be). We subset the data to CD8$^+$ T cells only at this point by excluding other T cell populations and contaminating immune populations. All further analyses and plots were generated in R (v.4.2) using tidyverse[73] functions and ShinyCell[74]. Heatmaps were created with the pheatmap package (v.1.0.12). Subset signature expression levels were calculated using the AddModuleScore function. Pseudobulk analysis was performed using the edgeR package[63]: counts of reads were summed per subset. Pseudobulk samples from the periphery of the data (gdT, MAIT) are filtered from the analysis and lowly expressed genes removed using filterByExpr(). The samples were then normalized using TMM and the calcNormFactors() function. Dimensional reduction was performed using the plotMDS function. Other single-cell datasets, such as Giles 2022[27], or subsets of the data (tumor only) were analyzed analogously to the methods above (leaving out CITE-seq and TCR where appropriate).

**TCR analysis.** TCRs from cellranger outputs were paired based on cell barcodes and merged with gene expression data. In cases where several contigs were detected the contigs with the highest unique molecular identifier was kept. TCR clonotype was defined by the joined alpha and beta CDR3 nucleotide sequences, and expanded TCR clonotypes were determined by filtering the list of clonotypes to those that are found in two or more cells. Antigen specificity of cells was based on TCR clonotype and donor HLA identity. HLA type was determined with ArcasHLA using the '−single' flag[75]. Reads covering the genome coordinates chr.6:28510120−33480577 were extracted into a separate fastq file per donor and processed individually. Plots of clonotype diversity and similarity within/across cell types were calculated using djvdj (https://rnabioco.github.io/djvdj/ v.0.1.0) and plotted with Complex-Heatmap[76]. For clonotype sharing plots, expanded clonotypes within each tissue were filtered to include only those that are shared across more than one subset. The number of shared clonotypes was quantified for each subset pairing, and the total number of cells contributing to clonotype sharing from each subset was counted. Visualizations were generated with custom R scripts using the ggraph package (v.2.0.5). For iterations of the sharing plots with filtering, input data were refined so that a shared TCR was removed from the plot if one of the cell types in the pairing contributed just one cell (for $n > 1$ filtering) or two or fewer cells (for $n > 2$ filtering). Other visualizations of clonotype cell fate and cluster sharing were generated using custom R scripts, as provided. Comparisons between TCR sequences was performed using TCRdist3 (refs. 39,40) using default weights with the distance matrix calculated as the sum pairwise distance for the alpha and beta chains. To determine potential virus-recognizing clones, published TCRs with known viral HLA-peptide specificities were obtained from VDJdb[41–43], following which edit distances were calculated for each clone against reference CDR3a/b peptide sequences. The lowest edit distance was obtained for each clone, with distances less than 1 considered viral-associated.

**Pancancer atlas processing.** Preprocessed Seurat objects were obtained[28]. SCTransform was used to normalize, scale and regress prefiltered dataset. Precomputed scores for dissociation induced genes (DIG), Malat1, mitochondrial percentage and cell-cycle (G2M

and S scores) were used for regression. T cell receptor variable/joining (^TR[A|B|G|D][V|J]), B cell receptor variable/joining (^IG[H|L|K][V|J|D]) DIG(^HSP/^DNAR), cell-cycle, DIG(^HSP/^DNAJ) and ribosomal (^RP([0-9]$^+$-|[LS])) genes were removed from VariableFeatures gene list before calculating PCs. Subsequent UMAP and FindNeighbors commands performed with 15 PCs. $T_{RM}$ and $T_{EX}$ scores were calculated with the AddModuleScore command using respective genesets. $T_{EM}$ and $T_{EMRA}$ scores were calculated using differentially expressed genes calculated from the BC dataset. Other external datasets were analyzed similarly.

## Signature acquisition and module scoring

$T_{RM}$ and $T_{EX}$ expression signatures were derived by summing raw counts, gene-wise, over each donor within each cluster, to create 'pseudobulk' samples for each donor/cluster combination. Samples composed of fewer than 20 cells were removed, and samples from the same meta-cluster were subsequently treated as belonging to the same group. Genes annotated as being of 'HLA', 'TRB', 'TRA', 'TRG' or 'TRD' type were removed, as were genes that failed to achieve a count above 5 in at least two samples in at least one group (breast) or a count above 3 in at least four samples in at least one group (liver). Counts were $\log_2$ transformed with a previous count of 1/2 (breast) or 1 (liver), followed by application of the normalization method RUV-III with samples from the same group nominated as replicates, human housekeeping genes nominated as 'negative control' genes, and $k = 20$ (breast) or $k = 15$ (liver) factors of unwanted variation. Normalization success was assessed as above. The edgeR package [*] was used fit gene-wise negative binomial generalized linear models for the given group structure with the output from RUV-III as additional model covariates, with a previous count of 1/2 (breast) or 1 (liver). Likelihood ratio tests were employed to test for DE, where a gene was judged to be DE if the Benjamini and Hochberg[77] adjusted $P$ value < 0.05. For breast, the $T_{RM}$ signature was defined by a contrast between the $T_{RM}$ group and the average of all other groups; the $T_{EX}$ signature was defined similarly. For liver, the $T_{RM}$ signature was defined by a contrast between the CD103$^+$ $T_{RM}$ group and the average of all other groups except the CD103$^-$ $T_{RM}$ group; the $T_{EX}$ signature was defined similarly. The breast $T_{RM}$ 'union' signature was derived using the same steps for breast above, except that samples obtained from the $T_{RM}$ or $T_{EX}$ clusters were combined into a single group. Signature scores were calculated from the SCTransformed assay using the 'AddModuleScore' function from the Seurat package[71] for up and down genes separately and combined by averaging the scores for the up genes and the sign reversed scores for the down genes. These scores were overlaid onto UMAP plots using the FeaturePlot function from Seurat. To generate the pancancer $T_{RM}$ signature we focused on the leading-edge genes contributing to the enrichment in the barcode plots (Fig. 3c and Extended Data Fig. 4g). Specifically, the leading-edge genes contributing to $T_{RM}$ signature associations between the BC and liver datasets were refined by intersecting the genes in the breast and liver $T_{RM}$ signatures (described above) and then subsetting to genes which are either (1) concordantly DE in both breast and liver or (2) DE in one with a concordant absolute $\log_2$FC > 0.5 in the other. The pancancer $T_{EX}$ signature was refined similarly (Extended Data Fig. 4g). Barcode enrichment plots were generated using limma, and gene set enrichment $P$ values were calculated using the camera function on the $\log_2$FC values for populations of interest against the background calculated with FindMarkers. All signatures are available in Supplementary Table 2.

## Survival analysis

Clinical information and normalized microarray data (for Fig. 2l and Extended Data Fig. 3f–h) for the study[58,78] were downloaded from the cBioPortal (https://www.cbioportal.org). Signature scores for each patient were calculated using the 'sig.score' function from the genefu package[79]. Kaplan–Meier curves were calculated using the R package survival and were plotted, with the log-rank test $P$ value, using the R package survminer.

Clinical information and microarray data (for Fig. 2m) for the iSPY study[31] were downloaded from supplementary material and GEO (accession GSE194040), respectively. Probes targeting the same gene were averaged. If a gene had missing values, these were imputed using the average of all nonmissing values across the gene. For each platform batch, relative log expression[66] values were computed, the median value for each sample was calculated, and then samples were ranked by this median value: the bottom, middle and top third rankings defined three separate 'pseudo-batches' of samples. For each PAM50 BC expression subtype in each pseudobatch, one of the following was performed: if there were between five and ten samples, five samples were selected randomly and averaged, gene-wise, to create one 'pseudo-sample'; or if there were more than ten samples, the bottom five ranked samples and the top five ranked samples were averaged, gene-wise, to create two pseudosamples. These pseudosamples were used as pseudoreplicates in the RUV-III-PRPS normalization methodology[80], with all genes nominated as 'negative control' genes, and $k = 7$ factors of unwanted variation. Normalization success was assessed with relative log expression and PCA plots[67]. Based on this normalized data, signature scores for each patient were calculated using the 'sig.score' function from the genefu package. Receiver operating characteristic (ROC) curves and $P$ values were calculated using the R packages pROC[81] and verification, respectively.

## CycIF of human breast tumors

FFPE-embedded tumor tissues from seven patients (six female, one male) were purchased from the Cooperative Human Tissue Network based on histologic criteria (Supplementary Table 3), for invasive carcinoma (ductal/lobular) and 5-μm thick sections were cut on Superfrost Plus histology slides (Fisher scientific) at the BWH Histopathology core, as described previously[29].

FFPE tissues were deparaffinized, rehydrated, subjected to antigen retrieval on a Leica BondRx and characterized by CycIF imaging, as described previously[29]. Briefly, tissues were stained overnight at 4 °C with primary antibodies or for 1 h at room temperature for secondary antibody conjugates in a dark, humidified chamber with antibodies diluted in Superblock (Thermo-Fisher) supplemented with 1 mg ml$^{-1}$ Hoechst 33258 (Bio-Rad Laboratory) rinsed for 30 min in PBS (room temperature) and mounted with 50% glycerol in PBS using a 24 × 60-mm coverslip. All samples were stained and imaged in together to reduce batch effects using the antibody panel (Supplementary Table 1). Stained tissues were imaged using a Cytefinder II HT (Rarecyte) automated slide scanning fluorescence microscope using a ×20 (0.6 numerical aperture) objective. After imaging, mounted slides were soaked in PBS (room temperature) to detach coverslips, and then immersed in PBS supplemented with 4.5% hydrogen peroxide and 24 mM sodium hydroxide and exposed to LED light for 1 h. Slides were then rinsed twice in PBS in preparation for the next staining cycle. Image processing, assembly, segmentation and single-cell quantification were performed using MCMICRO[30].

CycIF Images were processed with Cecelia[82]. Individual channels were denoised and segmented with Cellpose[83] using a radius of 10 μm and 8 μm, respectively. T cell (CD3e) and cancer cell (panCK) segmentation was based on their respective markers in combination with the DNA (Hoechst-stained nuclei) channel from the third imaging cycle, the same cycle as CD3e. Cell populations were gated using Cecelia's histocytometry module and spatial interactions analyzed. Cell distances were extracted using K-nearest-neighbor from the DBSCAN library[84]. Clusters within the CD8$^+$CD103$^+$KLF2$^-$ T cells were defined using the Seurat package. We first removed the individual donor effects and performed PCA using a set of 22 features (CD39, CD45RO, CD3E, CD25, CD73, CD8A, GZMB, CD103, LAG3, PD-1, TIM3, CD69, TCF1, FOXP3, GNLY, CD4, pJUN, pERK, KLF2, CD7, CD94, NKG2A) (RunPCA). Shared neighbor graphs (FindNeighbors) and UMAP reduction (RunUMAP) were performed using the first ten dimensions. Clusters were detected using a resolution of 0.8 (FindClusters).

## CRISPR–Cas9 editing of CD8⁺ T cells

CRISPR–Cas9 editing of naive CD8⁺ T cells was performed as described previously[85]. Single-guide RNAs (sgRNA) targeting: *Tcf7* (5′-UCUGCUCAUGCCCUACCCAC-3′, 5′-AGCUGGGGGACGC CAUGUGG-3′, 5′-UGUGCACUCAGCAAUGACCU-3′), *Cxcr3* (5′-GAA CAUCGGCUACAGCCAGG-3′, 5′-UGAGGGCUACACGUACCCGG-3′), *Cxcr6* (5′-UCUGUACGAUGGGCACUACG-3′, 5′-UGUGCCAAAGA CCCACUCAU-3′) and *Cd19* (5′-AAUGUCUCAGACCAUAUGGG-3′, 5′-GAGAAGCUGGCUUGGUAUCG-3′) were purchased from Synthego (CRISPRevolution sgRNA EZ Kit). sgRNA/Cas9 RNPs were formed by incubating 0.3 nmol of each sgRNA with 0.6 µl Alt-R S.p. Cas9 nuclease V3 (10 mg ml⁻¹; Integrated DNA Technologies) for 10 min at room temperature. Naive CD8⁺ T cells were negatively enriched from spleen and lymph nodes of gBT-I or OT-I mice by incubating cell suspensions with anti-CD4, anti-CD11b anti-F4/80, anti-Ter119 and anti-I-A/I-E monoclonal antibodies, followed by incubation with goat anti-rat IgG-coupled magnetic beads (Qiagen) before removing bead-bound cells. A total of 1 × 10⁷ enriched T cells were resuspended in 20 µl of P3 (P3 Primary Cell 4D-Nucleofector X Kit; Lonza), mixed with sgRNA/Cas9 RNP and electroporated using a Lonza 4D-Nucleofector system (DN100). Cells were rested for 30 min in a 96-well plate before direct transfer into recipient mice (cotransfer of 1:1 ratio of sg*Cd19*:sg*Tcf7* CRISPR-edited naive gBT-I cells, total of 2 × 10⁴ cells per mouse: Extended Data Fig. 8a–d) or activation in culture for 5 days with peptide-pulsed splenocytes (gB₄₉₈₋₅₀₅ (SSIEFARL)) in the presence of IL-2 (25 U ml⁻¹, Peprotech) at 37 °C, 5% CO₂, before transfer into recipient mice (cotransfer of 1:1 ratio of sg*Cd19*:sg*Cxcr3* or sg*Cd19*:sg*Cxcr6* CRISPR-edited naive gBT-I cells, total of 1 × 10⁶ cells per mouse; Extended Data Fig. 8m,n).

## SPLINTR methods

To construct the SPLINTR barcoding systems, violet-light excited GFP (VEX) was used to replace NGFR in MSCV-IRES-NGFR[86] using the *Nco*I and *Bam*H1 restriction sites, or eGFP into the SPLINTR lentivirus vector[46]. A semi-random oligonucleotide library synthesized by Integrated DNA Technologies with the following pattern (NNSWSNNWSW)₆ was amplified by eight cycles of PCR. The barcode library was cloned into the 3′ UTR of VEX using *Bam*H1 and *Mfe*I restriction sites at a 50:1 insert:vector ratio. Ten side-by-side ligation reactions were pooled and purified using two Monarch PCR and DNA Clean up columns (NEB) in a volume of 6 µl per column. The ligation reactions were pooled and split across two 25-µl aliquots of Endura Electrocompetent cells (Lucigen). Cells were recovered for 1 h postelectroporation, pooled and grown in 500 ml of LB supplemented with ampicillin (100 µg ml⁻¹) overnight at 37 °C. The plasmid library was extracted using NucleoBond Xtra kit (Macherey–Nagel). SPLINTR libraries were sequenced to a depth of 100 million paired-end reads per technical duplicate and reference libraries used for downstream analysis were generated as described previously[46]. SPLINTR retrovirus and lentivirus VEX library represents a highly diverse barcode library containing 3.2 × 10⁶ unique barcodes or 1.3 × 10⁶ barcodes, respectively.

Naive (T_N) OT-I T cells were barcoded via transduction of hematopoietic stem and progenitor cells (HSPC) and intrathymic transfer into sublethally irradiated chimeric mice. Briefly, bone marrow was isolated from OT-I mice, and lineage(lin)-positive cells were depleted using the EasySep Mouse Hematopoietic Progenitor Cell Isolation Kit (Stem Cell Technologies). Lin-negative HSPCs were plated in fibronectin-coated 12-well plates containing polyvinyl alcohol (PVA)-based medium: 1× Ham's F-12 Nutrient Mix liquid media (Gibco) supplemented with 10 mM HEPES, 1× P/S/G, 1 ITSX, 1 mg ml⁻¹ PVA along with TPO (100 ng µl⁻¹) and mouse SCF (10 ng µl⁻¹). For barcoding, 20 pools of 1 × 10⁶ cells (cultured for 6 days) were transferred to 24-well fibronectin-coated plates in 250 µl of PVA medium. Viral barcoding vectors were titrated to achieve 10% reporter expression (0.1 multiplicity of infection) to minimize multiple integrations. Cells were transduced via spinfection at 2,000*g* for 2 h (no break). After transduction,

medium was refreshed and cells were cultured for a further 48 h before VEX-positive and Lin- cells were sorted (ARIA Fusion, BD). The 20 pools of 3.5–5.5 × 10⁵ cells were plated individually into 48-well plates and expanded for 1-week. Recipient C57Bl/6 mice were then irradiated 4 Gy and HSPCs were transplanted intrathymically such that each mouse received an independent barcode pool. Eight weeks later, chimeric mice were bled, equal numbers of VEX⁺ OT-I T cells from each mouse were pooled, and cells were transferred into C57Bl/6 mice that were then inoculated with AT3-OVA tumors.

Effector (T_EFF) OT-I T cells were barcoded as follows. Naive OT-I T cells were isolated from spleens and LNs and enriched via negative selection as for CRISPR–Cas9 experiments. Enriched OT-I T cells were activated with anti-CD3 (2 µg ml⁻¹, 145-2C11, BioXCell) and anti-CD28 (1 µg ml⁻¹, 37.51, BioXCell) for 24 h before 'spinfection' with a pretitrated volume of SPLINTR retrovirus to ensure <5% (0.05 multiplicity of infection) transduction efficiency to limit several barcode integrations in a single-cell, in 24-well plates, precoated with Retronectin (32 µg ml⁻¹, Takara). Transduced cells were sorted 24 h later and transferred into mice bearing AT3-OVA tumors. At the endpoint, transduced OT-I T cell populations were sorted from spleens and tumors and lysed in Viagen lysis buffer with 0.5 µg ml⁻¹ proteinase K (Invitrogen) for DNA barcode sequencing.

Barcode sequences were amplified from genomic DNA using primers flanking the constant region of the barcode before adding i5 and i7 indexes compatible with next generation sequencing[46]. Libraries were sequenced on an Illumina NextSeq2000 using 100 bp single-end chemistry targeting 2 million reads per sample. The BARtab pipeline[87] (https://github.com/DaneVass/BARtab) was used to map the sequencing reads to a barcode reference library, perform quality control analysis, and generate a barcode counts table. The bartools R package[87] was used to collapse PCR replicates and generate barcode abundance bubble plots and correlation heatmaps for data visualization. Scatterplots comparing barcode abundance between two samples were generated using the R package ggplot2.

## Ethics approval

All animal experiments were approved by The University of Melbourne Animal Ethics Committee (ID nos. 21651 and 21938). All study on human specimens was approved by the Human Research Ethics Committee of the University of Melbourne (ID nos. 13009 and 14517). All participating patients provided written informed consent.

## Randomization and statistics

Test animals were assigned randomly to experimental groups in all experiments. No further randomization of data collection was performed. Data collection and analysis were not performed blinded to the conditions of the experiments. No statistical methods were used to predetermine sample sizes, but our sample sizes are similar to those reported in previous publications[25,47,62]. In all cases, data distribution was assumed to be normal, but this was not tested formally. No datapoints were excluded from the statistical analyses.

## Reporting summary

Further information on research design is available in the Nature Portfolio Reporting Summary linked to this article.

## Data availability

The single-cell CITE-seq and scRNA-seq data generated for this study have been deposited in the GEO under accession codes GSE267552, GSE309625 and GSE309444. Source data are provided with this paper.

## Code availability

The code generated and used for analysis of single-cell CITE and RNA-seq data is provided via Github at https://github.com/CSI-Doherty/Burn2025.

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

## Acknowledgements

We thank the Flow Cytometry Unit and Bioresources Facility at the Doherty Institute (University of Melbourne) for technical assistance; the NIH Tetramer Core Facility (NIH Contract 75N93020D00005 and RRID:SCR_026557) for providing SIINFEKL monomer; the Australian Donor Tissue Biobank for healthy tissue samples; Pfizer Inc. Cancer Immunology team members, the laboratories of L.K.M., W. Heath and Y. Alexandre (University of Melbourne) for comments and discussion. This work was supported by Pfizer Inc., Cancer Council Victoria Grants-in-Aid, and National Health and Medical Research Council (NHMRC) to L.K.M. Z.M. is supported by NCI grant R50-CA252138. Histopathology was supported by NIH core grant P30-CA06516.

## Author contributions

Conceptualization: T.N.B. and L.K.M. Methodology: T.N.B., J.S., L.C.G., M.O., E.N.W., E.Y.N.L., Z.M., M.L.M., S.N.C., S.L.P., M.E., T.P.S., P.K.S., S.G. and L.K.M. Investigation: T.N.B., M.O., E.N.W., K.M.M., R.B.E., A.Z., Z.M., P.Z., L.H.-Y.L., L.H. and S.M.G. Formal analysis: T.N.B., J.S., L.C.G., E.Y.N.L., S. Li, L.D., D.R., A.Z., D.S., P.M., H.J.L., H.L., J.J.W.S., S.N.M., K.A.C. and S.G. Resources: T.N.B., E.N.W., Z.M., K.K., J.W., T.G., G.E.R., M.V.P., S.C.H.T., P.K.S., F.R.C., S. Loi, T.S.F., B.J.B. and M.A.D. Writing—original draft: T.N.B. and L.K.M. Writing—review and editing: all authors Supervision: L.K.M. Funding acquisition: L.K.M.

## Competing interests

The authors declare no competing interests.

## Additional information

**Extended data** is available for this paper at https://doi.org/10.1038/s41590-025-02347-9.

**Correspondence and requests for materials** should be addressed to Laura K. Mackay.

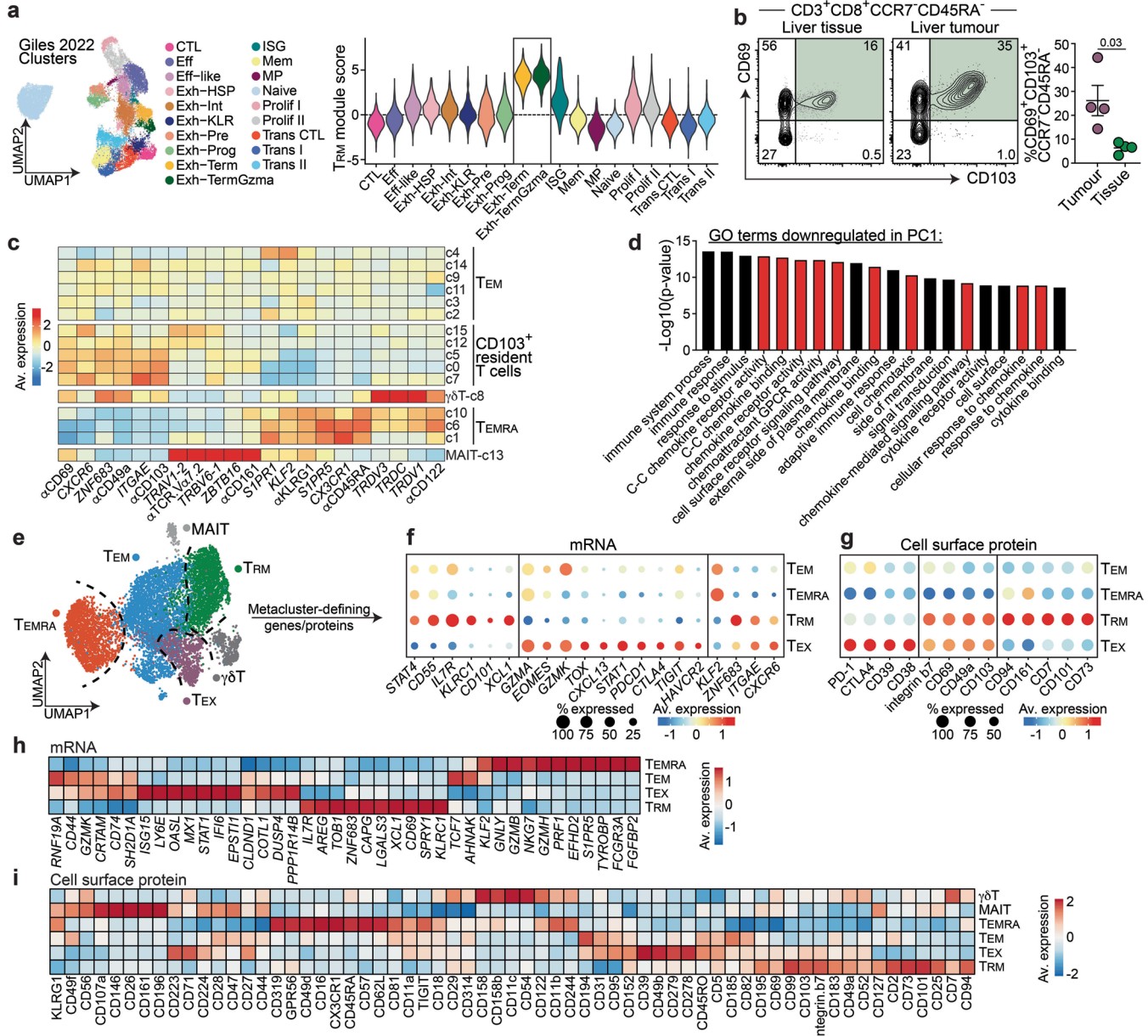

**Extended Data Fig. 1 | Identification of CD8+ T cell populations in tumours.**
**a**, $T_{RM}$ gene signature expression on LCMV-specific CD8$^+$ T cell clusters from scRNA-seq of the respective dataset[27] (Related to Fig. 1b–d). **b**, Flow cytometry of CD8$^+$ T cells isolated from liver tumours and liver tissue from N=4 colorectal cancer patients with liver metastases. Representative plots and summary data for CD69$^+$CD103$^+$CCR7$^-$CD45RA$^-$ CD8$^+$ T cells, analysed by two-sided Mann-Whitney test. Error bars indicate mean +/- SEM. **c**, Expression of respective lineage-defining genes and cell surface proteins from BC-CITEseq data

(Fig. 1i, j) displayed on heatmap by cluster. **d**, pathway enrichment of downregulated loading genes associated with PC1 from PCA plot in Fig. 1n. Red bars indicate pathways related to chemokine receptor signaling and trafficking **e**, Schematic showing metacluster annotations. **f-g**, Expression of selected shared and distinct $T_{RM}$ and $T_{EX}$ genes (**f**) or cell surface proteins (**g**) across metaclusters from BC dataset. **h-i**, heatmaps showing top DE genes (**h**) and cell surface proteins (**i**) from BC dataset.

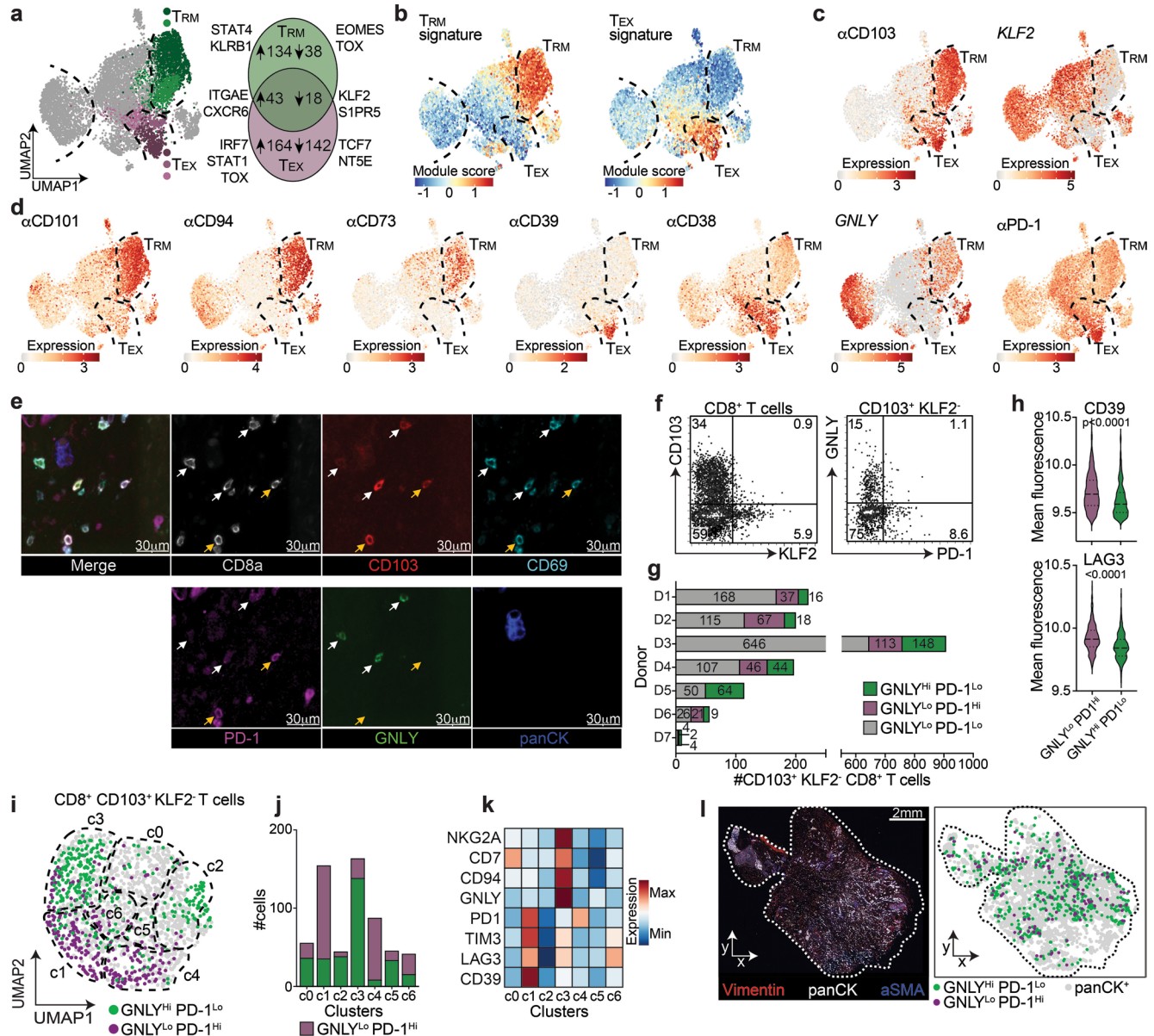

**Extended Data Fig. 2 | Lineage-defining genes and proteins enable in situ localization of $T_{RM}$ and $T_{EX}$. a**, $T_{RM}$ and $T_{EX}$ signature genes. Shared genes indicate genes are up (or down)-regulated in the $T_{RM}$ vs all *and* $T_{EX}$ vs. all comparisons (Fig. 2a). $T_{RM}$ (**b**) and $T_{EX}$ (**c**) gene signature module scores displayed on BC UMAPs for summary module score data (Fig. 2b, c). **c-d**, Expression of respective genes and cell surface proteins displayed on BC UMAPs. **e-l**, CyclF imaging of BC tumours related to Fig. 2f–h. **e**, representative image of BC tumour using CyclF approach showing respective stains. Representative of 7 independent donors. **f**, Histocytometric gating strategy showing CD8⁺ T cells (CD3⁺ CD8⁺ CD4⁻) concatenated from CyclF images from 7 donors showing CD103, KLF2, GNLY, and PD-1 expression on respective populations. **g**, number of CD8⁺ CD103⁺ KLF2⁻ T cells identified in each slide from respective donors, broken down by GNLY and PD-1 expression based on gates in (f). **h**, Expression of CD39 and LAG3 proteins on respective populations, pooled from 7 donors. Two-sided Mann-Whitney test. **i**, UMAP projection and clustering of CD8⁺ CD103⁺ KLF2⁻ T cells, coloured by gates defined in (f). **j**, number of cells from respective gates identified within each cluster. **k**, relative expression of respective proteins in each cluster. **l**, representative image of tumour section showing panCK and aSMA expression, and relative location of respective annotated cell types.

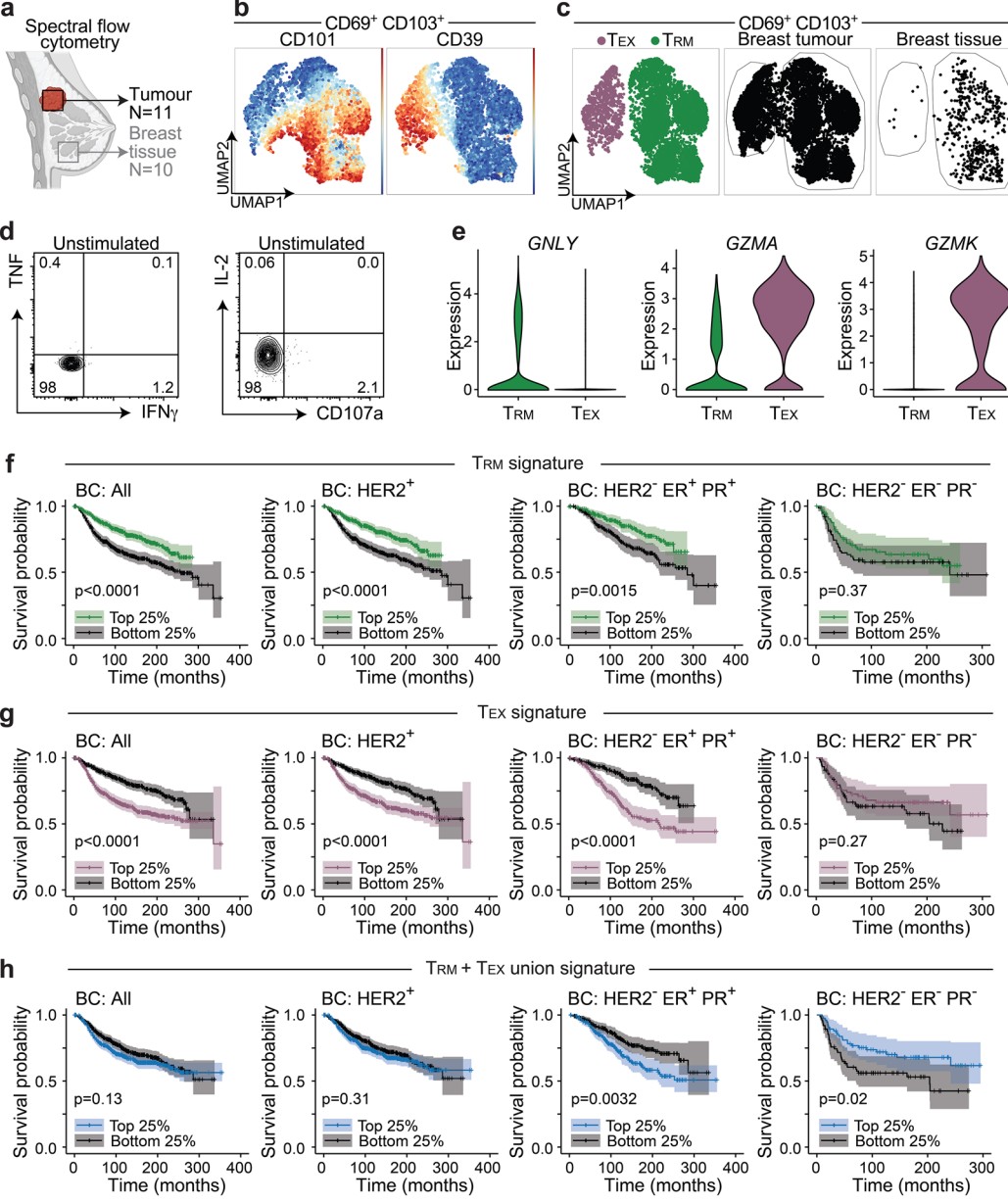

**Extended Data Fig. 3 | Functional assessment and survival associations of BC T$_{RM}$ and T$_{EX}$. a-e**, Flow cytometry of PMA-ionomycin restimulated BC and breast tissue-derived CD103$^+$ resident-phenotype T cells related to Fig. 2d–k. **a**, Schematic. **b-c**, CD69$^+$CD103$^+$CD45RA$^-$CCR7$^-$CD8$^+$ T cells represented in UMAP space based on the expression of CD69, CD101, CD39, CD94, CD73, CD103, CD45RA, CCR7, CD38 showing CD101 and CD39 expression (**b**) and annotated T$_{RM}$ and T$_{EX}$ populations and relative contribution by cells derived from BC tumours or tissue (**c**). **d**, Gating of TNF, IFNγ, IL-2, and CD107a for Fig. 2i–k based on unstimulated control. **e**, Expression of granulysin (*GNLY*), granzyme A (*GZMA*) and granzyme K (*GZMK*) on T$_{RM}$ and T$_{EX}$ populations from BC CITEseq dataset. **f-h**, Survival of BC patients from the dataset with the highest (top 25%) T$_{RM}$ (**f**), T$_{EX}$ (**g**), or combined T$_{RM}$ + T$_{EX}$ union (**h**) gene signature score enrichments compared to patients with lowest (bottom 25%) gene enrichment scores and plotted on Kaplan-Meier curves with log-rank test, as in (Fig. 2l), segregated by BC subtypes (as annotated).

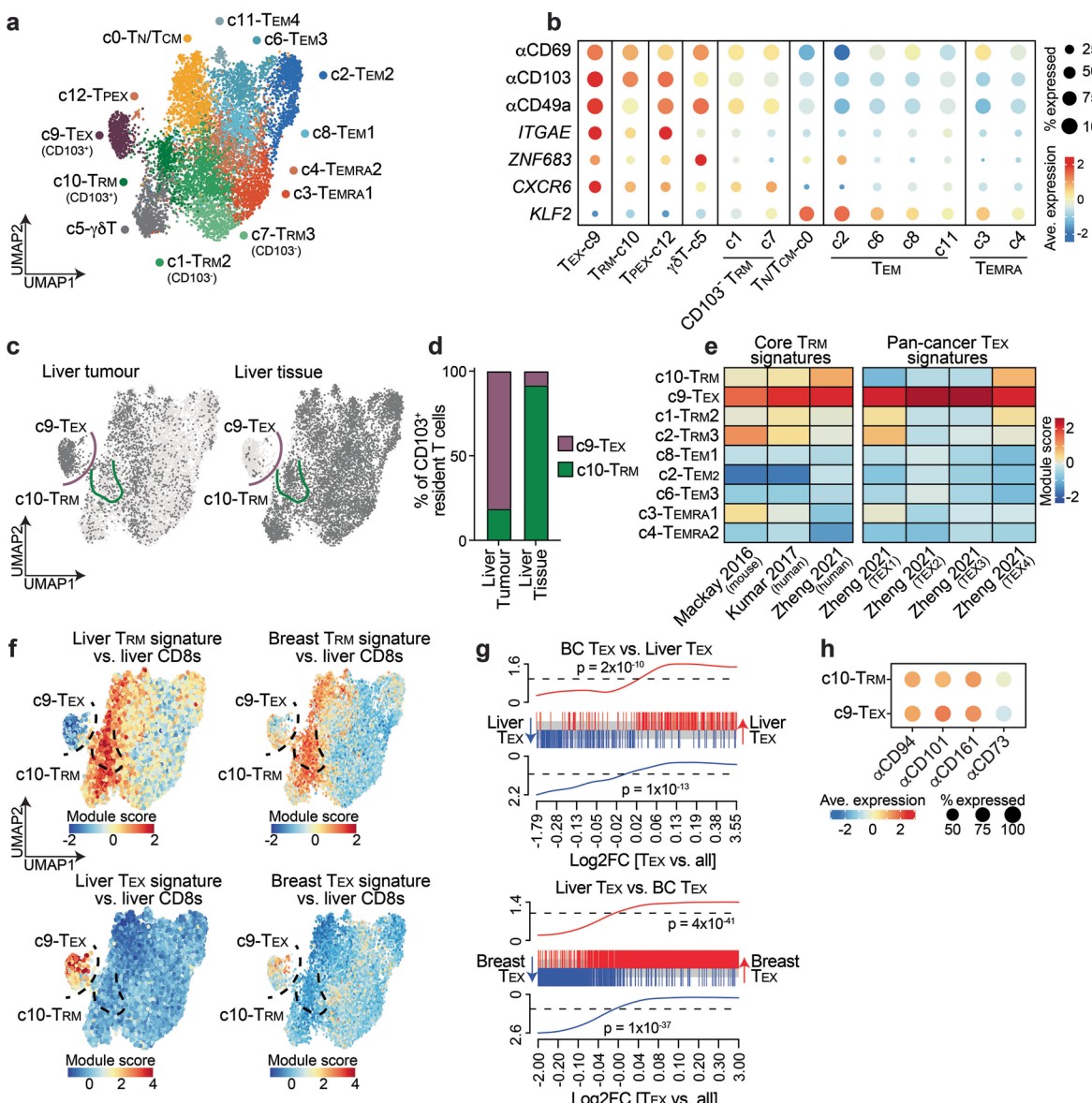

**Extended Data Fig. 4 | Deconvolution of CD103+ $T_{RM}$ and $T_{EX}$ cells in liver metastases. a-d**, CITEseq of CD3+CD8+CD4− non-MAIT cells isolated from secondary liver tumours (colorectal cancer patients, N=4) and non-cancerous liver tissue (N=6) as depicted in Fig. 3a. **a**, Data was Harmony integrated, and unified protein and RNA-seq data represented on weighted nearest neighbours UMAP, coloured by clusters that were annotated based on expression of lineage-defining cell surface proteins and genes. **b**, Expression of selected cell surface proteins (αCD69, αCD103,) and genes (*ITGAE, ZNF683, CXCR6, KLF2*)

on respective clusters. **c-d**, CD8+ T cells segregated by tissue of origin (**c**), and relative cluster composition of CD103+ resident T cells (c9+c10) isolated from BC tumours or tissue (**d**). **e**, Average module scores of published $T_{RM}$[2,21,28] and $T_{EX}$[28] cell gene signatures by annotated clusters. **f**, Module score overlays of relevant signatures on liver CITEseq dataset. **g**, gene-set enrichment analysis of BC $T_{EX}$ vs liver $T_{EX}$ signatures and vice versa. Barcode plots show GSEA enrichment using a running-sum statistic, p-val determined by permutation testing. **h**, Expression of respective cell surface proteins across annotated clusters.

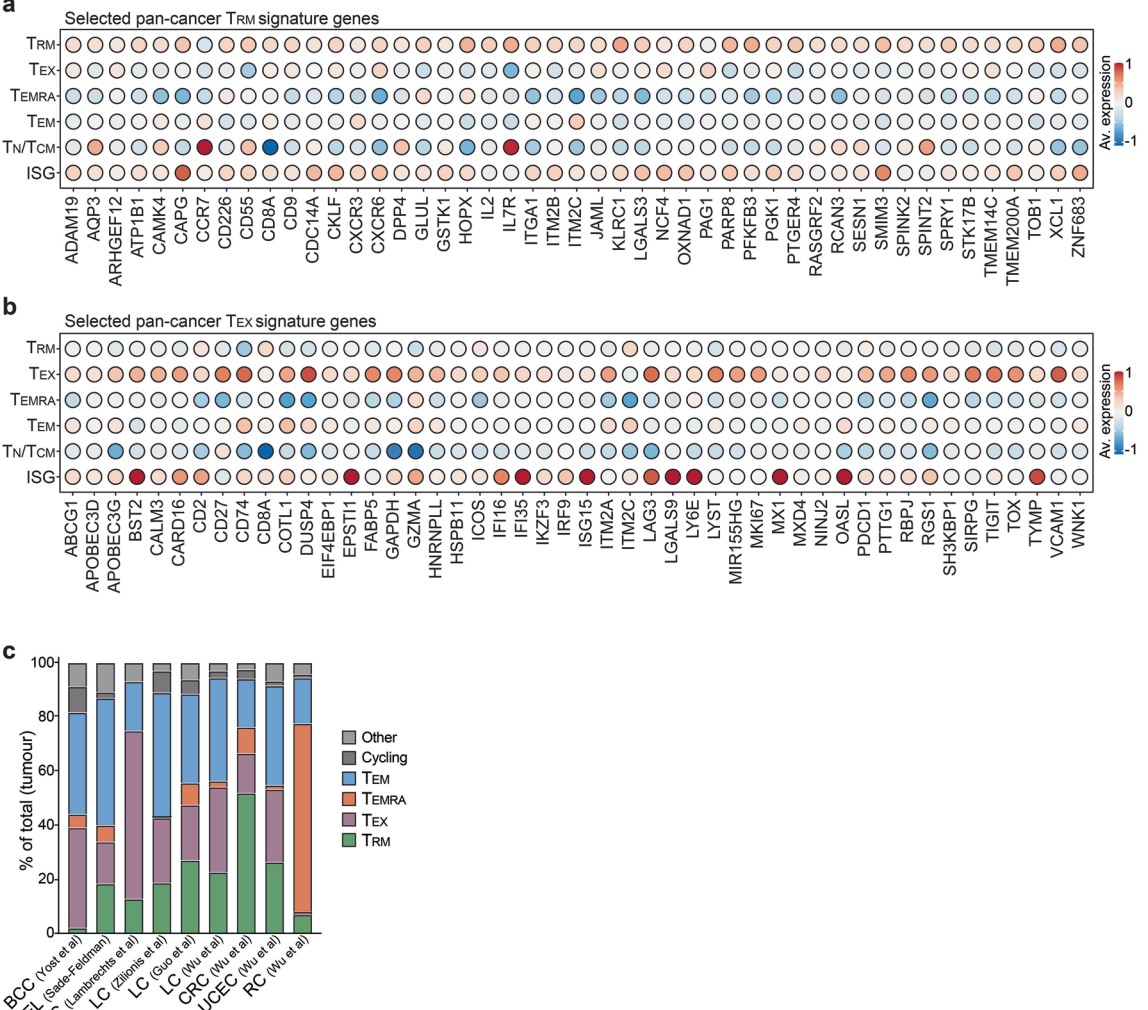

**Extended Data Fig. 5 | Pan-cancer T$_{RM}$ and T$_{EX}$ signature genes. a-b**, Relative expression of the top genes from pan-cancer T$_{RM}$ (**a**) and T$_{EX}$ (**b**) gene signatures across respective CD8$^+$ T cell populations in the pan-cancer atlas[28]. **c**, Relative frequencies of different CD8$^+$ T cell subsets by cancer from respective datasets[32–37].

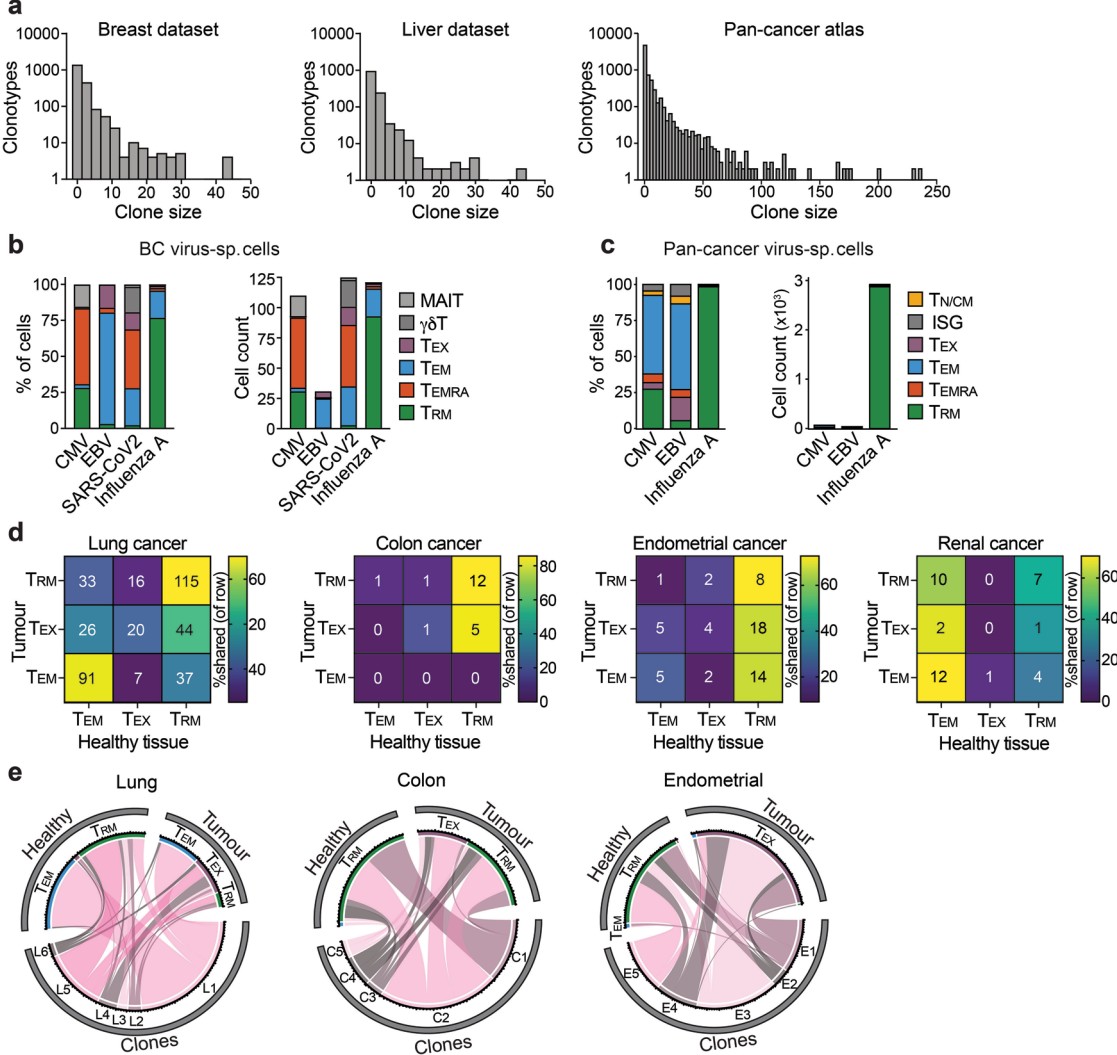

**Extended Data Fig. 6 | Assessment of virus-specific clones and TCR sharing across tissues. a**, Clone size distribution across respective datasets. **b-c**, Frequency and count of cells from BC dataset (b) and pan-cancer atlas (c) split by respective virus-specificity. **d-e**, Clonal sharing between tumour and healthy tissue-derived CD8⁺ T cells[36] related to Fig. 4j–l. Clonal sharing between subsets detected across tumour vs healthy tissue-derived CD8⁺ T cells, filtered on clones identified in both tissues. **d**, Heatmap scaled by % sharing in each row, split by respective cancers[36]. **e**, Circos plots indicating selected clones from respective cancers, and number of cells from each clonotype occupying each tissue and phenotype.

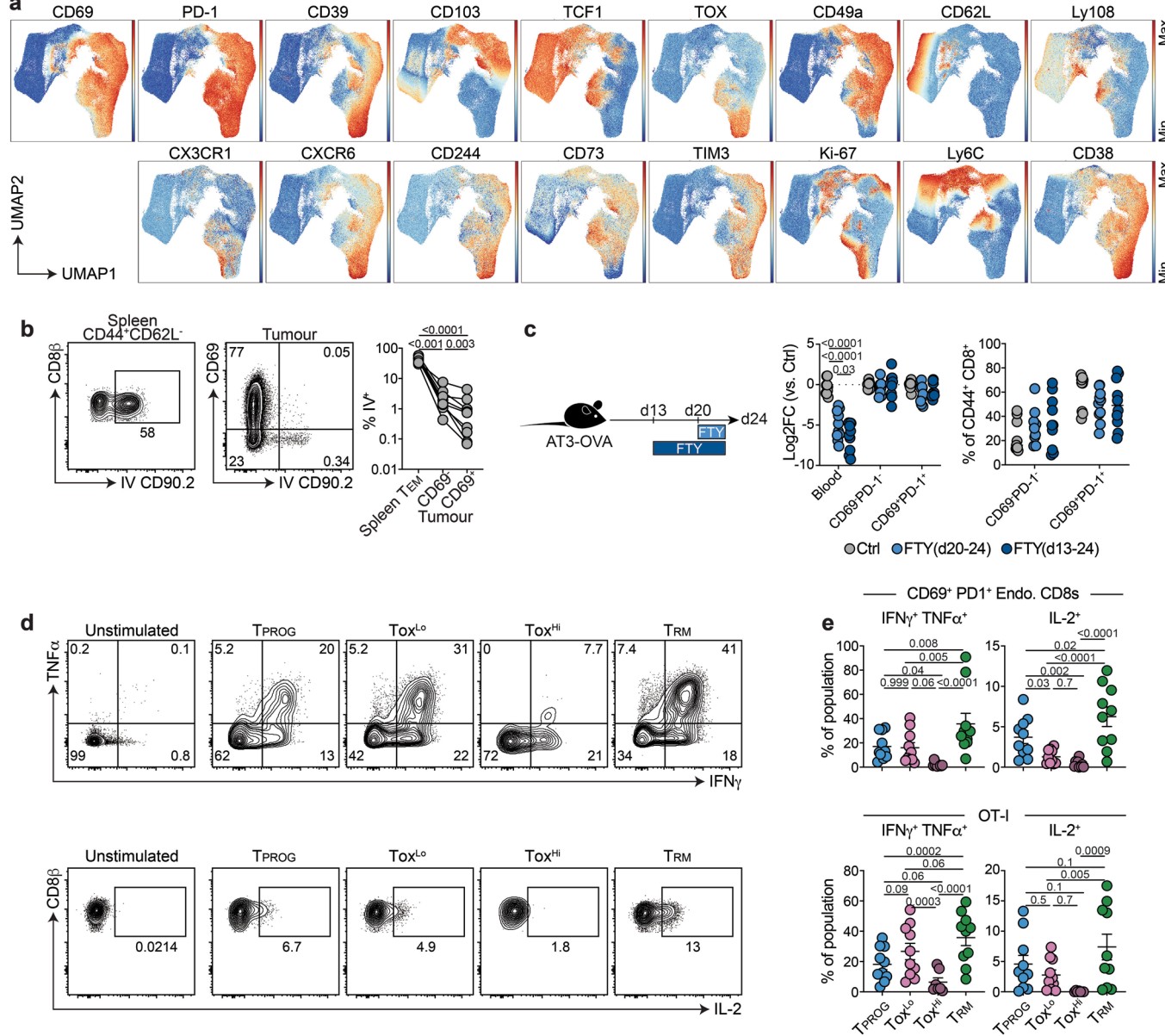

**Extended Data Fig. 7 | Phenotypic and functional assessment of AT3-OVA TIL populations. a**, Expression of respective proteins on CD8+ T cells isolated from AT3-OVA tumours projected in UMAP space, related to Fig. 5a–d. **b**, Gating and intravenous (IV) labelling of respective CD8+ T cell populations from spleen and tumour at d24 post-inoculation. **c**, FTY720 treatment of AT3-OVA bearing mice showing relative cell number (normalised to median of control group in each experiment) and frequency of CD69+ PD-1+ cells within CD44+ CD8+ T cells from tumours, and total CD8+ T cells from the blood. **d**, representative gating

of IFNγ, TNFα, and IL-2 on respective populations (as in Fig. 5e). Gates based on unstimulated control. **e**, % expression of IFNγ, TNFα, and IL-2 on respective endogenous CD8+ (top) and OT-I (bottom) T cell populations. Statistics: **a-e**, pooled from 2 independent experiments, minimum N=5 mice/group/experiment. **b**, one-way repeated-measures ANOVA with Holm-Sidak test, connected points from individual mice. **c**, Two-way ANOVA with Tukey's test. **e**, one-way repeated-measures ANOVA with Tukey post-test. Panel **c** adapted from ref. 47, Springer Nature America.

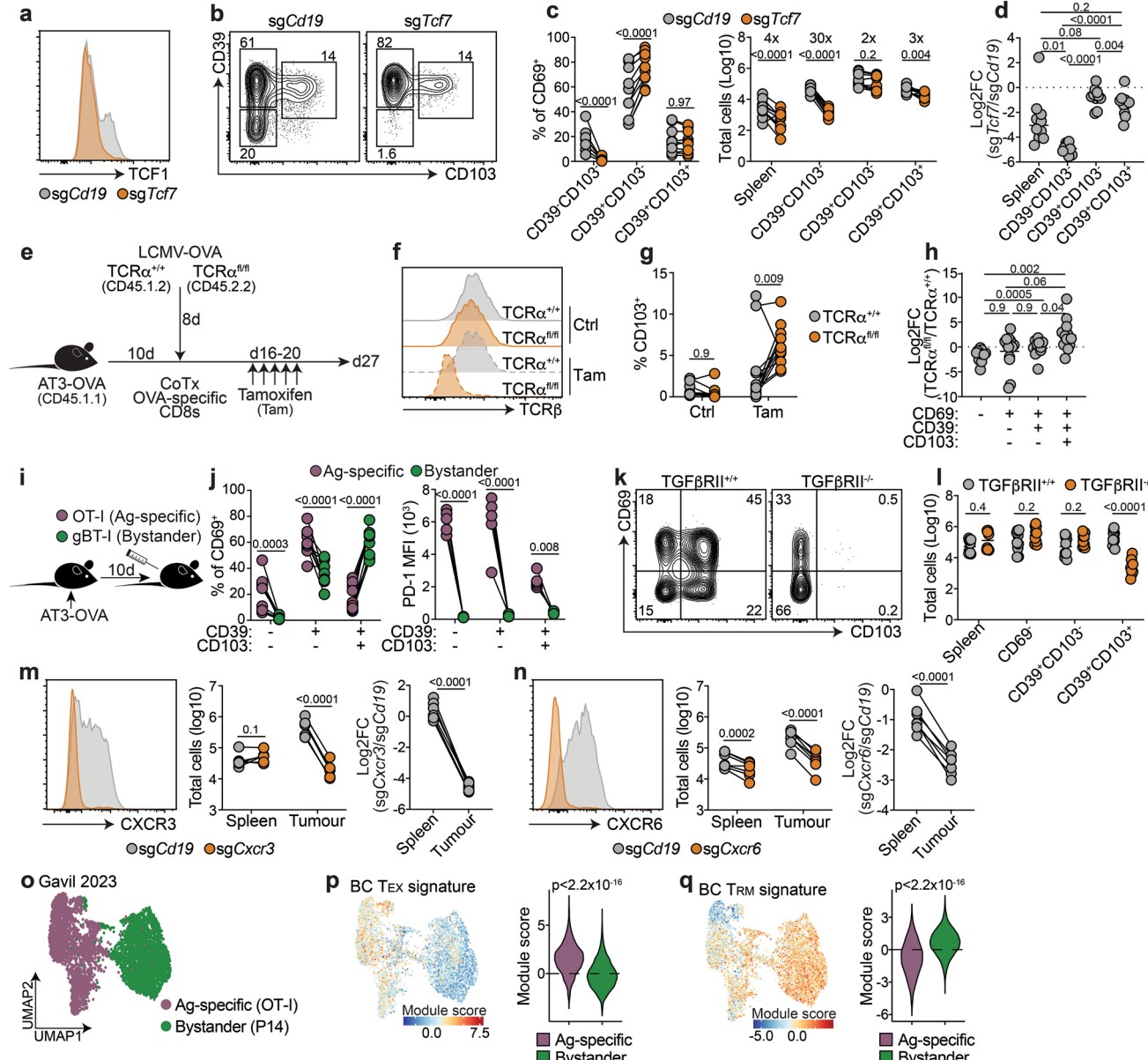

**Extended Data Fig. 8 | Mechanistic assessment of factors controlling formation of TIL populations. a-d**, CRISPR/Cas9 depletion of *Tcf7* from naïve OT-I T cells in AT3-OVA tumours harvested d24 post-inoculation. **a**, TCF1 expression on co-transferred sg*CD19* (control) and sg*Tcf7* treated OT-I T cells. **b**, Representative gating of CD69⁺ OT-I T cells. **c**, frequency and number of CD69⁺ OT-I T cells. **d**, Log2 fold-change of cell numbers. **e-f**, Impact of TCRα depletion on TIL populations. Congenically distinct TCRα⁺/⁺ Creᴱᴿᵀ² and TCRαᶠˡ/ᶠˡ Creᴱᴿᵀ² mice were infected with LCMV-OVA; SIINFEKL-tetramer⁺ CD8⁺ T cells were sorted from spleens at d8 post-infection and co-transferred into AT3-OVA bearing mice 10 d post-tumour. Tamoxifen (Tam) or vehicle control was given on d16-20 post-tumour, and harvested on d27. **e**, schematic. **f**, TCRβ staining. **g**, % CD103 expression. **h**, ratio of cell counts. **i-j**, Effector OT-I (Ag-specific) and gBT-I (bystander) CD8⁺ T cells were co-transferred into mice 10 d post-AT3-OVA inoculation, analysed 7 d later. **i**, Schematic. **j**, Frequency and PD-1 expression of CD69⁺ populations. **k-l**, Effector control (TGFβRII⁺/⁺) and TGFβRII⁻/⁻ bystander OT-I T cells transferred into AT3 tumour-bearing mice 10 d post inoculation.

**k**, gating, **l**, cell counts 6 d following transfer. **m-n**, Effector control (sg*CD19*) and CXCR3 (m, sg*CXCR3*) or CXCR6 (n, *sgCXCR6*) deficient gBT-I T cells co-transferred into AT3-OVA tumour bearing mice 10 d post-inoculation; spleen and tumours analysed 7 d later. Expression of CXCR3 or CXCR6, cell counts, and ratios of cells. **o-q**, Analysis of CITEseq of LCMV-specific P14 or tumour-specific OT-I T cells isolated from EO771-OVA BC tumours[10]. **o**, Unified cell-surface protein and RNA-seq expression data represented on weighted nearest neighbours UMAP coloured by T cell transgenic. **p-q**, quantification of BC Tₑₓ (**p**) and T_RM (**q**) signature module scores, p-value calculated by two-sided Wilcoxon signed-rank test. Statistics: **a-d**, i-l, pooled from 2 independent experiments, N=5 mice per experiment; m-n, representative of 2 independent experiments with a minimum of 5 mice per group. **c,g,j,m,n**, linked symbols indicate cells from the same mouse, 2-way repeated-measures ANOVA with Sidak post-test. **d,h**, 1-way repeated-measures ANOVA with Tukey post-test. **l**, 2-way ANOVA with Sidak post-test. p-q, Wilcoxon signed-rank test. Panels **e** and **i** adapted from ref. 47, Springer Nature America.

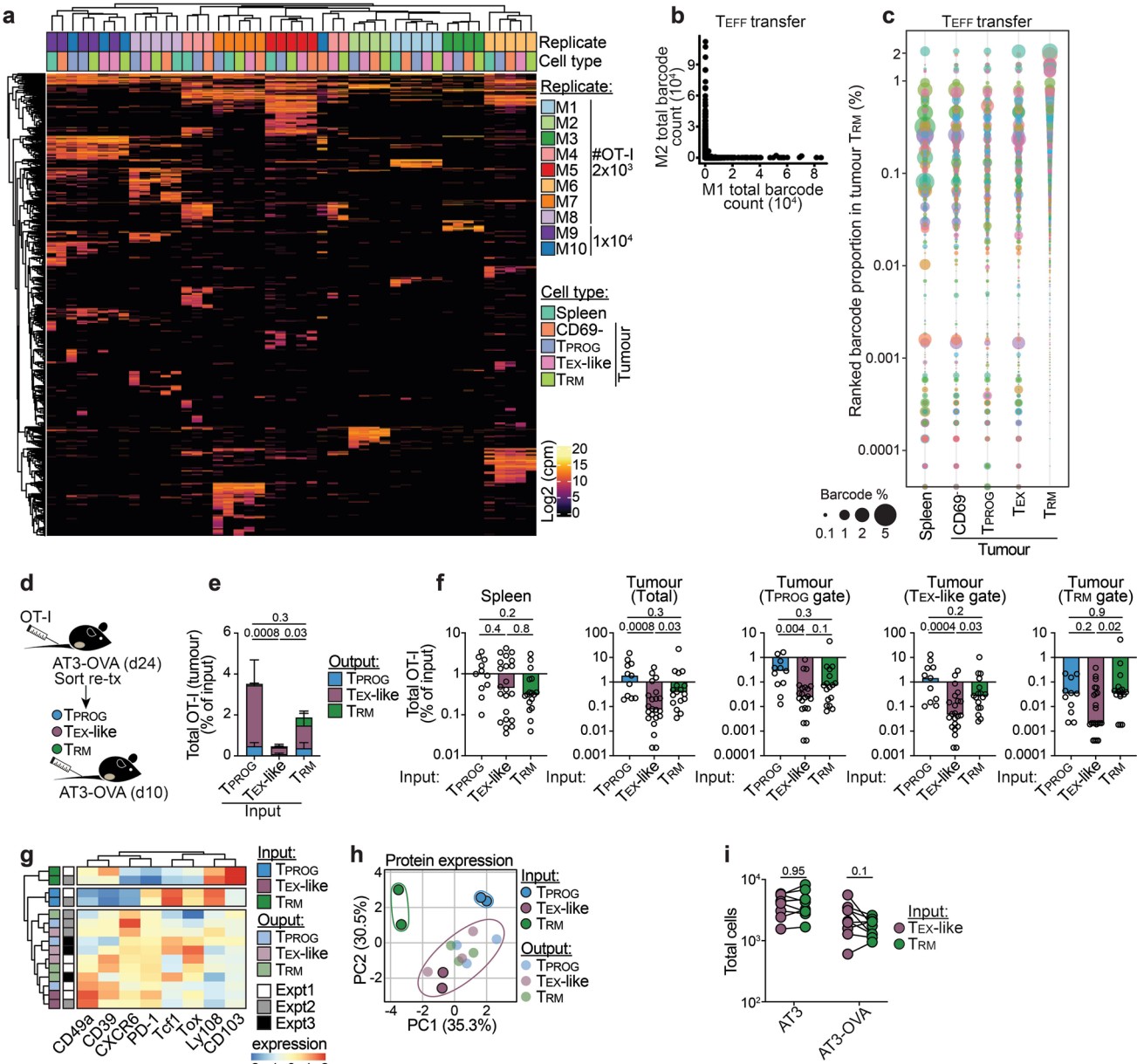

**Extended Data Fig. 9 | Developmental relationship between intratumoural T$_{RM}$ and T$_{EX}$. a-c,** Related to Fig. 6a–d. **a,** heatmap showing counts of individual barcodes (y-axis) across different replicate mice and sorted populations from naïve OT-I SPLINTR experiments. **b,** Total barcode-seq counts of each identified barcode from two (M1 vs M2) effector OT-I SPLINTR experimental replicates. **c,** Barcode-seq data from representative effector OT-I SPLINTR experiment (M1) sorted by frequency in the tumour T$_{RM}$ sample, where bubble size reflects clone proportion in sample. **d-h,** Sort and intravenous retransfer of AT3-OVA intratumoural populations. **d,** Schematic. **e,** Enumeration of transferred populations 28 d post-tumour inoculation as the frequency of input cell number. Error bars indicate mean +/- SEM. **f,** Cell numbers of respective populations isolated from spleen and tumours, normalised to input frequency, isolated 28 d post-tumour inoculation. Bars indicate mean. 1-way ANOVA on log-transformed

values, with Tukey post-test. **g,** Relative expression of proteins on respective populations of primary transferred (Input) and retransferred (Output) cells 28 d post-tumour inoculation determined by flow cytometry with unbiased hierarchical clustering indicated and **h,** PCA indicating expression of proteins in (**g**), each dot being concatenated samples from independent experiments. **i,** related to Fig. 6e–i, showing total cell counts isolated from tumours at d 22 post-tumour inoculation. Statistics: **a,** pooled from 3 independent experiments, with 10 total replicates (that is M1-10). **b,** M1 and M2 indicate independent experiments. **d-h,** Data pooled from 3 independent experiments, N=11 (T$_{PROG}$), 23 (T$_{EX}$), 17 (T$_{RM}$) mice. **i,** Data representative of 2-independent experiments, linked symbols indicate cells isolated from the same mouse, N=9 mice per group, analysed by 2-way repeated-measures ANOVA with Sidak post-test. Panel **d** adapted from ref. 47, Springer Nature America.

# Reporting Summary

## Statistics

For all statistical analyses, confirm that the following items are present in the figure legend, table legend, main text, or Methods section.

| n/a | Confirmed | |
|---|---|---|
| ☐ | ☒ | The exact sample size (*n*) for each experimental group/condition, given as a discrete number and unit of measurement |
| ☐ | ☒ | A statement on whether measurements were taken from distinct samples or whether the same sample was measured repeatedly |
| ☐ | ☒ | The statistical test(s) used AND whether they are one- or two-sided<br>*Only common tests should be described solely by name; describe more complex techniques in the Methods section.* |
| ☒ | ☐ | A description of all covariates tested |
| ☐ | ☒ | A description of any assumptions or corrections, such as tests of normality and adjustment for multiple comparisons |
| ☐ | ☒ | A full description of the statistical parameters including central tendency (e.g. means) or other basic estimates (e.g. regression coefficient) AND variation (e.g. standard deviation) or associated estimates of uncertainty (e.g. confidence intervals) |
| ☐ | ☒ | For null hypothesis testing, the test statistic (e.g. *F*, *t*, *r*) with confidence intervals, effect sizes, degrees of freedom and *P* value noted<br>*Give P values as exact values whenever suitable.* |
| ☒ | ☐ | For Bayesian analysis, information on the choice of priors and Markov chain Monte Carlo settings |
| ☒ | ☐ | For hierarchical and complex designs, identification of the appropriate level for tests and full reporting of outcomes |
| ☒ | ☐ | Estimates of effect sizes (e.g. Cohen's *d*, Pearson's *r*), indicating how they were calculated |

*Our web collection on statistics for biologists contains articles on many of the points above.*

## Software and code

Policy information about availability of computer code

| | |
|---|---|
| Data collection | Flow cytometry data were collected with SpectroFlo v3.0 (Cytek). CITEseq data were collected using chromium controller (10X Genomics). Cyclic IF data was obtained using a Cytefinder II HT (Rarecyte) slide scanning fluorescent microscope. |
| Data analysis | Spectral flow cytometry data was unmixed with SpectroFlo v3.0 software, and then analysed with FlowJo (v.10.10.0; Treestar) or OMIQ. CITEseq data were analysed with CellRanger (6.1.2), Seurat (4.3.0), R(4.2), Harmony as well as other tools cited in the methods section. CycIF data were analysed using Cecelia. All code is available on Github. |

For manuscripts utilizing custom algorithms or software that are central to the research but not yet described in published literature, software must be made available to editors and reviewers. We strongly encourage code deposition in a community repository (e.g. GitHub). See the Nature Portfolio guidelines for submitting code & software for further information.

## Data

Policy information about availability of data

All manuscripts must include a data availability statement. This statement should provide the following information, where applicable:
- Accession codes, unique identifiers, or web links for publicly available datasets
- A description of any restrictions on data availability
- For clinical datasets or third party data, please ensure that the statement adheres to our policy

> All data are available from the corresponding author upon reasonable request. The single-cell CITE and RNA-sequencing data generated from this study is available in GEO. Source data will be provided with this paper.

## Research involving human participants, their data, or biological material

Policy information about studies with human participants or human data. See also policy information about sex, gender (identity/presentation), and sexual orientation and race, ethnicity and racism.

| | |
|---|---|
| Reporting on sex and gender | Breast cancer patients were all female, except 1 male donor used for CycIF imaging. |
| Reporting on race, ethnicity, or other socially relevant groupings | Race and ethnicity were not recorded or relevant. Researchers were blinded to these characteristics. |
| Population characteristics | Medical diagnosis and treatments |
| Recruitment | Patients with breast cancer undergoing mastectomy or lumpectomy were recruited by Dr. Simon Tsao, at St Vincent's Hospital, Melbourne, Australia. Colorectal cancer patients with metastases to the liver undergoing surgical resection of tumours were recruited by Dr Marcos Perini without restriction or bias. |
| Ethics oversight | All study on human specimens was approved by the Human Research Ethics Committee of the University of Melbourne (ID nos. 13009 and 14517). All participating patients provided written informed consent. |

Note that full information on the approval of the study protocol must also be provided in the manuscript.

# Field-specific reporting

Please select the one below that is the best fit for your research. If you are not sure, read the appropriate sections before making your selection.

☒ Life sciences    ☐ Behavioural & social sciences    ☐ Ecological, evolutionary & environmental sciences

For a reference copy of the document with all sections, see nature.com/documents/nr-reporting-summary-flat.pdf

# Life sciences study design

All studies must disclose on these points even when the disclosure is negative.

| | |
|---|---|
| Sample size | Sample sizes are all reported in the respective figure legends and methods. For mouse experiments, sample sizes were chosen based on previous analyses in Prof. Mackay's laboratory demonstrating sufficient power to detect effects in similarly designed studies. Human experiments used a minimum of 5 independent donors, and findings were repeated and validated across different tumour settings, and cross-validated in public datasets |
| Data exclusions | No data were excluded from the respective analyses |
| Replication | All replication has been stated in the respective figure legends and methods. All mouse experiments were repeated a minimum of two times. |
| Randomization | Mice were randomised to individual groups. |
| Blinding | Investigators were not blinded to groups, as this was not necessary or relevant for the study. |

# Reporting for specific materials, systems and methods

We require information from authors about some types of materials, experimental systems and methods used in many studies. Here, indicate whether each material, system or method listed is relevant to your study. If you are not sure if a list item applies to your research, read the appropriate section before selecting a response.

## Materials & experimental systems

| n/a | Involved in the study |
|---|---|
| ☐ | ☒ Antibodies |
| ☐ | ☒ Eukaryotic cell lines |
| ☒ | ☐ Palaeontology and archaeology |
| ☐ | ☒ Animals and other organisms |
| ☒ | ☐ Clinical data |
| ☒ | ☐ Dual use research of concern |
| ☒ | ☐ Plants |

## Methods

| n/a | Involved in the study |
|---|---|
| ☒ | ☐ ChIP-seq |
| ☐ | ☒ Flow cytometry |
| ☒ | ☐ MRI-based neuroimaging |

## Antibodies

| Antibodies used | Species Specificity Fluorochrome Clone Source Catalog # RRID Dilution |
|---|---|
| | Human CD4 BUV496 SK3 BD 612936 AB_2870220 50 |
| | Human CD3 BUV737 UCHT1 BD 612750 AB_2870081 50 |
| | Human CD8b BUV805 2ST8.547 BD 749366 AB_2873737 200 |
| | Human CD103 BV480 Ber-ACT8 BD 746472 AB_2743774 25 |
| | Human CD45RA BV510 HI100 Biolegend 304138 AB_2561460 50 |
| | Human CCR7 PE-CF594 150503 BD 562381 AB_11153301 50 |
| | Human CD161 PE-Cy7 HP-3G10 Biolegend 339918 AB_11126745 50 |
| | Human TCR Va7.2 BV711 3C10 Biolegend 351732 AB_2629680 50 |
| | Human CD69 APC-Cy7 FN50 BD 557756 AB_396862 25 |
| | Human CD8a BV650 RPA-T8 Biolegend 301042 AB_2563505 100 |
| | Human CD73 BV421 AD2 Biolegend 344008 AB_11204424 25 |
| | Human CD38 ef450 HIT2 eBioscience 48-0389-42 AB_11151696 25 |
| | Human CD161 BV605 HP-3G10 Biolegend 339916 AB_2563607 25 |
| | Human CCR7 BV650 GO43H7 Biolegend 353234 AB_2563867 25 |
| | Human IFNg BV786 4S.B3 Biolegend 502509 AB_315234 25 |
| | Human TNF FITC MAb11 BD 562082 AB_395443 50 |
| | Human IL-2 PerCP-ef710 MQ1-17H12 eBioscience 46-7029-42 AB_1834419 25 |
| | Human CD101 PE BB27 Biolegend 331012 AB_2716107 25 |
| | Human CD39 PE-Dazzle594 A1 Biolegend 328224 AB_2564319 25 |
| | Human CD107a PE-Cy5 eBioH4A3 Invitrogen 15-1079-42 AB_10547280 100 |
| | Human IL-17A PE-Cy7 BL168 Biolegend 512315 AB_2295923 25 |
| | Human CD94 APC DX22 Biolegend 305508 AB_2133129 25 |
| | Mouse CD44 BUV395 IM7 BD 568507 AB_2739963 400 |
| | Mouse Va2 BUV615 B20.1 BD 751416 AB_2875415 400 |
| | Mouse CD4 BUV805 RM4-4 BD 741913 AB_2871227 400 |
| | Mouse CD103 BV480 M290 BD 566118 AB_2739520 200 |
| | Mouse CD39 BV711 Y23-1185 BD 567295 AB_2916538 400 |
| | Mouse CD8b BV750 H35-17.2 BD 747505 AB_2872172 500 |
| | Mouse PD-1 BV785 29F.1A13 Biolegend 135225 AB_2563680 200 |
| | Mouse TCF1 AF488 C63D9 Cell Signalling 6444S AB_2797627 200 |
| | Mouse CD90.2 PerCP 53-2.1 Biolegend 140316 AB_10642813 400 |
| | Mouse CD69 PE-Cy5 H1.2F3 Biolegend 104510 AB_313113 200 |
| | Mouse Tim3 PE-Cy7 RMT3-23 Biolegend 119716 AB_2571933 200 |
| | Mouse TOX e660 TXRX10 Thermo 50-6502-82 AB_2574265 200 |
| | Mouse CD45.1 APC-R700 A20 BD 565813 AB_2744397 200 |
| | Mouse CD45.2 PE-Cy7 104 Biolegend 109830 AB_1186098 200 |
| | Mouse CD39 PE 24DMS1 eBioscience 12-0391-82 AB_1210740 400 |
| | Mouse CD45.2 SparkNIR685 104 Biolegend 109864 AB_2876424 200 |
| | Mouse CD4 BUV496 RM4-5 BD 741050 AB_2870665 400 |
| | Mouse CD8b AF700 YTS156.7.7 Biolegend 126618 AB_2563949 400 |
| | Mouse CD45.1 PE-Cy7 A20 BD 560578 AB_1727488 200 |
| | Mouse CD39 AF700 24DMS1 eBioscience 56-0391-82 AB_2662998 200 |
| | Mouse CD103 PerCP-Cy5.5 2E7 Biolegend 121416 AB_2128621 200 |
| | Mouse Va2 BV421 B20.1 BD 562944 AB_2737910 400 |
| | Mouse CD49a BUV395 Ha31/8 BD 740262 AB_2740005 200 |
| | Mouse Ly108 BV421 13G3 BD 740090 AB_2739850 200 |
| | Mouse CXCR6 PE-ef610 SA051D1 Biolegend 151104 AB_2566546 200 |
| | Mouse TCRb APC-Fire750 H57-597 BD 561080 AB_398534 200 |
| | Mouse CD45.1 APC A20 Biolegend 110714 AB_313503 200 |
| | Mouse CD69 PE-Cy7 H1.2F3 BD 552879 AB_394508 200 |
| | Mouse CD103 FITC 2E7 Thermo 11-1031-82 AB_465176 200 |
| | Streptavidin PE  BD 554061 AB_10053328 |
| | SIINFEKL monomer  NIH Tetramer core  400 |
| | Human CD39 AB_2889212 none EPR20627 Abcam ab223842 |
| | Human Rabbit IgG AB_2534114 Alexa Fluor 488 polyclonal Invitrogen A-11070 |
| | Human CD105 AB_354598 none polyclonal R&D Systems AF1097 |
| | Human Goat IgG AB_2535853 Alexa Fluor 555 polyclonal Thermo Fisher Scientific A-21432 |

Human CD33 AB_2943218 none PWS44 BioCare ACI 3116 A
Human Mouse IgG AB_2535806 Alexa Fluor 647 polyclonal Thermo Fisher Scientific A-21237
Human CD45RO AB_2237910 none UCHL1 Dako M0742
Human Mouse IgG2a AB_2338855 Alexa Fluor 488 polyclonal Jackson ImmunoResearch, Inc. 115-545-206
Human CD3E AB_3668891 Alexa Fluor 555 D7A6E Cell Signaling Technology 57869S
Human CD25 AB_2572851 none GLLZDMY Thermo Fisher Scientific 14-0256-82
Human Mouse IgG1 AB_2338916 Alexa Fluor 647 polyclonal Jackson ImmunoResearch, Inc. 115-605-205
Human phospho-Histone 3 AB_3668896 Alexa Fluor 750 D2C8 Cell Signaling Technology 43185S
Human NT5E/CD73  AB_2716625 Alexa Fluor 488 D7F9A Cell Signaling Technology 14627BC
Human Ki-67 AB_2797836 phycoerythrin D3B5 Cell Signaling Technology 12160S
Human CD31 AB_2857973 Alexa Fluor 647 EPR3094 Abcam ab218582
Human Cytokeratin (pan) AB_2868569 Alexa Fluor 750 AE-1/AE-3 Novus Biologicals NBP2-33200AF750
Human CD8a AB_2574412 Alexa Fluor 488 AMC908 Thermo Fisher Scientific 53-0008-80
Human GZMB  Alexa Fluor 555 D6E9W Cell Signaling Technology 29268S
Human CD103 AB_2884945 Alexa Fluor 647 EPR4166(2) Abcam ab225153
Human LAG3  Alexa Fluor 488 EPR4392(2) Abcam ab225277
Human CD11c AB_3331656 Alexa Fluor 555 D3V1E Cell Signaling Technology 77882S
Human PD-1 AB_2728811 Alexa Fluor 647 EPR4877(2) Abcam ab201825
Human Alpha-actin-2 AB_2868436 Alexa Fluor 750 1A4 R&D Systems IC1420S-025
Human TIM3 AB_2799468 Alexa Fluor 488 D5D5R Cell Signaling Technology 54669S
Human Catenin beta-1 AB_2868511 Alexa Fluor 555 E247 Abcam ab202496
Human CD69  Alexa Fluor 647 EPR21814 Abcam ab313397
Human TCF1/TCF7 AB_2797627 Alexa Fluor 488 C63D9 Cell Signaling Technology 6444S
Human FOXP3 AB_2573609 eFluor 570 236A/E7 eBioscience 41-4777-82
Human GNLY   Alexa Fluor 647 E2T3D Cell Signaling Technology 64458S
Human CD68 AB_2798886 Alexa Fluor 488 D4B9C Cell Signaling Technology 24850S
Human CD66b AB_2750201 phycoerythrin 6/40c BioLegend 392903
Human CD20 AB_11151691 eFluor 660 L26 eBioscience 50-0202-82
Human Mast Cell Tryptase AB_2943323 Alexa Fluor 790 AA1 Santa Cruz Biotechnology sc-59587 AF790
Human Vimentin AB_10829352 Alexa Fluor 488 D21H3 Cell Signaling Technology 9854S
Human HLA -DRA AB_2889281 Alexa Fluor 555 EPR3692 Abcam ab215312
Human HER2 AB_2889201 Alexa Fluor 647 EPR19547-12 Abcam ab225510
Human CD4 AB_2728839 Alexa Fluor 488  R&D Systems FAB8165G
Human CD206  Alexa Fluor 555 E2L9N Cell Signaling Technology 48352S
Human Progesterone receptor AB_2890175 Alexa Fluor 647 YR85 Abcam ab199455
Human c-Jun (pS73) AB_2798004 Alexa Fluor 488 D47G9 Cell Signaling Technology 12714S
Human phospho-MAPK  Alexa Fluor 555 D13.14.4E Cell Signaling Technology 76032S
Human YAP1 AB_2728837 Alexa Fluor 647 D8H1X Cell Signaling Technology 38707S
Human PCNA AB_3669072 Alexa Fluor 750 PC10 Cell Signaling Technology 24114BC
Human HLA-A and HLA-B AB_2943099 Alexa Fluor 488 EPR1394Y Abcam ab198376
Human IBA1 AB_2943227 Alexa Fluor 555 E4O4W Cell Signaling Technology 36618S
Human ER alpha AB_2728817 Alexa Fluor 647 EPR4097 Abcam ab205851
Human CD15 AB_493257 Alexa Fluor 488 HI98 BioLegend 301910
Human CD7 AB_2889193 Alexa Fluor 647 EPR4242 Abcam ab199023
Human KLF2  Alexa Fluor 555 E7K8Y Cell Signaling Technology 73198BC
Human CD11b AB_2637195 Alexa Fluor 488 C67F154 Thermo Fisher Scientific (eBioscience) 53-0196-82
Human NKG2A AB_2943211 Alexa Fluor 647 EPR23737-127 Abcam ab300745
Human CD94  Alexa Fluor 555 EPR21003 Abcam ab318870
Human MX1 AB_2799122 Alexa Fluor 488 D3W7I Cell Signaling Technology 79373BC
Human ICOS AB_2800142 Alexa Fluor 555 D1K2T Cell Signaling Technology 79403BC
Human CD16 AB_626925 Alexa Fluor 647 DJ130c Santa Cruz Biotechnology sc-20052 AF647

| Validation | Antibodies used for flow cytometry were validated for the corresponding application by the manufacturers. Antibodies were also tested internally for the corresponding applications internally through preliminary experiments using primary cells from humans and mice before performing experiments using patient cells. |
| --- | --- |

# Eukaryotic cell lines

Policy information about cell lines and Sex and Gender in Research

| Cell line source(s) | AT3-OVA cells were provided by Professor Philip Darcy, Peter MacCallum Cancer Centre. B16F1-gB.GFP (B16-gB) cells were provided by Jason Waithman (University of Western Australia). |
| --- | --- |
| Authentication | These cell lines were not authenticated. |
| Mycoplasma contamination | These cell lines were tested negative for mycoplasma contamination before experimental use. |
| Commonly misidentified lines (See ICLAC register) | No commonly misidentified cell lines were used |

# Animals and other research organisms

Policy information about studies involving animals; ARRIVE guidelines recommended for reporting animal research, and Sex and Gender in Research

| | |
|---|---|
| Laboratory animals | C57Bl/6 female mice were used between 7-12 weeks of age at the beginning of the experiments |
| Wild animals | NA |
| Reporting on sex | All mice used were females due to the focus on breast cancer models. |
| Field-collected samples | NA |
| Ethics oversight | All animal experiments were approved by The University of Melbourne Animal Ethics Committee (ID nos. 21651 and 21938). |

Note that full information on the approval of the study protocol must also be provided in the manuscript.

# Plants

| | |
|---|---|
| Seed stocks | *Report on the source of all seed stocks or other plant material used. If applicable, state the seed stock centre and catalogue number. If plant specimens were collected from the field, describe the collection location, date and sampling procedures.* |
| Novel plant genotypes | *Describe the methods by which all novel plant genotypes were produced. This includes those generated by transgenic approaches, gene editing, chemical/radiation-based mutagenesis and hybridization. For transgenic lines, describe the transformation method, the number of independent lines analyzed and the generation upon which experiments were performed. For gene-edited lines, describe the editor used, the endogenous sequence targeted for editing, the targeting guide RNA sequence (if applicable) and how the editor was applied.* |
| Authentication | *Describe any authentication procedures for each seed stock used or novel genotype generated. Describe any experiments used to assess the effect of a mutation and, where applicable, how potential secondary effects (e.g. second site T-DNA insertions, mosiacism, off-target gene editing) were examined.* |

# Flow Cytometry

## Plots

Confirm that:

☒ The axis labels state the marker and fluorochrome used (e.g. CD4-FITC).

☒ The axis scales are clearly visible. Include numbers along axes only for bottom left plot of group (a 'group' is an analysis of identical markers).

☒ All plots are contour plots with outliers or pseudocolor plots.

☒ A numerical value for number of cells or percentage (with statistics) is provided.

## Methodology

| | |
|---|---|
| Sample preparation | Cells were isolated from tissues as described in the methods. Single cell suspensions were then stained with surface stain and fixable Live/Dead stain (Zombie Near IR, Zombie Aqua, Zombie Yellow). When needed, cells were fixed/permeabilised with FoxP3 transcription factor kit (eBioscience) and stained with intracellular antibodies. |
| Instrument | Cytek Aurora Flow Cytometer |
| Software | SpectroFlo v3.0 software was used for spectral unmixing, and then analysed with FlowJo (v.10.10.0; Treestar) or OMIQ |
| Cell population abundance | Subset abundance was determined by manual gating in FlowJo or high-dimensional analysis using OMIQ, with clustering using the FlowSOM algorithm |
| Gating strategy | Lymphocytes were gating based on FSC/SSC, singlets gated based on FSC-W vs FSC-H and SSC-W vs SSC-H. Live cells were gated based on live cell exclusion. Further gates are described for respective figures |

☒ Tick this box to confirm that a figure exemplifying the gating strategy is provided in the Supplementary Information.

