## [Peer Review File · Nature Immunology]

Antigen reactivity defines tissue-resident memory and exhausted T cells in tumours

Corresponding Author: Professor Laura Mackay

Version 0:

Decision Letter:

21st Feb 2025

Dear Professor Mackay,

Thank you for transferring your manuscript and rebuttal to the Nature reviewers for "Antigen reactivity defines tissue-resident memory and exhausted T cells in tumours".

Although there are clearly some major issues raised by the reviewers, we would be happy to look at a revision in which you perform the outlined experiments in your plan. We are particularly keen to see expansion of the mechanistic insight noted by reviewer 2 with regards to what that reviewer considers to be the main novelty in figure 5 and 6. Also, we note that reviewer 3 is concerned that the paper somewhat 'muddies the waters' more than it clarifies (private comments to the editors). So although the paper is generally very well laid out and uses clear language, we will be looking for enhanced clarity in the nomenclature, cellular identification and related conclusions. I would be happy to set up a VC call at some point if you wish.

That said, we look forward to the revision and suggest you link up with John Wherry to try as best you can to sync up the resubmission of your two papers, but please note this is not essential, it just makes things easier on our end if we have overlapping reviewers.

Nature Immunology is committed to improving transparency in authorship. As part of our efforts in this direction, we are now requesting that all authors identified as 'corresponding author' on published papers create and link their Open Researcher and Contributor Identifier (ORCID) with their account on the Manuscript Tracking System (MTS), prior to acceptance. ORCID helps the scientific community achieve unambiguous attribution of all scholarly contributions. You can create and link your ORCID from the home page of the MTS by clicking on 'Modify my Springer Nature account'. For more information please visit www.springernature.com/orcid.

We shall hope to receive your revised version as soon as possible. If you anticipate a delay of more than 3 months, however, please let us know. We will be happy to consider your revision so long as nothing similar has been accepted for publication at Nature Immunology or published elsewhere. Should your manuscript be substantially delayed without notifying us in advance and your article is eventually published, the received date may be that of the revised, not the original, version.

Please note, Extended Data figures and tables are online-only (appearing in the online PDF and full-text HTML version of the paper), peer-reviewed display items that provide essential background to the Article but are not included in the printed version of the paper due to space constraints or being of interest only to a few specialists. A maximum of ten Extended Data display items (figures and tables) is typically permitted. When re-submitting your manuscript, please ensure that any supplementary figures and tables that are more critical to the manuscript's conclusions are converted to Extended data to increase these data's visibility.

If you are not interested in submitting a suitably revised manuscript in the future please let me know immediately so we can close your file. If you have any questions, please contact me.

Please use the link below to submit a suitably revised manuscript and updated response to referees when they are ready.

Link Redacted

Sincerely,

Nick Bernard, PhD
Senior Editor
Nature Immunology

Version 1:

Reviewer comments:

Reviewer #1

(Remarks to the Author)

The authors have addressed all of my comments satisfactorily. I have no additional concerns.

Reviewer #2

(Remarks to the Author)

The authors have carefully and comprehensively addressed all of the comments through new data, analyses, and extensive revisions throughout. It is an important study that addresses major questions in the field of tissue resident memory T cells, their generation, functional maintenance and the role of tumors in affecting these processes.

Reviewer #3

(Remarks to the Author)

In this revised and transferred manuscript, Burn et al., made several revisions and demonstrated that location/Ag-density have a pronounced impact on T cell differentiation and responsiveness to CBI. The authors do a nice job of characterizing T cells in tumors and provide updated gene signatures that can bin cells, derived from the same tumor, into one of two resident categories: low function/proliferation/chronically stimulated (Tex) and more function/putatively not chronically stimulated or cells that at least have found a respite from Ag (Trm). The human data was a particular strength. Although I still sometimes struggled with the vocabulary and articulation of some messages, the revised text was much improved and appropriately circumspect.

Remaining comment/food for thought:

Fig 5: The authors create a rule that every T cell in a tumor must be classified as either exhausted or resident, and then create criteria for binning cells into one classification or the other. While this still remains tautological, I agree that there are reasonable criteria included in the classification. But should the populations be so unequivocally discrete? Are all Trm a "different" cell type or do they include cells (in this paper's analysis) that are not yet fully "exhausted"? This relates to Gavil et al. claim that more time within tumors promotes a gradual increase in exhaustion or exhaustion markers among tumor specific T cells. Putting this another way, are some "Trm" just cells that are recent arrivals in the tumor, and on a trajectory towards exhaustion, but not there yet. All this is compatible with the authors data that Trm can be driven to exhaustion, and so I am not saying anything that the authors don't know, but I think the transitional states among tumor-specific cells gets lost in the paper's binary presentation.

Decision Letter:

Our ref: NI-A39737A

2nd Sep 2025

Dear Dr. Mackay,

Thank you for submitting your revised manuscript "Antigen reactivity defines tissue-resident memory and exhausted T cells in tumours" (NI-A39737A). It has now been seen by the original referees and their comments are below. The reviewers find that the paper has improved in revision, and therefore we'll be happy in principle to publish it in Nature Immunology, pending minor revisions to satisfy the referees' final requests and to comply with our editorial and formatting guidelines.

We will now perform detailed checks on your paper and will send you a checklist detailing our editorial and formatting requirements in about a week. Please do not upload the final materials and make any revisions until you receive this additional information from us.

If you had not uploaded a Word file for the current version of the manuscript, we will need one before beginning the editing process; please email that to immunology@us.nature.com at your earliest convenience.

Thank you again for your interest in Nature Immunology Please do not hesitate to contact me if you have any questions.

Sincerely,

Nick Bernard, PhD
Senior Editor
Nature Immunology

Reviewer #1 (Remarks to the Author):

The authors have addressed all of my comments satisfactorily. I have no additional concerns.

Reviewer #2 (Remarks to the Author):

The authors have carefully and comprehensively addressed all of the comments through new data, analyses, and extensive revisions throughout. It is an important study that addresses major questions in the field of tissue resident memory T cells, their generation, functional maintenance and the role of tumors in affecting these processes.

Reviewer #3 (Remarks to the Author):

In this revised and transferred manuscript, Burn et al., made several revisions and demonstrated that location/Ag-density have a pronounced impact on T cell differentiation and responsiveness to CBI. The authors do a nice job of characterizing T cells in tumors and provide updated gene signatures that can bin cells, derived from the same tumor, into one of two resident categories: low function/proliferation/chronically stimulated (Tex) and more function/putatively not chronically stimulated or cells that at least have found a respite from Ag (Trm). The human data was a particular strength. Although I still sometimes struggled with the vocabulary and articulation of some messages, the revised text was much improved and appropriately circumspect.

Remaining comment/food for thought:

Fig 5: The authors create a rule that every T cell in a tumor must be classified as either exhausted or resident, and then create criteria for binning cells into one classification or the other. While this still remains tautological, I agree that there are reasonable criteria included in the classification. But should the populations be so unequivocally discrete? Are all Trm a "different" cell type or do they include cells (in this paper's analysis) that are not yet fully "exhausted"? This relates to Gavil et al. claim that more time within tumors promotes a gradual increase in exhaustion or exhaustion markers among tumor specific T cells. Putting this another way, are some "Trm" just cells that are recent arrivals in the tumor, and on a trajectory towards exhaustion, but not there yet. All this is compatible with the authors data that Trm can be driven to exhaustion, and so I am not saying anything that the authors don't know, but I think the transitional states among tumor-specific cells gets lost in the paper's binary presentation.

made.

Point by Point Response for “Antigen reactivity defines tissue-resident memory and exhausted T cells in tumours”

We thank the Reviewers for their thoughtful comments on our manuscript, and we will address these comments as outlined in our response below. All Reviewers recognise the significance of this study, with Reviewer 1 stating “the study is interesting and addresses an important knowledge gap”, Reviewer 2 stating “the experiments are well done, text succinct and well explained, and the findings are of interest” and Reviewer 3 stating that our manuscript “nicely illustrates the differences between populations of TILs, demonstrating the limitations of previously published analyses, and brings together sophisticated analyses to diverse datasets”. Nonetheless, the Reviewers raise reasonable concerns that we can address experimentally, which build upon our manuscript's central message to reinforce the distinct development of T_{RM} and T_{EX} in the context of cancer. Given the continued conflation of these distinct cell types across the current literature, we believe the publication of this study is timely and important.

In brief, our revision experiments will:

(i) Determine the relative localisation of T_{RM} and T_{EX} in tumours

As outlined in detail below, we propose new experiments that utilise our discovery of novel protein markers that distinguish breast cancer (BC)-associated T_{RM} and T_{EX} to perform high-dimensional immunofluorescence microscopy to identify the relative localisation of these cells and the potential interacting cell types in human BC. We will perform concurrent experiments in mice to determine whether tumour-antigen specificity dictates differential localisation of tumour-specific and bystander T_{RM} .

(ii) Determine whether distinct naïve T cells preferentially give rise to T_{EX} or T_{RM}

The SPLINTR lineage tracing model, which Reviewer 2 described as ‘beautiful’, allowed us to determine that T_{RM} and T_{EX} can develop from effector T cells that enter tumours without bias. To expand upon this and further develop this part of the study, we will generate naïve SPLINTR-barcoded T cells that will allow us to determine whether in vivo-primed naïve T cells are preferentially directed to a T_{EX} or T_{RM} fate.

(iii) Determine the role of T_{RM} in preventing tumour recurrence

Given the positive association between expression of the T_{RM} gene signature and survival of BC patients, Reviewer 3 asks whether we can show causality of this association, and specifically how T_{RM} may provide protection. The possibility exists that T_{RM} are developing in conjunction with the resolution of inflammation, as is the case for virus-specific T_{RM} upon viral clearance. Thus, the high expression of the T_{RM} in excised tumours may reflect a successful immune response that is correlative with but not specifically dependent on T_{RM} function, especially given many of the *bona fide* T_{RM} in the excised tumours are likely bystander T cells. However, in a non-mutually exclusive hypothesis, we envisage that tumour-specific T_{RM} present in surrounding, non-cancerous tissue, may provide long-term protection from tumour recurrence. To this point, we will conduct experiments to investigate the role of tumour-associated T_{RM} in preventing tumour recurrence. Specifically, we will use immune checkpoint blockade to regress murine BC tumours, followed by memory T cell depletion combined with FTY720 treatment to block T cell recirculation. These experiments will assess the necessity of T_{RM} in maintaining local tumour control and preventing recurrence.

Referee #1:

In “Antigen reactivity defines tissue-resident memory and exhausted T cells in tumours”, the authors investigated the difference between Trm and Tex. They showed that Tex and Trm have different gene expression and clonality. Trm signature in human cancer is associated with better survival but not response to ICB. Computational analysis suggested that Trm in human tumor mainly consist of bystanders. The authors further validated their in silico analysis in mouse tumor models where low affinity clones and bystanders preferentially differentiate into Trm rather than Tex. Overall, the study is interesting and addresses an important knowledge gap.

Major concerns

1) Methods: Was lymphocyte enrichment by Percoll gradient also done with BC samples? If not, why? Catalog number of freezing medium should be provided in the Methods. Why does liver require a different freezing medium? How was the difference between frozen samples and fresh samples controlled during analysis?

Percoll enrichment was not performed on the BC samples given that the cell preparation from this tissue contains much less cellular debris than from the liver and was not required. Liver-derived lymphocytes are highly susceptible to cell death; therefore, we used Cryostor CS10 freezing medium to improve the viability of these samples.

The Reviewer makes an important point regarding sample control, and indeed we took great care to treat all our samples consistently for analyses where we performed direct comparisons. Specifically, we used only fresh samples for flow cytometry, while all cells for sequencing were frozen and processed together in a single sequencing run. Additionally, we digested liver samples with collagenase, consistent with the protocol used for breast cancer samples, to ensure consistency and to control for any gene expression changes induced by collagenase digestion (PMID: 35761084). Nonetheless, we chose not to merge our BC and liver datasets for reasons that the Reviewer raises, despite this being a common approach in the literature. Further, the predominant transcriptional differences that we assessed in this study are from cells isolated from the same tissue (e.g. T_{RM} vs T_{EX} from the same tumour) and are thus controlled for the respective processing strategies for the respective tissues and tumours. Further, the consistent similarities in gene expression across T cell populations in both datasets (as well as published datasets) strongly suggest that the observed gene expression differences are primarily driven by underlying biological factors rather than any technical variability. We will clarify the justification for our chosen methodology and reagents including catalog numbers in the revised version of our manuscript.

2) The numbers in Fig 4b are very similar. Are the differences statistically significant (e.g. 0.77 vs 0.73)? Are they biologically significant?

The Reviewer points out that the clonal distinction between T_{RM} vs T_{EX} vs T_{EM} are very similar and indeed this is a point we would like to make. Importantly, we would like to point out that T_{RM} , T_{EX} , and T_{EM} are largely distinct populations with respect to their TCR repertoires. We can show from our BC dataset that the difference between the T_{RM}/T_{EX} sharing vs the T_{EM}/T_{EX} sharing is statistically significant. We will include this additional statistical testing for Fig. 4b and discuss the biological implications of this data in the revised manuscript.

3) Fig 4c: What are the percentages of shared clones between TEM and TEX and between TEM and TRM?

The percentage of shared clones between T_{EM} and T_{EX} is 24.7%, and between T_{EM} and T_{RM} is 24.6%. To better illustrate these comparisons in Fig. 4c we will reproduce the Venn diagram to highlight all of these comparisons.

4) Fig 4d: The cell numbers of clonal overlap in the three pairs look similar. There is no statistics. It seems that there are limited shared clones among all cell types. Is this correct? The authors should show distribution of clonal size within each subset.

The Reviewer is correct that there are limited shared clones among all cell types, and this is indeed the point of Fig. 4d. We will add the requested data that shows the total clone size distribution within each subset and ensure that this correct interpretation of the reviewer is explicitly stated in the text. This Figure also accentuates the point that when there is clonal sharing between subsets, there is unequal sharing. As presented, when we increase the minimum number of cells (from 1 to 2) that are required to be present in each cluster for a clone to be considered 'shared' between clusters, over half of the 'shared' clones are eliminated.

5) Fig 4h: Any example of the viruses predicted to be recognized? Is it possible to analyze Trm vs Tex phenotype in EBV-specific or CMV specific T cells?

The Reviewer makes an excellent point. We have detected CMV, EBV, influenza, and SARS-CoV2-specific T cells in these datasets and will present the specificity and phenotype of these cells in the revised manuscript.

6) Do Trm and Tex have different localization within the tumor?

This is a key and important point which has not yet been addressed in the literature and that we are currently pursuing using several innovative approaches. In human tumours, we hypothesise that T_{RM} and T_{EX} coexist in the same niches, and that TCR specificity rather than location dictates the phenotypic distinction of these cell types. We have also identified various proteins that can discriminate T_{RM} and T_{EX} in human BC, and we will assess the ability of these proteins to dissect these cell types *in situ* using the following high-dimensional immunofluorescence microscopy modalities.

1. We have imaged 38 parameters on slides from 5 BC patients using the CycIF technique (PMID: 27925668, 38360200) in collaboration with the developers of this technique (Sorger Lab, Harvard Medical School). This has been complemented with 21-plex imaging of 5 additional human BC slides using the MACsima immunofluorescence imaging platform (PMID: 35115587). We are currently building computational algorithms to annotate and determine whether T_{RM} and T_{EX} are differentially located.
2. Using our murine BC model (AT3-OVA) and experimental plans as conducted in other experiments per Fig. 5b and 5g, we will image the relative location of tumour-specific T_{EX} and T_{RM}, and bystander T_{RM} using a 21-parameter MACsima platform, and confocal imaging. In this model, OVA-expressing tumour cells are marked by expression of GFP, therefore, we can assess where T_{RM} and T_{EX} are located with respect to tumour antigen.

Together, these approaches will generate extensive new data to determine whether T_{RM} and T_{EX} exhibit distinct localisation in tumours, which will represent a significant advance for the field and address the Reviewer's pertinent question.

7) Do bystander effector/memory T cells preferentially become Trm in tumors compared to other non-lymphoid organs? If so, why? What is the signal that attract bystanders to the tumor?

The Reviewer raises an interesting point regarding how effector/circulating memory T cells enter tumours. We and others have shown that inflammation is sufficient to recruit T cells into non-lymphoid tissues, regardless of TCR specificity (PMID: 22509047; 15485630). Tumours are inflammatory environments and likely recruit T cells through similar means. Specifically, CXCR6 and CXCR3 have been implicated in T cell entry into tissues (PMID: 24162776, 34343496). Thus, we hypothesise that bystander T cells require CXCR3 and CXCR6 signals to enter AT3-OVA BC tumours. To test this, we will CRISPR/Cas9-edit CXCR3 or CXCR6 from bystander T cells, transfer to AT3-OVA bearing mice, and determine the ability of these cells to enter the TME.

8) Line 300: "These results show that tumour-derived TPROG, TEX, and TRM cells were more similar to each other and clonally distinct from CD69- and spleen-derived populations." It seems to contradict Fig 4 where the authors concluded that Trm and Tex are clonally distinct.

This is an important distinction that we will clarify in the text. Clonal populations in the human data are designated as such by shared TCR expression. In contrast, all T cells in the SPLINTR barcoding experiment express the same TCR (OT-I transgenic TCR), and in this context, a single clone indicates an OT-I T cell with a unique barcode. These two points are not meant to be contradictory, and we apologise for the confusion. The key points are as follows:

1. T_{RM} and T_{EX} from human tumours largely have different TCRs. However, this is not an absolute as some sharing (although limited) does occur.
2. We hypothesise that T_{EX} isolated from tumours are predominantly tumour-specific (and express CD39 which has been shown to demarcate tumour-specific T cells (PMID: 29769722)). We also show that virus-specific cells in tumours can adopt a T_{RM} phenotype. However, we do not assume that there are no tumour-specific T_{RM} within tumour samples.
3. In fact, in murine tumours, we identified tumour-specific TRM-phenotype cells. Therefore, we aimed to investigate whether these cells differentiate from the same or distinct type of effector cells as T_{EX} cells.
4. Thus, the interpretation of the SPLINTR lineage barcoding is that a tumour-specific effector cell can give rise to either a T_{RM} or T_{EX} cell upon entry into tumours. We will expand upon these experiments in our revised manuscript in response to comments from Reviewer 2.

Minor concerns

1) Please provide statistics in Fig 1c.

This was fairly pointed out to us by both Reviewer's 1 and 3, and we will add additional analysis of this dataset and add relevant statistics to alleviate these concerns.

2) Line 197: "CD94, CD101, CD73, and CD161 were specifically enriched in TRM cells in BC but not liver-derived tumours." Why are the markers of Trm are different in BC vs HCC?

The reason for the differential protein expression across various tumours is not unexpected, as T_{RM} cells in different tissues typically exhibit distinct protein expression profiles (PMID: 37392736). This is a key point that we will accentuate in the text – that individual genes/proteins should not be blindly used to identify populations of T cells across different contexts.

3) What are the leading-edge genes in Fig 3c?

A selection of these genes is presented in Extended Data Fig. 5, and we will provide the full gene list in a supplementary spreadsheet.

Referee #2:

This study by MacKay and colleagues examines tissue resident CD8⁺ memory T cells (TRM) relative to functionally exhausted CD8⁺memory T cells (TEX) in tumors and tumor models, identifying distinguishing features of these cell types. TRM are a memory subset distinguished by their long-term residence in diverse tissues, with predominant populations in mucosal and barrier sites. TRM express a suite of surface markers and a distinct transcriptional profile that enables them to mobilize in tissues, interact with tissue cells and structures (e.g., epithelial cells, collagen), but also mediate the functional responses typical of CD8⁺ T cell memory, including rapid production of effector cytokines such as IFN-gamma, TNF-alpha, and cytolytic mediators (granzymes, perforin). In the context of protective immunity, TRM have been shown in numerous infection models by this group and many others, to provide site-specific protective responses through TRM-derived functions, recruitment of other immune cells to the tissue, and other mechanisms. In this context, the comparison and demonstration that TRM are not functionally exhausted is obvious. However, in human cancers and mouse tumor models, TRM are associated with the tumor and it has also been shown extensively that TEX are among tumor-infiltrating T cells and are targets for checkpoint blockade. Interestingly, TRM express multiple markers that are also expressed by and associated with TEX particularly in chronic infection such as PD-1, CD101, CTLA4, TIGIT, and others, and for this reason, there has been confusion in the field of whether the target population for checkpoint blockade is TRM or TEX (or both).

The authors endeavor to resolve any confusion on what constitutes a resident or exhausted T cell by comparing transcriptional profiles of TRM and TEX from published datasets of mouse LCMV infection models, mouse tumor models, and in a scRNAseq dataset of T cells from breast tumors (BC) and normal breast tissue (there is a previously published scRNAseq dataset from this group in 2018 where they identified TRM as part of BC and also found them in normal breast tissues but the BC data here appear to be new). They found that human BC contain TRM like in normal breast tissue but that a fraction of the TRM also exhibited a TEX signature marked by higher expression of the transcription factor Tox (shown in many studies from a few years ago to be enhanced in TEX cells and to drive a TEX functional program), the memory-associated transcription factor EOMES, and functional markers like GRMK, associated with CD8 T cells in aging. They then show that the proportion of TEX in the tumor was associated with reduced survival using data from a cancer dataset (like TCGA but called METABRIC) and increased TEX correlated to better responses to Immune checkpoint blockade (ICB) therapy; these data appear to be confirmatory as TEX has long been associated

with hyporesponsiveness to tumors and is a target of ICB therapy. The authors then investigate the origin of these TEX cells with a TRM phenotype in clonal analysis from the scRNAseq of the BC dataset and using different mouse models to investigate the antigen specificity and origin of the TEX that get into the tumors and take up residence.

Overall, the experiments are well done, text is succinct and clearly explained, and the findings are of interest. However, the study is mixed in terms of novelty—the first part (Figs. 1-3) largely repeats or repurposes data already published on TRM and TEX, by comparing them to each other and then revealing similar bottom line aspects that have already been reported on TRM and TEX function and associations with cancer survival and ICB responses.

We thank the Reviewer for the summary of our study and agree with most of the points raised. However, we respectfully assert that the specific question we aimed to address – i.e. the direct comparison of T_{RM} and T_{EX} in human tumours - has not been addressed in the literature. Further, we believe that the continued conflation of these cell types within the literature (including by our group) makes our study timely and important.

Regarding our BC study with Sherene Loi's Lab in 2018, this study solely relied on CD103 expression on tumour-infiltrating CD8⁺ T cells (absent of healthy tissue-derived cells) to designate these cells as " T_{RM} ". Given the advancement of our understanding of tissue-resident memory and exhausted T cells in the following years, and as shown in this study, this designation led to the conflation between bona fide T_{RM} , and exhausted T cells, an error that was subsequently repeated by many different groups. Therefore, it was unclear whether the survival associations were due to association with T_{RM} , T_{EX} , or a combination of the two. Further, various studies have associated ' T_{RM} ', based on poorly designed gene signatures or ubiquitously expressed cell surface proteins, with positive responses to ICB, which we again assess as an incorrect statement that needs to be corrected in the literature.

To do this, we took great care in carefully designing new experiments in which we could carefully pair tumour-derived and healthy tissue-derived CD8⁺ T cells from both breast cancer/breast tissue and liver tumours/liver tissue and where the tissue samples were treated consistently. This was essential to have confidence in our annotation of these cell types. We then utilised our novel approach to disentangle T_{RM} and T_{EX} within tumours and show that these cell types can be identified to varying extents across many different cancer types. We dispute that we repurposed published datasets, and that if we had, we could not have had confidence in our cell-type annotations. Thus, we are confident that our current study, for the first time, has reconciled distinct populations of CD103⁺ T cells in human tumours (namely T_{RM} and T_{EX}), associating the presence of these cell types with distinct roles in tumour control.

Another concern I have with the first part is the juxtaposition of migration and functional capacity as being features of distinct subsets, when they can and do co-exist in the same cells; tissue residence is a feature of homing and localization and doesn't necessary reflect a polarized function--this group and others have shown that TRM can mediate different types of functional outputs depending on site, stimulation conditions, and even metabolic pathways.

The points raised here by the Reviewer are critically important to our study. We agree with the Reviewer and, in the manuscript, argue that both T_{RM} and T_{EX} have similar migration capabilities. However, T_{RM} maintained heightened functionality compared to T_{EX} within the *same* microenvironment. Further, T_{RM} and T_{EX} have distinct developmental requirements despite residing in the same location. We agree that T_{RM} across different tissues have distinct

functionalities, however, our functional analyses are performed on T_{RM} and T_{EX} cells isolated from the same tissue, and we have not attempted to assess the difference in function across tumours for this reason.

The latter part of the study (Figures 5 and 6) I found to be more interesting and original; these experiments begin to address the mechanisms for the distinct functional capacity of tissue homing T cells that take up residence in tumors. While they presented compelling results, they weren't fully developed. See below for specific points.

We thank the Reviewer for their enthusiasm in our mechanistic experiments and will expand upon these experiments in our revised manuscript. Specific experiments are outlined below.

1. Many of the results and analysis in the first three figures address similar questions and comparisons as done by Gavil, et al, 2023. The lead author of the present manuscript also recently published a comparison of epigenetic profiles between TRM and TEX in Immunity, revealing gene expression differences between TRM and TEX similar to those shown here in Figs. 1 and 2.

The study by Gavil et al was highly important for the field, showing the non-recirculating nature of exhausted T cells in murine tumours, and that residence of tumour-specific T cells correlated with high expression of exhaustion-related molecules such as Tim3. The study from our group in Immunity compared T_{RM} across different organs to T_{EX} from the spleen of chronic LCMV-infected mice. Indeed, both of these studies point to overlapping features shared by T_{RM} and T_{EX} in mice, accentuating the need and impetus for our current study.

Neither of these studies proved or disproved the possible presence or absence of bona fide T_{RM} in human tumours. Given the similarities in T_{RM} and T_{EX} in the aforementioned studies, and several others in the literature, it was possible that bona fide T_{RM} were excluded from tumours, and that the 'tumour-associated T_{RM} ' that have been described were, in fact, wholly a population of exhausted T cells. Our approach enabled the disentanglement of these cell types, and the attribution of distinct contributions to tumour control to these different cell types. Further, the assumption that cell surface proteins and gene signatures that are used to identify T_{RM}/T_{EX} in mice can be directly applied to the same cell types in humans is an oversight which has led to many of the issues in the field to date (e.g. utilising CD103 expression to identify tumour-infiltrating T_{RM} in humans), further necessitating and differentiating our study from both Gavil et al, and the Immunity paper from our own group.

The genes upregulated in TEX versus TRM such as GZMK and lower PRF1, etc, are similar to those genes upregulated in CD8+T cells associated with aging (PMID 33271118). It may be interesting to compare the TEX-specific genes to this aging signature.

We have compared our signatures to the aging-specific genes from this dataset and the Reviewer is correct that there is some overlap between our BC T_{EX} gene signature and the aging signature from this study, and we can add this data at the Editor's request.

2. The description and conclusion of the TCR clonal analysis in Fig. 4 is not consistent with the results. The authors start out by stating that there is very little clonal overlap between TRM and TEX in the BC dataset in part (a) and (b). However, the Venn diagram in part (c) shows considerable clone sharing between TEX and TRM; out of 263 TEX clones, 57 of them or 21.6% overlap with TRM clones. Moreover, there appears to be even more overlap of TEX

clones with TEM cells in BC—presumably those that derived from tumor-infiltrating T cells. It would be informative to show the derivation of the expanded TEX clones as overlapping with TRM or TEM; these data are important because they indicate that TEX is a functional state due to the environment and antigen that can arise from cells with different migration abilities. The role of tissue and environment in driving these different functional programs was shown in the Gavil, 2023 study.

We apologise for the confusion in describing these data. This point was also picked up by Reviewer 1, and we will expand the analysis to include the sharing between T_{RM} , T_{EM} , and T_{EX} . There are indeed clones shared between T_{RM} and T_{EX} , however, our major point here was to show that this sharing is limited (largely driven by 1 shared cell). However, we will amend this figure and the associated text to address this important point.

3. The mouse experiments with different TCR specificities and fate mapping in Figures 5 and 6 show interesting findings. That bystander (non-tumor reactive) T cells migrate more extensively to the tumor compared to antigen reactive ones and adopt a TRM signature is very interesting. Why do the non-tumor reactive cells migrate to the tumor so extensively? Are they also in the spleen or other organs and is their migration and TRM phenotypes due to TGF-beta? How does this finding relate to low affinity also promoting TRM, and other studies that show that antigen drives T cells into tissues? Interestingly, the results are similar to previous findings that chronic antigen stimulation reduced TRM maintenance (PMID 22504644).

We do not find that bystander T cells migrate more extensively into tumours compared to antigen-reactive cells. Instead, we show in Figure 5, that when bystander T cells do enter the tumour, they are more likely to adopt a T_{RM} phenotype compared to tumour-specific cells. These bystander T cells are also found in the spleen and other organs. We will amend the text to clarify this point in the revised manuscript.

The Reviewer also asks why bystander T cells migrate to tumours and whether they are present in other organs. We and others have shown that effector and circulating memory T cells can enter inflamed tissues independently from antigen recognition (e.g. PMID: 22509047). Indeed, it is highly likely that inflammation within the TME drives the recruitment of effector/memory cells into this site. To address this possibility (in line with a similar comment from Reviewer 1), we will assess the requirement of CXCR3 and CXCR6 for bystander T cells to enter AT3-OVA tumours.

We also agree with the Reviewer's point regarding low affinity T cells preferentially becoming T_{RM} and have shown this in Figure 5. This more likely reflects the developmental capacity of low-affinity T cells to adopt the T_{RM} cell fate rather than a difference in capacity of these cells to enter tumours, given the ability of tumour-agnostic T cells to enter the tumours.

We hypothesise that the phenotype of bystander T_{RM} in the tumour is likely due to TGF β as the Reviewer mentions, and as is the case for tumour-specific cells (PMID: 36827978). We will assess this directly by CRISPR/Cas9-mediated excision of TGF β RII on bystander T cells and determine the requirement for TGF β -signalling on the development of bystander T_{RM} in AT3-OVA tumours.

4. The fate mapping system is beautiful but I found the second part using re-transfer into tumor bearing hosts to be confusing. How durable is the initial fate of the cells in same host if they wait longer?

We apologise for the confusion in the description of the retransfer experiments. We will amend the text to add clarity to this point. Regarding the durability of the T_{RM} and T_{EX} cell phenotypes in AT3-OVA tumours, we are bound by ethical constraints that limit the size of the tumours ($<1000\text{mm}^3$) that we can maintain. Thus, these phenotypes can only be assessed for as long as two weeks post-transfer. In anticipation of this question, we have staggered the retransfer of T cells (i.e. cells are transferred from tumours at $\sim d28$ into tumours at $d10$), which allows for a greater period in which to monitor the stability of these phenotypes. However, to expand upon this point further, we will perform co-transfer experiments with the following alterations:

1. Congenically distinct T_{EX} - and T_{RM} - phenotype OT-I T cells will be sorted from primary AT3-OVA tumours and co-transferred into secondary mice bearing tumours. This will enable the direct competitive assessment of the stability of these phenotypes.
2. Sorted OT-I T cells will be directly transferred into secondary tumours (rather than intravenously as done previously). Since both T_{RM} and T_{EX} downregulate the transcriptional machinery expected to be required for trafficking through the blood and lymph, these cells may fail to efficiently migrate to the tumours when transferred via the IV route. In our initial experiments, direct intratumoural transfer of sorted T_{RM} and T_{EX} resulted in a significant increase in recovered cell numbers. This will further allow a more robust interrogation of the phenotype stability of these cell types.
3. We will also transfer these cells into mice bearing both AT3-OVA and AT3 parental (no-OVA antigen) tumours. This will enable us to further determine the role of cognate antigen in the stability of the T_{RM} and T_{EX} phenotypes over time.

Additionally, given the Reviewer's enthusiasm for our fate mapping experiments, we propose new experiments to build upon this aspect of the manuscript. While our prior experiments (Figure 6) revealed that effector T cells of the same clonal origin could give rise to both T_{RM} and T_{EX} populations in tumours, this result did not preclude the possibility that naïve T cell clones may exhibit a predetermined bias for T_{RM} vs T_{EX} formation. Since our prior submission, we have developed the capacity to SPLINTR barcode naïve T cells via intra-thymic transfer of SPLINTR-barcoded T cell progenitors. This enables us to assess lineage commitment fate decisions very early following initial T cell stimulation. To this end, in the revised manuscript, we propose to utilise naïve-barcoded OT-I T cells to determine whether T_{RM} vs. T_{EX} fates may be dictated at the initial priming stage of T cell activation.

Referee #3:

Burn et al. explore the relationship and potential roles of Trm and Tex TILs. The authors nicely illustrate the transcriptional differences between various TILs in multiple mouse and human disease contexts, demonstrating the limitations of previously published Trm gene signatures. While published Tex gene signatures reliably identify Tex cells to the exclusion of Trm cells, some published Trm cell gene signatures are non-specific and fail to distinguish Trm and Tex cells – highlighting a danger in using poorly validated gene signatures as a proxy for specific biological properties. Burn et al. re-derive Trm and Tex gene signatures. In other words, certain correlates are used to define Trm and different correlates are used to define Tex, and then as would logically follow, Trm and Tex are shown to be different based on these correlates. These new gene signatures correlate differently with overall survival in cancer patients, reconfirming in some scenarios that TRM-associated signatures correlate with survival. Notably, the author's Tex gene signature, but not the Trm gene signature, is a moderate predictor of ICB response in a cohort of breast cancer patients. Comparing clonal overlap between Trm and Tex cells, the authors suggest that Trm cells in tumors that do not exhibit transcriptional signatures of

chronic-stimulation (a.k.a. Tex signatures) are likely bystanders in the TME. Interestingly, the authors investigate the ontogeny and differentiation potential of CD103⁺ Tex cells. They provide evidence that reduced antigen-load and/or TCR avidity correlates with tumor-specific T cell expression of CD103. They also demonstrate that CD103⁺ tumor-specific T cells have greater plasticity than CD103-CD39⁺ Tex-term cells. The paper brings together sophisticated analyses to diverse data sets, which is certainly impressive. However, some issues diminished enthusiasm. Showing that Trm and Tex express different gene signatures was an inevitable result from the authors' analytical approach. While one could argue that the study provides an improved signature that could distinguish what could be labeled Trm vs. Tex, it was not clear what the underlying biological definition of these terms were, which limited its value. In addition, the abstract concludes that "These results reveal the unique roles of Trm and Tex cells in tumor control, underscoring the need for distinct strategies to harness T cell populations in novel cancer therapies," but the role of Trm was not revealed, nor why the strong Trm correlation with survival exists, and what (if anything) about Trm should be harnessed in novel cancer therapies.

We thank the Reviewer for the positive summary of our study, with the major critique being that we did not show what T_{RM} do, and specifically why the presence of T_{RM} is associated with the long-term survival of BC patients. Understanding the role of T_{RM} in tumours is difficult, especially in humans, and the association of T_{RM} with long-term patient survival could be a combination of various, non-mutually exclusive mechanisms (detailed below) that will be difficult to tease apart. However, we will attempt to resolve some aspects of this in our revised manuscript, with specific experiments and analyses detailed in our response below. We will also point to specific examples in the literature that support our stated hypotheses below. We envisage two likely reasons for the association of T_{RM} gene expression with positive survival outcomes, both of which rely on the knowledge that the datasets (including METABRIC) are developed from the sequencing of tumours that have been surgically removed to deliver a curative outcome and are not biopsies of tumours that are then measured long-term. This has the important implication that some excised tumours may contain more healthy/non-cancerous tissue regions than others. Therefore, the association of T_{RM} with long-term survival may be a result of the following non-mutually exclusive mechanisms.

1. The development of T_{RM} is associated with the effective clearance of tumour antigen. Tumour-specific T_{RM} then reside long-term in the same tissue which may suppress tumour outgrowth from residual cancer cells following tumour resection via mechanisms we have previously described (PMIDs: 30598548, 36827978).
2. T_{RM} develop in tissues following the resolution of inflammation. The development of T_{RM} and high expression of the T_{RM} gene signature may signify the nearing resolution of tumour growth in the excised tissue. Thus, resected tumours that contain greater regions of non-cancerous tissue may signify effective tumour control that correlates with the positive long-term prognoses of these patients. This is a point we will articulate more clearly in the discussion, and which has been made by others in recent reviews (PMID: 39142275, 37821656).

To address these points experimentally, we will assess whether T_{RM} maintain local tumour control following tumour regression. Specifically, we will inoculate mice with AT3-OVA, then 8-10 days post tumour inoculation (when tumours are palpable [$\sim 100\text{mm}^3$]), we will transfer effector OT-I T cells and treat with anti-PD-1/CTLA4, which results in tumour regression. To determine the role of T_{RM} specifically in maintaining local control over tumours, we will utilise in vivo depletion strategies to ablate either all memory CD8⁺ T cells (anti-CXCR3) or circulating memory T cells (anti-Gr1), and determine whether tumours recur following these

respective depletion approaches. We will also include mice treated with FTY720 to block recirculation and involvement of newly recruited T cells. This will allow us to determine whether T_{RM} elicit in situ control against tumours.

Further, we hypothesise that tumour-specific T_{RM} are present in human tissues following complete tumour regression. Thus, we will utilise publicly available datasets (including PMID: 37001526) where patients have been treated with ICB regimens, determine TCRs that are expanded in blood following ICB which are likely to correlate with tumour-specific effector T cells, and determine the phenotype of these cells per our T_{EX} and T_{RM} gene signatures in biopsied tissue where tumours have regressed. We expect that in non-cancerous tissue, we will identify inferred tumour-specific $CD8^+$ T cells defined as T_{RM} using our novel gene signature.

Specific comments:

The authors should very clearly state their exact definitions of Trm and Tex. E.g., were Trm defined as a memory T cell type based primarily on markers or some universal signature (e.g. line 91), or is Trm meant to signify a non-circulating behaviour of a (memory?) T cell? Were published Trm gene signatures based on resting memory cells that were defined by migration properties?

We will amend the text to make this clearer. Our conceptual definition of T_{RM} and T_{EX} is that both T_{RM} and T_{EX} are non-recirculating T cell populations and express or downregulate genes leading to tissue retention. However, T_{RM} are distinct from T_{EX} by the definition that they are memory T cells not currently activated being activated by cognate antigen. In practice, specifically defining these cell types in human tumours by these conceptual notions is impractical. Therefore, we made the assumption that both T_{RM} and T_{EX} share expression of T_{RM} signature genes from studies in both mice and humans. The T_{RM} gene signatures from mice were from phenotypically defined T_{RM} that were experimentally shown to be resident in prior experiments, either through parabiosis (liver T_{RM}) or transplantation (skin T_{RM}) (PMID: 27102484).

It is hard to know if the gene lists presented would be useful and accurate outside of the relatively few models used. Indeed, by figure 5, when the group uses their newly defined Trm and Tex gene signatures (defined in human breast cancer) in the mouse B16 melanoma model, the signatures are helpful on average but look to be imprecise on a sizeable fraction of cells.

The Reviewer makes an excellent point regarding caution upon using signatures to identify different populations of cells in scRNAseq datasets, as the signatures are only as useful as the populations of cells used to define them in the initial setting. This is indeed the reason for the incorrect annotation of T_{RM} and T_{EX} in several published datasets. We will add additional discussion in the revised manuscript to reinforce this point. However, we respectfully disagree with the interpretation that our signatures are imprecise. Indeed, we can segregate distinct populations of T_{RM} and T_{EX} from several distinct human tumours from the Pan-cancer T cell atlas (PMID: 34914499). In the revised manuscript, we will expand upon this point and include further analyses from additional datasets, specifically to include skin and lung cancers that are predicted to have high frequencies of T_{EX} given the relatively high tumour mutational burden of these cancers.

Finally, the variable expression of signatures across species is to be expected, as it is well understood that individual genes are differentially regulated across mice and humans, including often utilised genes for T_{RM} identification such as *ZNF683* (HOBIT) and *ITGAE* (CD103). However, this comparison examines the relative enrichment of these signatures as a composite across bystander and antigen-specific cells within the murine melanoma tumours, suggesting that TCR signalling in tumours partially drives the distinction between T_{RM} and T_{EX} cells.

The authors miscommunicated the central messages of REF 16. This study showed that resident cells (as defined by migration property) 1) exist in tumors, but that 2) conventional markers of residence did NOT correlate well with migration properties in tumors, particularly when assessing tumor-specific T cells. No causality between Trm gene signatures and residence within tumors was shown.

We agree with the Reviewer's interpretation of this important study and apologise for the incorrect statement. In the revised manuscript, we will qualify our statement to reflect that it is still unknown whether T_{RM} and T_{EX} utilise the same mechanisms of residency in tumours, despite sharing expression patterns of various genes involved in T cell residency.

Figure 1a: Please reconsider the phrasing of this sentence, which uses "Figure 1a" as the sentence subject instead of the analysis represented in the figure.

We will adjust this in the text as suggested.

Figure 1c + 1f: No statistical test is presented to demonstrate significant TRM gene score enrichment. All that is demonstrated is that >50% of cells score above 0 for TEX and virus-specific cells in MFP/Tumour. Please consider including Ext Fig 1a in the main figure or adding the TEX-term cells as a separate population in the Fig 1c violin plot. This would clarify the bimodal distribution seen in CL13-D30 and clarify the authors' conclusions.

This is an excellent suggestion, and we will adjust the analysis and presentation of these results.

The authors have recently proposed that CD103 expression defined a subset of Trm that was less functional than other subsets of Trm (Christo et al., NI 2021). Burn et al., used CD103 expression as a criteria for calling a cell a Trm. Where do CD103- Trm fit into the larger picture? Because Fig 2 shows cytokine production from CD103+ Trm, how best to reconcile these results with the Christo study?

This is an interesting point. In the study as presented, we have compared two populations of CD103⁺ T cells that we define as either T_{EX} or T_{RM} from human tumours, thus, this does not conflict with the Christo study referenced by the Reviewer. Within the breast cancer dataset, there are no clear populations of CD69⁺CD103⁻ T_{RM} as most of these cells express genes associated with non-resident cells, including *KLF2*, suggesting these are less likely to be bona fide T_{RM} . However, we do see clearer populations of CD69⁺CD103⁻ T_{RM} within the liver dataset, thus, we will perform restimulation experiments on human liver-derived T_{RM} to address the reviewer's comment.

Figure 2b-c: Please report the sensitivity and specificity of each newly derived signature to convey their utility in reliably identifying each cell-type/subset. While the box plots convey some information, it is clear that some cells still exhibit enrichment of genes from alternative subsets (e.g. TRM that express TEX gene signature and vice versa)

We will include sensitivity and specificity analyses for the requested figure in the revised manuscript.

Ext Fig 2a-c: no in text reference to this extended data

We apologise for this oversight and will address this in the revised manuscript.

Fig 2j: Please provide a more substantial explanation of the ROC analysis to improve interpretability for the reader. What is the source data? What is the prognostic predictor/subset used to represent TRM-signature and TEX-signature? What constitutes treatment response? For such a critical point in the paper, the provided explanation of the analysis is scant.

A fair critique that we will address in the revised manuscript. This analysis is similar to that performed in our previously published study in which this data was also analysed, and we will better describe this data in our current manuscript in line with this (PMID: 36827978).

Figure 3: The clustering methodology appears to outperform the use of gene signatures for identifying cell populations. What is the advantage or value of gene signatures when clustering reliably enables cell annotation? Also, is this not a circular approach to defining T cell subsets? Are there better methods to cross-validated gene signatures as a means to determine cell identity?

We will better explain our approach for annotating these cells in the methods, but we have essentially done as the reviewer has suggested. The pan-cancer dataset is clustered without regard for signature expression, then clusters are assigned a cell-type label based on the enrichment of defined signatures. We would argue that this is a standard approach for defining cell types in scRNAseq and apologise for the confusion that we have caused on this point.

Figure 5: The use of “Tex-like” and “Trm-like” TEX cells is confusing and seems contradictory to the overarching conclusions of figures 1-4 that emphasize the difference between TRM and TEX cells. It appears the only distinction in this figure (and figure 6) is to study the differences between CD103+ and CD103- TEX cells.

Thank you for pointing out the confusion here, and we will attempt to clarify the respective populations and comparisons of each in the revised manuscript.

Figure 5q-r: While the mean gene signature enrichment scores appear significantly different, the significant overlap in enrichment scores between Ag-specific and bystander scores warrants acknowledgment and discussion. This overlap suggests that these gene signatures have limited sensitivity and specificity.

This is a good point which we will carefully clarify. As we described above, it is to be expected that there is not a perfect alignment of signatures across mouse and human. However, we do believe the interpretation is valid that overall, our T_{EX} and T_{RM} signatures from BC are segregated at least in part by the respective presence or absence of intratumoural cognate antigen, which is supportive of the rest of the data in this figure.

Figure 5o-r: Clarify figure legend description to describe time between cell transfer and

isolation for scRNA-seq. Transfer 14-20 days after tumor cell inoculation and then isolation 14-20 days after cell transfer?

We apologise for this lack of clarity. The timing of cell transfer and tumour growth is a feature of the epicutaneous model of B16 melanoma, whereby tumours begin growth within a wide timeframe, and some tumours are controlled permanently by endogenous skin T_{RM} (PMID: 305985548). We will amend the methods section to better describe this system. Briefly, T cell transfer occurs following observation of tumour growth, and mice are then culled at a later timepoint when tumours reached their ethical endpoint (1000mm³). Together, this accounts for the range of the timeframes reported.

Figure 6 is confusing with the use of the ‘TRM’ label, which appears to be a stand in for CD103+ TEX cells.

We will better explain and define these populations in the revised manuscript.

Discussion: The authors state that “Our study for the first time reconciles the respective roles of two critical T cell subsets associated with tumour control, namely CD8+ TRM and TEX cells.” This seems like an overstatement. This study also does not really show what Trm do.

As described in our response above, we will perform multiple experiments to address this pertinent question in the field.

Point by Point Response for “Antigen reactivity defines tissue-resident memory and exhausted T cells in tumours” NATURE 2024-03-04823A (NI-A39737A)

We thank the Reviewers for their thoughtful and constructive feedback. In response, we have transferred our manuscript to *Nature Immunology* and incorporated substantial new data addressing the points raised. These revisions span **Figures 2-6** and **Extended Data Figures 2, 3, 6-9**, and collectively clarify the relationships between T cell exhaustion, memory and tissue residency within the tumour microenvironment. Below, we summarize the major additions and key findings from these new experiments.

1. Spatial localisation of T_{RM} and T_{EX} in human tumours.

To assess the spatial distribution of T_{RM} and T_{EX} cells within the tumour microenvironment, we performed high-dimensional cyclic immunofluorescence (CycIF) imaging on human breast cancer (BC) samples. Using newly identified markers to distinguish BC-associated T_{RM} and T_{EX}, we quantified their proximity to PanCK⁺ tumour cells. While both populations were broadly distributed throughout the tumour tissue, T_{EX}-like cells showed a modest but statistically significant enrichment adjacent to PanCK⁺ regions. This finding suggests that T_{EX} differentiation may be influenced by tumour cell proximity, consistent with a role for ongoing antigen recognition in shaping T_{EX} cell fate.

2. Lineage origins of T_{RM} and T_{EX} cells

To determine whether T_{RM} and T_{EX} cells arise from distinct naïve precursors, we extended our SPLINTR lineage tracing experiments by transferring barcoded naïve CD8⁺ T cells into tumour-bearing mice. These data show that both T_{RM} or T_{EX} populations arise from shared naïve progenitors, with no evidence of lineage pre-determination. Barcode sequencing analyses further show that intratumoural T cells, regardless of fate, are more similar to one another than to peripheral CD69⁻ or splenic T cells, suggesting that local environmental cues, rather than intrinsic lineage bias, drive the divergence of T_{RM} and T_{EX} fates.

3. Mechanistic insights into T_{RM}/T_{EX} cell development and function

We performed a comprehensive set of new experiments (Figures 4-6; Extended Data Figures 6-9) to further dissect the developmental and functional distinctions between T_{RM} and T_{EX} cells, including:

- Extensive phenotypic profiling of antigen-specific and bystander TIL populations
- Functional assays highlighting differences in cytokine production
- Indirect residency assays to evaluate migratory capacity
- Correlation of T_{RM} and T_{EX} fate with tumour antigen load
- Inducible TCR deletion to test the role of TCR signalling in T_{RM} development
- Investigated the role of Tcf1 in shaping intratumoural T cell fate
- Examined TGFβ signalling in the development of bystander T_{RM} cells
- Tested the contribution of CXCR6 and CXCR3 to bystander T cell infiltration
- Assessed stability of T_{RM} and T_{EX} phenotype following retransfer into tumours in the presence or absence of cognate antigen.

We also provide further human TCR sequencing analyses, which reveal that tumour-derived T_{EX} cells frequently share TCRs with T_{RM} found in adjacent, non-tumour tissue. This finding suggests that tumour-specific T_{RM} may persist in the surrounding tissue post-resection and potentially contribute to long-term local tumour control.

Together, these data strengthen our central conclusion: that T_{RM} and T_{EX} cells comprise transcriptionally and functionally distinct co-resident populations within tumours. By integrating spatial, clonal, and mechanistic analyses, our revised manuscript provides a more nuanced framework for understanding how tissue residency and antigen experience shape the intratumoural CD8⁺ T cell landscape. We believe these additions fully address the Reviewers' concerns and substantially enhance the clarity and impact of our study for publication in *Nature Immunology*.

Referee #1:

In “Antigen reactivity defines tissue-resident memory and exhausted T cells in tumours”, the authors investigated the difference between Trm and Tex. They showed that Tex and Trm have different gene expression and clonality. Trm signature in human cancer is associated with better survival but not response to ICB. Computational analysis suggested that Trm in human tumours mainly consist of bystanders. The authors further validated their in silico analysis in mouse tumor models where low affinity clones and bystanders preferentially differentiate into Trm rather than Tex. Overall, the study is interesting and addresses an important knowledge gap.

Major concerns

- 1) Methods: Was lymphocyte enrichment by Percoll gradient also done with BC samples? If not, why? Catalog number of freezing medium should be provided in the Methods. Why does liver require a different freezing medium? How was the difference between frozen samples and fresh samples controlled during analysis?

The Reviewer makes an important point regarding sample control, and indeed, we took great care to treat all our samples consistently for analyses where we performed direct comparisons. Percoll enrichment was not performed on BC samples, as these preparations yielded relatively clean single-cell suspensions with minimal debris, making additional enrichment unnecessary. In contrast, liver-derived lymphocytes are more prone to cell death during processing; thus, we used CryoStor CS10 (StemCell Technologies, #07930) to improve post-thaw viability. This has now been clarified in the revised Methods, including catalog numbers for all freezing media.

We were careful to control for potential processing effects. All flow cytometry analyses were performed on fresh samples (**Fig. 1g-h; Extended Data Fig. 1b**), while all single-cell RNA-seq samples were cryopreserved and processed in parallel within a single sequencing run for each experiment. To ensure consistency across tissues, liver samples were digested using the same collagenase protocol as BC samples, thereby controlling for digestion-associated gene expression changes (PMID: 35761084).

Importantly, our analyses focus on intra-tissue comparisons (e.g., T_{RM} vs T_{EX} within the same tumour or tissue), inherently controlling for tissue-specific processing effects. Although it is

common in the literature to merge datasets from distinct tissues, we intentionally analysed BC and liver samples separately to avoid confounding factors. Notably, the observed transcriptional concordance between T cell states across tissues (BC vs liver) and with published datasets, further supports the biological validity of our observations.

2) The numbers in Fig 4b are very similar. Are the differences statistically significant (e.g. 0.77 vs 0.73)? Are they biologically significant?

The Reviewer accurately points out that the numerical differences in clonal sharing are modest, and we agree that this is a point we had not sufficiently emphasised. In our BC dataset, there is a statistically significant difference in Jaccard dissimilarity between T_{EM}/T_{EX} and T_{RM}/T_{EX} comparisons, although this was not observed in the pan-cancer atlas. We have now added these analyses and improved the visualisation of data spread in **Fig. 4b**. Notably, in all cases, Jaccard values exceeding 0.6 indicate limited clonal overlap, consistent with our interpretation that these populations are largely clonally distinct (also supported by data in **Fig. 4c-d**).

3) Fig 4c: What are the percentages of shared clones between T_{EM} and T_{EX} and between T_{EM} and T_{RM} ?

The percentage of shared clones between T_{EM} and T_{EX} is 24.7%, and between T_{EM} and T_{RM} is 24.6%. These values are now included in the revised Venn diagram in **Fig. 4c**.

4) Fig 4d: The cell numbers of clonal overlap in the three pairs look similar. There is no statistics. It seems that there are limited shared clones among all cell types. Is this correct? The authors should show distribution of clonal size within each subset.

The Reviewer is correct that clonal sharing among subsets is limited, and indeed, this is what we hoped to illustrate in this figure. As the required threshold for overlap increases (from 1 to 3 shared cells), most intersections are lost, highlighting minimal clonal redundancy. We have annotated these frequencies in **Fig. 4d** and now include clonal size distributions per dataset in **Extended Data Fig. 6a** that reflects the distribution of relative clonal size across subsets in **Fig. 4a**.

5) Fig 4h: Any example of the viruses predicted to be recognized? Is it possible to analyze T_{RM} vs T_{EX} phenotype in EBV-specific or CMV specific T cells?

The Reviewer makes an excellent point. We identified TCRs predicted to be specific for CMV, EBV, influenza, and SARS-CoV-2-specific. Data stratified by virus-specificity are now presented in **Extended Data Fig. 6b-c**. Notably, influenza-specific cells were more frequently associated with a T_{RM} phenotype within tumours as compared to EBV or CMV-specific cells.

6) Do T_{RM} and T_{EX} have different localization within the tumor?

This is an important and unresolved question, and we thank the Reviewer for prompting this analysis. Using protein markers that distinguish T_{RM} and T_{EX} in human BC, we used high-dimensional CycIF imaging to determine the spatial distribution of T_{RM} vs T_{EX} -phenotype cells, imaging 47 parameters on slides across 7 BC patients (**Fig. 2f-h**; **Extended Data Fig. 2e-l**).

T_{RM} and T_{EX} cells were broadly co-localised across tumour regions, consistent with the hypothesis that TCR specificity, rather than niche, predominantly dictates their phenotypes. Nonetheless, we found that on average, T_{EX} were situated closer to PanCK⁺ tumour cells, suggesting they are more likely to be responding to tumour-antigen *in situ* and supporting a model whereby proximity to antigen-rich zones promotes T_{EX} differentiation.

7) Do bystander effector/memory T cells preferentially become Trm in tumors compared to other non-lymphoid organs? If so, why? What is the signal that attract bystanders to the tumor?

The Reviewer raises an interesting point regarding how effector/circulating memory T cells enter tumours. We and others have shown that inflammation is sufficient to recruit T cells into peripheral tissues, regardless of TCR specificity (PMID: 22509047; 15485630). We hypothesised that inflammation within tumours would likely recruit T cells through similar means. Specifically, CXCR6 and CXCR3 have been implicated in T cell entry into tissues (PMID: 24162776, 34343496). Thus, we tested whether bystander T cells require CXCR3 and CXCR6 signals to enter AT3-OVA tumours. Using CRISPR/Cas9-mediated deletion, we show that both chemokine receptors are necessary for bystander T cell entry into AT3-OVA tumours (**Extended Data Fig. 8m-n**). These findings support the idea that inflammation-driven chemokine signalling facilitates bystander T cell accumulation and subsequent T_{RM} cell differentiation in tumours.

8) Line 300: “These results show that tumour-derived TPROG, TEX, and TRM cells were more similar to each other and clonally distinct from CD69- and spleen-derived populations.” It seems to contradict Fig 4 where the authors concluded that Trm and Tex are clonally distinct.

We appreciate this opportunity to clarify. In our human datasets, clonal identity is defined by shared TCR sequences. In contrast, SPLINTR lineage tracing uses unique barcodes to track progeny from mono-specific OT-I T cells that express the same TCR. To avoid confusion, we no longer refer to barcoded lineages as “clones” in the SPLINTR context.

Key clarifications are as follows:

1. T_{RM} and T_{EX} cells in human tumours are largely clonally distinct, though rare clonal overlap is observed (**Fig. 4a-d**).
2. We hypothesise that tumour-derived T_{EX} cells (CD39⁺) are predominantly tumour-specific (CD39 has been shown to demarcate tumour-specific cells (PMID: 29769722), while virus-specific cells in tumours more frequently adopt a T_{RM} phenotype (**Fig. 4f-h**). However, we do not assume that there are no tumour-specific T_{RM} within tumour samples, and indeed we add new data to show that putative tumour-specific T_{RM} may be present in tumour-adjacent healthy tissue (**Fig. 4j-l, Extended Data Fig. 6d-e**).
3. In murine tumours, we identified tumour-specific T_{RM}-phenotype cells (**Fig. 5a-j, Extended Data Fig. 7**), prompting us to investigate whether tumour-specific T_{RM} and T_{EX} cells arise from common or distinct precursors.
4. Thus, SPLINTR lineage tracing revealed that tumour-specific effector cells can give rise to both T_{EX} and T_{RM} cells upon tumour entry. We have substantially expanded upon these experiments in our revised manuscript (in response to comments from Reviewer 2) to show that naïve T cells also lack a pre-determined fate (**Fig. 6a-d, Extended Data Fig. 9a-c**).

Minor concerns

1) Please provide statistics in Fig 1c.

We have performed additional analyses for this study to show enrichment of the T_{RM} signature score across the annotated clusters from this dataset (presented in **Extended Data Fig. 1a**) and added statistics to Fig. 1c.

2) Line 197: “CD94, CD101, CD73, and CD161 were specifically enriched in TRM cells in BC but not liver-derived tumours.” Why are the markers of Trm are different in BC vs HCC?

We agree that the differential expression of T_{RM}-associated markers between breast and liver tumours is an important observation. This variation is consistent with published findings showing that T_{RM} exhibit tissue-specific protein expression profiles (PMID: 37392736, 34426691, 39708817). Our results reinforce the notion that individual genes or proteins should not be used in isolation to define T_{RM} populations across different contexts. We now highlight this point more clearly in the revised text (line 426-432).

3) What are the leading-edge genes in Fig 3c?

Selected leading-edge genes are now presented in **Extended Data Fig. 5a-b**. The full gene list is included in the Supplementary information.

Referee #2:

This study by MacKay and colleagues examines tissue resident CD8⁺ memory T cells (TRM) relative to functionally exhausted CD8⁺memory T cells (TEX) in tumors and tumor models, identifying distinguishing features of these cell types. TRM are a memory subset distinguished by their long-term residence in diverse tissues, with predominant populations in mucosal and barrier sites. TRM express a suite of surface markers and a distinct transcriptional profile that enables them to mobilize in tissues, interact with tissue cells and structures (e.g., epithelial cells, collagen), but also mediate the functional responses typical of CD8⁺ T cell memory, including rapid production of effector cytokines such as IFN-gamma, TNF-alpha, and cytolytic mediators (granzymes, perforin). In the context of protective immunity, TRM have been shown in numerous infection models by this group and many others, to provide site-specific protective responses through TRM-derived functions, recruitment of other immune cells to the tissue, and other mechanisms. In this context, the comparison and demonstration that TRM are not functionally exhausted is obvious. However, in human cancers and mouse tumor models, TRM are associated with the tumor and it has also been shown extensively that TEX are among tumor-infiltrating T cells and are targets for checkpoint blockade. Interestingly, TRM express multiple markers that are also expressed by and associated with TEX particularly in chronic infection such as PD-1, CD101, CTLA4, TIGIT, and others, and for this reason, there has been confusion in the field of whether the target population for checkpoint blockade is TRM or TEX (or both).

The authors endeavor to resolve any confusion on what constitutes a resident or exhausted T cell by comparing transcriptional profiles of TRM and TEX from published datasets of mouse LCMV infection models, mouse tumor models, and in a scRNAseq dataset of T cells from

breast tumors (BC) and normal breast tissue (there is a previously published scRNAseq dataset from this group in 2018 where they identified TRM as part of BC and also found them in normal breast tissues but the BC data here appear to be new). They found that human BC contain TRM like in normal breast tissue but that a fraction of the TRM also exhibited a TEX signature marked by higher expression of the transcription factor Tox (shown in many studies from a few years ago to be enhanced in TEX cells and to drive a TEX functional program), the memory-associated transcription factor EOMES, and functional markers like GRMK, associated with CD8 T cells in aging. They then show that the proportion of TEX in the tumor was associated with reduced survival using data from a cancer dataset (like TCGA but called METABRIC) and increased TEX correlated to better responses to Immune checkpoint blockade (ICB) therapy; these data appear to be confirmatory as TEX has long been associated with hyporesponsiveness to tumors and is a target of ICB therapy. The authors then investigate the origin of these TEX cells with a TRM phenotype in clonal analysis from the scRNAseq of the BC dataset and using different mouse models to investigate the antigen specificity and origin of the TEX that get into the tumors and take up residence.

Overall, the experiments are well done, text is succinct and clearly explained, and the findings are of interest. However, the study is mixed in terms of novelty—the first part (Figs. 1-3) largely repeats or repurposes data already published on TRM and TEX, by comparing them to each other and then revealing similar bottom line aspects that have already been reported on TRM and TEX function and associations with cancer survival and ICB responses.

We thank the Reviewer for their thoughtful and constructive feedback. While we agree with several points raised, we respectfully assert that primary aim of our study - the direct side-by-side comparison of T_{RM} and T_{EX} in human tumours - has not previously been addressed. Given the ongoing conflation of these cell types in the literature, including in our earlier work, we believe this study is both timely and necessary.

In our 2018 study with Sherene Loi's group, tumour-infiltrating CD8⁺ T cells were classified as " T_{RM} " based solely on CD103 expression, and without comparison to matched healthy tissue. As we now show, this approach likely conflated bona fide T_{RM} and T_{EX} cells, a misclassification propagated across multiple subsequent studies. As a result, prior associations between CD103⁺ T cells and clinical outcomes such as survival or ICB response could not be confidently attributed to T_{RM} , T_{EX} , or both. This issue has been further compounded by the inappropriate application of murine-derived makers and gene signatures (e.g. CD103 or generic " T_{RM} " classifiers) to human tumour datasets.

A strength of the present study lies in its rigorous experimental design. We generated new CITE-seq datasets from matched tumour and adjacent healthy tissues (breast and liver), processed in parallel, to enable robust and reliable identification of T_{RM} and T_{EX} in human tumour samples. These data allowed us to define the functional and spatial distinctions between these populations, and to assess their respective associations with survival and ICB response. We did not repurpose published datasets for these comparisons, and doing so would have undermined our ability to confidently annotate these cell types. Taken together, we believe our findings reconcile previously conflated populations of CD103⁺ T cells and provide a much-needed framework for accurately interpreting the roles of T_{RM} and T_{EX} in human tumour immunity.

Another concern I have with the first part is the juxtaposition of migration and functional capacity as being features of distinct subsets, when they can and do co-exist in the same cells; tissue residence is a feature of homing and localization and doesn't necessarily reflect a polarized function--this group and others have shown that TRM can mediate different types of functional outputs depending on site, stimulation conditions, and even metabolic pathways.

We thank the Reviewer for this important point. We agree that tissue residency and functional capacity are not mutually exclusive, and that TRM can display diverse functions depending on context. In our study, we do not suggest that migratory potential and functionality are inherently dichotomous traits. Rather, we find that TRM and T_{EX} exhibit similar migration capabilities and localisation in tumours, yet differ markedly in their functional capacity, with TRM displaying heightened cytokine production and cytotoxic potential compared to T_{EX} in the same tissue context.

We also agree that TRM from different tissues can exhibit distinct functional profiles. To control for this, all functional comparisons in our study were restricted to TRM and T_{EX} isolated from the same tissue and processed in parallel, allowing us to attribute observed differences to cell state rather than tissue context.

The latter part of the study (Figures 5 and 6) I found to be more interesting and original; these experiments begin to address the mechanisms for the distinct functional capacity of tissue homing T cells that take up residence in tumors. While they presented compelling results, they weren't fully developed. See below for specific points.

We thank the Reviewer for their enthusiasm in our mechanistic experiments and have substantially expanded upon these experiments in our revised manuscript. Specific experiments are outlined below.

1. Many of the results and analysis in the first three figures address similar questions and comparisons as done by Gavil, et al, 2023. The lead author of the present manuscript also recently published a comparison of epigenetic profiles between TRM and T_{EX} in Immunity, revealing gene expression differences between TRM and T_{EX} similar to those shown here in Figs. 1 and 2.

The study by Gavil et al was highly important for the field, showing the non-recirculating nature of tumour-specific T_{EX} in murine tumours. Similarly, our Immunity paper compared TRM across tissue to T_{EX} from the spleen in chronic LCMV infection, highlighting the overlapping features between TRM and T_{EX} populations in mice. However, neither study addressed whether bona fide TRM exist in human tumours, nor how they may be distinguished from T_{EX} in this setting.

Given this overlap, it remained entirely plausible that bona fide TRM are absent from tumours altogether, and that previously reported 'tumour-associated TRM' represent exhausted T cells misclassified on the basis of limited markers such as CD103. Indeed, the assumption that murine-derived markers and gene signatures translate directly to human tumours has contributed to misclassification in the literature, including in our own prior work. Our study directly addresses this by generating new CITE-seq datasets from matched tumour and adjacent healthy tissues, enabling robust and parallel identification of TRM and T_{EX} across human

tumours. This allowed us not only to demonstrate the presence of bona fide T_{RM} in human tumours, but to define their distinct transcriptional and functional features, and to evaluate their respective associations with survival and ICB response. In this way, our study builds on and extends prior work, while resolving long-standing ambiguity in the field.

The genes upregulated in T_{EX} versus T_{RM} such as *GZMK* and lower *PRF1*, etc, are similar to those genes upregulated in $CD8^+$ T cells associated with aging (PMID 33271118). It may be interesting to compare the T_{EX} -specific genes to this aging signature.

We have compared our signatures to the aging-specific genes from this dataset (PMID: 33271118) and confirm the assumptions of the Reviewer, showing that there is overlap between our BC T_{EX} gene signature and the aging signature from this study (**Fig. R1**). We have discussed this in the revised manuscript (line 426-429)

2. The description and conclusion of the TCR clonal analysis in Fig. 4 is not consistent with the results. The authors start out by stating that there is very little clonal overlap between T_{RM} and T_{EX} in the BC dataset in part (a) and (b). However, the Venn diagram in part (c) shows considerable clone sharing between T_{EX} and T_{RM} ; out of 263 T_{EX} clones, 57 of them or 21.6% overlap with T_{RM} clones. Moreover, there appears to be even more overlap of T_{EX} clones with T_{EM} cells in BC—presumably those that derived from tumor-infiltrating T cells. It would be informative to show the derivation of the expanded T_{EX} clones as overlapping with T_{RM} or T_{EM} ; these data are important because they indicate that T_{EX} is a functional state due to the environment and antigen that can arise from cells with different migration abilities. The role of tissue and environment in driving these different functional programs was shown in the Gavil, 2023 study.

We thank the Reviewer for this helpful comment and apologise for the confusion in our original description of these data. We have now clarified this section and expanded our analysis to include clonal overlap across T_{RM} , T_{EM} , and T_{EX} (**Fig. 4c**). As correctly noted by the Reviewer, T_{EX} clones in our BC dataset show greater overlap with T_{EM} than with T_{RM} , consistent with lower Jaccard dissimilarity scores. However, this was not observed in the pan-cancer dataset, where clonal overlap between populations was more comparable (**Fig. 4b**). Indeed, the Venn diagram in Fig. 4c shows shared clones between T_{RM} and T_{EX} , however, our key point here was to show that this sharing is relatively limited and largely driven by a single shared cell. This is more clearly illustrated when thresholds for clonal sharing are increased as displayed in **Fig. 4d**.

To further explore these clonal relationships, we performed additional analyses using a publicly available dataset with extensive TCR coverage from colorectal, uterine, lung and renal cancers, including matched tumour and adjacent non-cancerous tissue (PMID: 32103181). In these data, we found multiple instances of T cell clones shared between tumour-derived T_{EX} and T_{RM} residing in adjacent healthy tissue (**Fig. 4j-l, Extended Data Fig. 6d-e**). These findings support the Reviewer's suggestion that T_{EX} is a functional state dictated by environment and antigen abundance.

3. The mouse experiments with different TCR specificities and fate mapping in Figures 5 and 6 show interesting findings. That bystander (non-tumor reactive) T cells migrate more extensively to the tumor compared to antigen reactive ones and adopt a TRM signature is very interesting. Why do the non-tumor reactive cells migrate to the tumor so extensively? Are they also in the spleen or other organs and is their migration and TRM phenotypes due to TGF-beta? How does this finding relate to low affinity also promoting TRM, and other studies that show that antigen drives T cells into tissues? Interestingly, the results are similar to previous findings that chronic antigen stimulation reduced TRM maintenance (PMID 22504644).

Extent of bystander T cell migration into tumours: We do not necessarily find that bystander T cells migrate more extensively into tumours compared to antigen-reactive cells, as it is difficult to disentangle the relative contributions of circulating precursor frequency, *in situ* proliferation, and migratory efficiency. Rather, we show in **Fig. 5m** and **Extended Data Fig. 8i-j** that, when bystander T cells do enter the tumour, they are more likely to adopt a T_{RM} phenotype compared to tumour-specific cells.

Mechanism of bystander entry into tumours: The Reviewer raises an important point regarding how bystander T cells are recruited into tumours. We and others have shown that effector and circulating memory T cells can enter inflamed tissues in an antigen-independent manner (e.g. PMID: 22509047). In line with this, we hypothesised that inflammation within the TME may drive bystander T cell recruitment. To test this, we assessed the requirement of CXCR3 and CXCR6 in mediating bystander T cell entry. As shown in **Extended Data Fig. 8m-n**, both chemokine receptors are required for efficient bystander T cell entry into tumours. CXCR3-deficient T cells were unaffected in the spleen, whereas CXCR6 deficiency impaired cells in both spleen and tumours, predominantly in the tumour (**Extended Data Fig. 8m-n**).

Low-affinity T cells and T_{RM} differentiation: We agree with the Reviewer that low-affinity T cells have a greater propensity to adopt a T_{RM} phenotype. In **Fig. 5k-l**, we show that low-affinity T cells are more likely to become T_{RM}, likely reflecting differences in developmental fate rather than differences in tumour entry, given that tumour-agnostic T cells efficiently enter tumours.

Role of TGFβ in bystander T_{RM} phenotype: We also agree that TGF-β may drive the T_{RM} phenotype in bystander T cells, as previously described for tumour-specific T_{RM} (PMID: 36827978). To test this, we transferred OT-I:TGFβRII^{-/-} effector T cells into mice bearing AT3 tumours lacking OVA and observed a marked reduction in CD103⁺ T_{RM}-phenotype cells (**Extended Data Fig. 8k-l**), consistent with TGFβ-dependent T_{RM} cell differentiation.

4. The fate mapping system is beautiful but I found the second part using re-transfer into tumor bearing hosts to be confusing. How durable is the initial fate of the cells in same host if they wait longer?

We thank the Reviewer for their positive comments on our fate mapping approach and the opportunity the clarity the retransfer experiments.

To assess the durability of T_{RM} and T_{EX} phenotypes within the same host, we were constrained by animal ethics requirements that limit tumour size to 1000mm³, restricting longitudinal analysis to 3-4 weeks post-inoculation. To extend this observation period, we performed retransfer experiments in which T_{RM} and T_{EX} OT-I cells isolated from late-stage tumours (day 24-28) were transferred into earlier-stage tumours (day 10). This approach enabled prolonged tracking of cell fate within the tumour microenvironment.

We also thank the Reviewer for highlighting the need to improve clarity in this section. In response, we have revised the manuscript text and performed several additional experiments to strengthen the mechanistic insight provided by these data. These include:

1. ***Temporal phenotypic changes in situ:*** We now present time-course data (**Fig. 5h-j**) showing that the frequency of T_{RM} phenotype cells increases over time and is inversely correlated with tumour size at early time-points, and cognate-antigen load at later times. Conversely, Tox^{Hi} T_{EX} cells are positively associated high cognate antigen load.
2. ***Phenotype plasticity following retransfer:*** In our original retransfer experiments, OT-I TIL populations were sorted from late-stage tumours and transferred intravenously into secondary tumour-bearing hosts (input, **Extended Data Fig. 9d-h**). Due to low cell recovery after retransfer in these experiments, phenotypic assessment was performed on pooled populations, which on average resembled primary T_{EX} -phenotype cells (output, **Extended Data Fig. 9g-h**).

Given that both T_{RM} and T_{EX} cells downregulate the transcriptional machinery for trafficking through the blood and lymph, it was possible that these cells fail to efficiently migrate to tumours when transferred via the IV route. This raised the possibility that differences in migratory capacity rather than intrinsic plasticity underpinned the difference in cell numbers following IV transfer of T_{RM} vs T_{EX} phenotype cells. To address this, we performed new experiments where T_{RM} and T_{EX} -phenotype OT-I TIL were directly co-transferred into secondary tumours that either lacked or expressed the OVA antigen (**Fig. 6e-i**).

These data revealed that:

- Most T_{RM} -phenotype cells maintain their identity in the absence of antigen, but a substantial proportion adopt features of T cell exhaustion when retransferred into antigen-expressing secondary tumours (**Fig. 6f-i**), consistent with analyses performed with intravenously retransferred T_{RM} cells (**Extended Data Fig. 9e-f**).
- In tumours lacking antigen, ~50% of T_{EX} -like phenotype cells acquired T_{RM} phenotype while ~50% retained the T_{EX} -like phenotype (**Fig. 6f-g**), suggesting heterogeneity within this population. We suspect that the Tox^{Lo} T_{EX} -like population gives rise to T_{RM} -

phenotype cells, however, in the absence of a Tox-reporter mouse we could not directly assess this and have been careful not to overinterpret this finding in the text.

- When transferred intravenously, more T_{RM}-phenotype cells can be isolated from secondary tumours than T_{EX}-phenotype cells (**Extended Data Fig. 9e-f**). However, this difference was not observed following direct intratumoural transfer (**Extended Data Fig. 9i**), suggesting that T_{RM} phenotype cells may have a greater ability to recirculate and populate distal sites compared to T_{EX}-phenotype cells.
3. **SPLINTR tracing from naïve precursors.** While our prior SPLINTR-barcoding experiments demonstrated that effector T cells of shared origin could give rise to both T_{RM} and T_{EX} populations in tumours (**Fig. 6a,d; Extended Data Fig. b-c**), this did not preclude the possibility that naïve T cells might be pre-committed towards one fate over the other. To test this, we developed an approach to SPLINTR-barcode naïve T cells via intra-thymic transfer of barcoded T cell progenitors, enabling lineage tracing from the earliest stages of T cell development. These new experiments revealed that intratumoural T_{RM} and T_{EX} cells arise from shared naïve precursors (**Fig. 6a-c**), reinforcing the conclusion that these populations share a developmental origin and are more closely related to one another than to circulating T cells (**Fig. 6a-d; Extended Data Fig. 9a-c**).

In sum, we believe the extensive new experiments and analyses presented in Figures 5 and 6 address the Reviewer's concern that these ideas were not fully developed, and that their inclusion substantially enhances our study.

Referee #3:

Burn et al. explore the relationship and potential roles of Trm and Tex TILs. The authors nicely illustrate the transcriptional differences between various TILs in multiple mouse and human disease contexts, demonstrating the limitations of previously published Trm gene signatures. While published Tex gene signatures reliably identify Tex cells to the exclusion of Trm cells, some published Trm cell gene signatures are non-specific and fail to distinguish Trm and Tex cells – highlighting a danger in using poorly validated gene signatures as a proxy for specific biological properties. Burn et al. re-derive Trm and Tex gene signatures. In other words, certain correlates are used to define Trm and different correlates are used to define Tex, and then as would logically follow, Trm and Tex are shown to be different based on these correlates. These new gene signatures correlate differently with overall survival in cancer patients, reconfirming in some scenarios that TRM-associated signatures correlate with survival. Notably, the author's Tex gene signature, but not the Trm gene signature, is a moderate predictor of ICB response in a cohort of breast cancer patients. Comparing clonal overlap between Trm and Tex cells, the authors suggest that Trm cells in tumors that do not exhibit transcriptional signatures of chronic-stimulation (a.k.a. Tex signatures) are likely bystanders in the TME. Interestingly, the authors investigate the ontogeny and differentiation potential of CD103⁺ Tex cells. They provide evidence that reduced antigen-load and/or TCR avidity correlates with tumor-specific T cell expression of CD103. They also demonstrate that CD103⁺ tumor-specific T cells have greater plasticity than CD103-CD39⁺ Tex-term cells. The paper brings together sophisticated analyses to diverse data sets, which is certainly impressive. However, some issues diminished enthusiasm. Showing that Trm and Tex express different gene signatures was an inevitable

result from the authors' analytical approach. While one could argue that the study provides an improved signature that could distinguish what could be labeled Trm vs. Tex, it was not clear what the underlying biological definition of these terms were, which limited its value. In addition, the abstract concludes that "These results reveal the unique roles of Trm and Tex cells in tumor control, underscoring the need for distinct strategies to harness T cell populations in novel cancer therapies," but the role of Trm was not revealed, nor why the strong Trm correlation with survival exists, and what (if anything) about Trm should be harnessed in novel cancer therapies.

We thank the Reviewer for their thoughtful comments on our study. We appreciate the concern regarding the clarity of our definitions for T_{RM} and T_{EX} , as well as the need for greater insight into the functional relevance of T_{RM} and their association with long-term survival. In response, we have clarified our criteria for annotating T_{RM} and T_{EX} , added new analyses to explore potential contributions of T_{RM} to tumour control, and revised key sections of the manuscript text (including the abstract) to more accurately reflect the scope and limitations of our findings.

Defining and annotating T_{RM} and T_{EX} populations

In brief, we use the following operational definitions for defining cell types:

- i. T cell exhaustion – a functional impairment of T cells
- ii. T cell memory – persistence in the absence of antigen or ongoing antigenic stimulation
- iii. T cell residency – a lack of migratory capacity

Putative T_{RM} and T_{EX} populations are initially identified based on enrichment of transcriptional signatures (in human data) or phenotypic markers (in mouse models). We functionally validated these annotations by demonstrating that T_{EX} -annotated cells are impaired in effector function, while T_{RM} -annotated cells remain functional and persist in antigen-free contexts.

In human tumours, direct experimental validation of residency is not feasible. Therefore, we annotated putative T_{RM} and T_{EX} populations using expression of genes and signatures derived from murine models, and cross-validated in human transplant settings. We have added further description our approach to the manuscript (line 115-121). These annotations were supported by functional data showing T_{EX} cells exhibited reduced functional potential (**Fig. 2i-k, Extended Data Fig. 3a-d**), while memory-phenotype cells were enriched in putatively antigen-free contexts, either as bystanders or within non-malignant tissues (**Fig. 4f-h,j-l**).

We disagree that the distinction between T_{RM} and T_{EX} was an inevitable conclusion of our analytical approach. Our goal was to determine whether these populations could be distinguished within the same tumour microenvironment. Indeed, prior literature left open the possibility that all TILs expressing T_{RM} -associated signatures might also exhibit T_{EX} -associated features, and that no tumour-derived populations would transcriptionally align with T_{RM} -like cells from non-malignant tissues. The identification of distinct populations was therefore an empirical outcome rather than an inevitable result.

In mouse models we performed high-dimensional flow cytometry to first phenotypically annotate antigen-specific T cell clusters (**Fig. 5a-d, Extended Data Fig. 7a**). Then we used indirect measures, including intravascular labelling (**Extended Data Fig. 7b**), and FTY720 treatment (**Extended Data Fig. 7c**) to show that almost all intratumoural populations are in disequilibrium with the blood, supporting that these cells are resident. We emphasise in the text

that these are indirect measures of residency (line 291-293). We further assess the functionality of distinct T_{RM} populations (**Fig. 5g; Extended Data Fig. 7d-e**) and correlate the T_{RM}-phenotype with loss of cognate antigen (**Fig. 5h-j**) corroborating the annotations of exhaustion and memory per our operational definitions. We hope that these data, together with the revised text, provide a clearer rationale for the terminology used throughout the manuscript.

The role of T_{RM} in tumour control

Understanding the role of T_{RM} in tumours has long proven to be a difficult hurdle, particularly in human studies. The association between T_{RM} gene expression and long-term patient survival likely reflects a combination of non-mutually exclusive mechanisms, as detailed below, that are difficult to disentangle experimentally. While we acknowledge that we have not explicitly shown what T_{RM} do, our data provide insights that may suggest their functional role and will form the basis of future studies. The Reviewer correctly points out that the original abstract and discussion overstated this aspect, and we have revised the text to reflect a more nuanced interpretation of our findings.

We propose two plausible explanations for the observed association between T_{RM} gene expression and improved patient survival, both of which consider the nature of the datasets analysed (e.g. METABRIC), which are developed from sequencing surgically resected tumours rather than longitudinal biopsies. This has the important implication that some excised tumours may contain greater regions of healthy or non-malignant tissue than others. Accordingly, the association of T_{RM} and survival could reflect the following:

1. T_{RM} development is associated with the effective clearance of cognate antigen (**Fig. 5i-j**). Following surgical resection, tumour-specific T_{RM} may persist in the tissue and suppress the outgrowth of residual cancer cell, as suggested by our previous murine studies (PMIDs: 30598548, 36827978).
2. T_{RM} typically form following the resolution of inflammation. Thus, high expression of the T_{RM} gene signature in resected tumour samples may reflect effective immune control at the time of resection. In this context, the presence of T_{RM} and non-malignant tissue may serve as a surrogate marker of effective tumour control and better prognosis. We have clarified this point in the discussion, consistent with interpretations from recent reviews (PMID: 39142275, 37821656).

To further test the hypothesis that tumour-specific T_{RM} are present in tissues adjacent to the tumour, we analysed a dataset with extensive TCR sequencing depth from both tumours and surrounding healthy tissue. We found that TCRs expressed by T_{EX} in tumours (likely to be tumour-specific) are frequently shared with T_{RM} in the adjacent non-malignant tissue (PMID: 32103181; annotated using our gene signatures – **Fig. 4j-l; Extended Data Fig. 6d-e**). These findings support a model in which tumour-specific T_{RM} cells are maintained following tumour clearance and may contribute to durable local immune surveillance.

Specific comments:

The authors should very clearly state their exact definitions of Trm and Tex. E.g., were Trm defined as a memory T cell type based primarily on markers or some universal signature (e.g. line 91), or is Trm meant to signify a non-circulating behaviour of a (memory?) T cell? Were published Trm gene signatures based on resting memory cells that were defined by migration properties?

We have amended the manuscript to clarify these definitions and refer the Reviewer to our more detailed response above. Conceptually, we define both T_{RM} and T_{EX} as non-recirculating (i.e., resident) T cell populations that remain within tissues due to expression or repression of genes associated with tissue retention. T_{RM} are further distinguished by their classification as memory T cells that are not currently experiencing cognate antigen stimulation, whereas T_{EX} are defined by persistent antigen exposure leading to functional impairment.

Given the constraints of working with human tumour samples, where direct assessment of recirculation or antigen exposure is not feasible, we employed a strategy to annotate T_{RM} and T_{EX} based on gene signatures and markers derived from murine studies that defined T_{RM} as resident in prior experiments. We appreciate that the RNA sequencing performed on T_{RM} in the Mackay et al, Science study (PMID: 27102484) was not performed on cells isolated from parabionts or transplanted tissues. However, the populations that contributed to the 'core T_{RM} gene signature' in this study ($CD69^+CD103^+$ phenotype cells from the skin and SI-IEL, and $CD69^+CD62L^-$ cells from the liver following resolution of acute infection) have been experimentally confirmed to be resident in functional assays. We also used this signature because it has been widely adopted in the literature to identify T_{RM} in tumours, which is the flawed approach we set out to critically assess.

Overall, we believe that this strategy allowed us to define these populations in a biologically meaningful way, while acknowledging the limitations of these annotations.

It is hard to know if the gene lists presented would be useful and accurate outside of the relatively few models used. Indeed, by figure 5, when the group uses their newly defined T_{RM} and T_{EX} gene signatures (defined in human breast cancer) in the mouse B16 melanoma model, the signatures are helpful on average but look to be imprecise on a sizeable fraction of cells.

The Reviewer makes an excellent point about the need for caution when applying gene signatures across datasets, particularly as their utility depends on the populations used to define them in the original setting. We have now stated this explicitly in the discussion (line 429-432). This context-dependence has indeed contributed to the incorrect annotation of T_{RM} and T_{EX} in several published studies. However, we respectfully disagree with the interpretation that our signatures are imprecise. Using these gene signatures, we are able to resolve distinct populations of T_{RM} and T_{EX} from several distinct human tumours from the Pan-cancer T cell atlas (PMID: 34914499). In the revised manuscript, we have expanded upon this point and have extended our analyses to additional datasets (**Fig. 3h, Extended Data Fig. 5c**; PMID: 32103181, 30388456, 31359002, 29988129, 30979687, 29942094), further supporting the broader utility of our signatures across human tumours.

Regarding the mouse data, we agree that variable expression of signatures across species is to be expected, as it is well understood that individual genes are differentially regulated across mice and humans, including often utilised genes for T_{RM} identification such as *ZNF683* (HOBIT) and *ITGAE* (CD103). Nonetheless, the analysis in **Fig. 5o-r; Extended Data Fig. 8o-q** compares the relative enrichment of these signatures across bystander and antigen-specific cells in mouse models (including B16 melanoma and published murine BC, PMID: 39142275). These data support the notion that TCR signalling contributes to the phenotypic distinction between T_{RM} and T_{EX} cells in tumours.

The authors miscommunicated the central messages of REF 16. This study showed that resident cells (as defined by migration property) 1) exist in tumors, but that 2) conventional markers of residence did NOT correlate well with migration properties in tumors, particularly when assessing tumor-specific T cells. No causality between Trm gene signatures and residence within tumors was shown.

We agree with the Reviewer's interpretation of this important study by the Masopust group and apologise for our earlier misstatement. We have revised our statement to reflect that causality between T_{RM} gene signatures and the migration properties of TILs was not shown.

Figure 1a: Please reconsider the phrasing of this sentence, which uses "Figure 1a" as the sentence subject instead of the analysis represented in the figure.

We have adjusted the text as suggested.

Figure 1c + 1f: No statistical test is presented to demonstrate significant TRM gene score enrichment. All that is demonstrated is that >50% of cells score above 0 for TEX and virusspecific cells in MFP/Tumour. Please consider including Ext Fig 1a in the main figure or adding the TEX-term cells as a separate population in the Fig 1c violin plot. This would clarify the bimodal distribution seen in CL13-D30 and clarify the authors' conclusions.

We thank the Reviewer for this helpful suggestion and we have now incorporated Extended Data Fig. 1a into the updated main **Fig. 1d**. We agree that the CL13-D30 population exhibits a bimodal distribution, and we performed additional analysis that confirm the T_{RM} gene signature is specifically enriched in the terminally exhausted T cells (Exh-Term) population, as shown in new **Extended Data Fig. 1a**. We have also included statistical comparisons for **Fig. 1c and Fig. 1f** to support the observed differences in gene signature enrichment.

The authors have recently proposed that CD103 expression defined a subset of Trm that was less functional than other subsets of Trm (Christo et al., NI 2021). Burn et al., used CD103 expression as a criteria for calling a cell a Trm. Where do CD103⁻ Trm fit into the larger picture? Because Fig 2 shows cytokine production from CD103⁺ Trm, how best to reconcile these results with the Christo study?

This is an important point, and we appreciate the opportunity to clarify. In our human breast cancer dataset, both T_{RM} and T_{EX} -annotated populations express CD103. We did not observe a distinct population of CD103⁻ T_{RM} , and most CD69⁺CD103⁻ cells expressed recirculation-associated genes such as *KLF2*, suggesting they are unlikely to represent bona fide resident populations.

In the Christo study (NI 2021), we found that CD103⁺ and CD103⁻ T_{RM} populations in murine tissues exhibited functional differences shaped by TGF β imprinting. To investigate whether such differences extend to human tissue, we analysed CD69⁺CD103⁺ and CD69⁺CD103⁻ T cells from normal human breast tissue and found comparable cytokine production (**Fig. R2a**), indicating that functional differences observed in mice may be tissue-specific. Furthermore, in murine tumours, CD103⁺ T_{RM} -phenotype cells exhibited greater functionality than other antigen-specific TIL populations (**Fig. R2b**; also shown in **Fig. 5g, Extended Data Fig. 7d-e**).

Thus, while we support the assertion that CD103⁺ TIL populations are functional, we do not make claims about their relative functionality compared to CD69⁺CD103⁻ T_{RM} cells that may exist in other tissues, as we anticipate substantial context-dependent variation between distinct tumours and tissue sites.

Fig. R2: *a*, PMA/Ionomycin restimulation of CD45RA⁺CD8⁺ T cells from human breast tissue, gated on CD69⁺CD103^{+/-}. *b*, PMA/Ionomycin restimulation of OT-I or endogenous T cells from mouse AT3-OVA tumours

Figure 2b-c: Please report the sensitivity and specificity of each newly derived signature to convey their utility in reliably identifying each cell-type/subset. While the box plots convey some information, it is clear that some cells still exhibit enrichment of genes from alternative subsets (e.g. T_{RM} that express T_{EX} gene signature and vice versa)

We have now performed sensitivity and specificity analyses as suggested, revealing that both the T_{RM} and T_{EX} gene signatures are highly sensitive and specific, as reflected by high AUC values (**Fig. R3**). While we agree that individual cells can show varying degrees of enrichment for both signatures, we do not annotate cells individually. Rather, we annotate clusters based on the relative enrichment of each gene signatures. We also reiterate that caution should be exercised when applying gene signatures to annotate cell types in scRNA-seq data, including our own new signatures, as their performance depends on the context of the data. We have now explicitly stated this in the discussion (line 429-432).

Fig. R3. Sensitivity and specificity analysis of T_{RM} and T_{EX} gene signatures as determined by ROC and area under the curve (AUC) where AUC = 1 suggests perfect sensitivity/specificity.

Ext Fig 2a-c: no in text reference to this extended data

We apologise for this oversight and have addressed this in the revised manuscript.

Fig 2j: Please provide a more substantial explanation of the ROC analysis to improve interpretability for the reader. What is the source data? What is the prognostic predictor/subset used to represent TRM-signature and TEX-signature? What constitutes treatment response? For such a critical point in the paper, the provided explanation of the analysis is scant.

A very fair critique. We have now expanded the description of this analysis (updated **Fig. 2m**) and referred to the source data in both the main text and figure legends.

Specifically, the ROC analyses compare the abilities of the T_{RM} and T_{EX} signatures to predict improvement in pathologic complete response (pCR) rate in breast cancer patients treated with anti-PD-1 immune checkpoint blockade (ICB), using data from the I-SPY trial (PMID: 35623341). The data includes 69 patients in the pembrolizumab arm and 210 controls. pCR data were obtained from Supplemental Table S2 of the original study (PMID: 35623341). Patient-level biomarker scores, subtype classifications, and clinical/response data were as reported in that study. Signature scores were calculated from the normalized microarray data as described in our Methods section.

For each ROC curve, the vertical axis represents Sensitivity = True Positive Rate (TPR), and the horizontal axis represents $1 - \text{Specificity} = \text{False Positive Rate (FPR)}$. The curves correspond to increasingly strong signature scores, with each point representing a threshold at which (FPR, TPR) is calculated. As shown, an increasing T_{RM} signature does not predict improved pCR in the ICB-treated cohort compared to controls (**Fig 2m**, left panel), whereas a stronger T_{EX} signature is associated with a significantly higher pCR rate in patients treated with the ICB (**Fig 2m**, right panel).

Figure 3: The clustering methodology appears to outperform the use of gene signatures for identifying cell populations. What is the advantage or value of gene signatures when clustering reliably enables cell annotation? Also, is this not a circular approach to defining T cell subsets? Are there better methods to cross-validated gene signatures as a means to determine cell identity?

We agree that careful interpretation is needed when using gene signatures for cell annotation. To clarify, we first performed unsupervised clustering of the respective datasets without reference to gene signature expression. Cell-type labels were then assigned to clusters based on the enrichment of defined signatures. We would argue that this is a standard approach for annotating cell types in scRNA-seq and avoids circularity, since clustering is independent of the gene sets used for annotation. We apologise for the confusion and have clarified this methodology to ensure the transparency of our approach.

Figure 5: The use of “Tex-like” and “Trm-like” TEX cells is confusing and seems contradictory to the overarching conclusions of figures 1-4 that emphasize the difference between TRM and TEX cells. It appears the only distinction in this figure (and figure 6) is to study the differences between CD103+ and CD103- TEX cells.

Thank you for pointing out the confusion here. We have provided an extensive amount of new data (Fig. 5a-j, Extended Data Fig. 7 & 8a-h) and have revised the wording throughout Figures 5 and 6 to more clearly define the respective cell types in the mouse AT3 BC tumours, as described in our initial response to this Reviewer.

Figure 5q-r: While the mean gene signature enrichment scores appear significantly different, the significant overlap in enrichment scores between Ag-specific and bystander scores warrants acknowledgment and discussion. This overlap suggests that these gene signatures have limited sensitivity and specificity.

This is a valuable point, and we have taken care to not overstate the conclusions from this analysis. As described above, it is expected that there is not a perfect alignment of signatures across humans and mice. Nonetheless, we believe observed trend remains valid; overall, the T_{EX} and T_{RM} signatures from human BC are differentially enriched based on the respective presence or absence of intratumoural cognate antigen, which is supportive of the rest of the data in this figure. We also wish to highlight that this analysis was performed on aggregated antigen-specific and bystander TIL populations, rather than specific clusters. While we could incorporate granular cluster-level analyses if required, a deep annotation of the different cell types within melanoma tumours was not the goal of this analysis, which was to assess the relative enrichment of T_{EX} and T_{RM} signatures in bystander versus tumour-specific T cells.

Figure 5o-r: Clarify figure legend description to describe time between cell transfer and isolation for scRNA-seq. Transfer 14-20 days after tumor cell inoculation and then isolation 14-20 days after cell transfer?

We apologise for this lack of clarity. The timing of T cell transfer and tumour growth reflects the nature of the epicutaneous B16 melanoma model, whereby tumour growth occurs over a variable timeframe, and some tumours are controlled entirely by endogenous skin T_{RM} (PMID: 305985548). We have now amended the methods section to more clearly describe this system. Briefly, T cell transfer was performed upon visual observation of tumour growth (~14-20 days after tumour inoculation), and mice were then culled at a later timepoint when tumours reached their ethical endpoint (1000mm³, ~14-20 d after T cell transfer). Together, this accounts for the range of the timeframes reported.

Figure 6 is confusing with the use of the ‘TRM’ label, which appears to be a stand in for CD103+ TEX cells.

We apologise for this confusion. We have performed further experiments and provided an in-depth description of the different TIL populations that we assess here, as detailed in our earlier response to the Reviewer.

Discussion: The authors state that “Our study for the first time reconciles the respective roles of two critical T cell subsets associated with tumour control, namely CD8+ TRM and TEX cells.” This seems like an overstatement. This study also does not really show what Trm do.

We agree with this assessment and have altered this and other statements to reflect the correlative limitations of our data.